# A new confuciusornithid bird with a secondary epiphyseal ossification reveals phylogenetic changes in confuciusornithid flight mode

Renfei Wang[1,2], Dongyu Hu [2✉], Meisheng Zhang[1], Shiying Wang[2], Qi Zhao[3], Corwin Sullivan[4,5] & Xing Xu[2,3,6✉]

The confuciusornithids are the earliest known beaked birds, and constitute the only species-rich clade of Early Cretaceous pygostylian birds that existed prior to the cladogenesis of Ornithothoraces. Here, we report a new confuciusornithid species from the Lower Cretaceous of western Liaoning, northeastern China. Compared to other confuciusornithids, this new species and the recently reported *Yangavis confucii* both show evidence of stronger flight capability, although the wings of the two taxa differ from one another in many respects. Our aerodynamic analyses under phylogeny indicate that varying modes of flight adaptation emerged across the diversity of confuciusornithids, and to a lesser degree over the course of their ontogeny, and specifically suggest that both a trend towards improved flight capability and a change in flight strategy occurred in confuciusornithid evolution. The new confuciusornithid differs most saliently from other Mesozoic birds in having an extra cushion-like bone in the first digit of the wing, a highly unusual feature that may have helped to meet the functional demands of flight at a stage when skeletal growth was still incomplete. The new find strikingly exemplifies the morphological, developmental and functional diversity of the first beaked birds.

[1] College of Earth Sciences, Jilin University, Changchun, China. [2] Shenyang Normal University, Paleontological Museum of Liaoning, Key Laboratory for Evolution of Past Life in Northeast Asia, Liaoning Province, Shenyang, China. [3] Key Laboratory of Vertebrate Evolution and Human Origins, Institute of Vertebrate Paleontology and Paleoanthropology, Chinese Academy of Sciences, Beijing, China. [4] Department of Biological Sciences, University of Alberta, Edmonton, AB, Canada. [5] Philip J. Currie Dinosaur Museum, Wembley, AB, Canada. [6] Center for Vertebrate Evolutionary Biology, Yunnan University, Kunming, China. ✉email: hudongyu@synu.edu.cn; xuxing@ivpp.ac.cn

Confuciusornithidae is a clade of Early Cretaceous pygostylian birds known from the Jehol Biota of East Asia[1], and represents the earliest known toothless, beaked birds. Five genera and eleven species, recovered from the Dabeigou, Yixian and Jiufotang formations (~135–120 Ma), have been described and assigned to this family, though the validity of some species is questionable[2–6]. Confuciusornithids are the only species-rich pygostylian clade known to have existed prior to the cladogenesis of Ornithothoraces, and are represented by thousands of exceptionally preserved specimens that collectively provide rich information on confuciusornithid morphology, taxonomy, flight ability, growth, diet, and ecology[3,5,7–13]. Here, we report a new confuciusornithid species, *Confuciusornis shifan* sp. nov., from the Jiufotang Formation. *Confuciusornis shifan* differs from other confuciusornithids in a number of morphological and developmental features, which have implications for understanding confuciusornithid taxonomic diversity, morphological disparity, development, and flight behavior.

## Results

**Systematic paleontology**. Avialae Gauthier, 1986
  Pygostylia Chiappe, 2002
  Confuciusornithidae Hou et al., 1995
  *Confuciusornis* Hou et al., 1995
  *Confuciusornis shifan* sp. nov.

**Etymology**. The specific name is derived from the Mandarin "shifan", meaning a paragon of all teachers, in honor of Confucius. The name also commemorates the 70th anniversary of Shenyang Normal University (Shenyang Shifan Daxue).

**Holotype**. PMoL-AB00178, a nearly complete and mostly articulated skeleton, preserved on a single slab (Fig. 1; Supplementary Table 1).

**Locality and horizon**. Xiaotaizi Village, Lamadong Town, Jianchang County, Liaoning Province, China; second unit of Lower Cretaceous Jiufotang Formation (119 Ma)[14,15].

**Diagnosis**. Referable to Confuciusornithidae based on the presence of the following confuciusornithid features[5]: upper and lower jaws edentulous and beaked; mandible with large rostral and small caudal external mandibular fenestrae; boomerang-shaped furcula without a hypocleidium; coracoid and scapula fused into a scapulocoracoid; humeral deltopectoral crest triangular with sharp distodorsal corner; intermediate phalanx of major digit bowed; and claw of major digit significantly reduced. Distinguishable from other confuciusornithids in having the following features (*indicates autapomorphies): ventral process of surangular absent (process present in *Confuciusornis sanctus* and *Eoconfuciusornis zhengi*) and splenial centrally perforated*; synsacrum keeled*; angle of 75° between scapula and coracoid (compared to 65° in *Yangavis confucii*, and 90° in *C. sanctus* and *Changchengornis hengdaoziensis*); coracoid/scapula length ratio of 0.56 (compared to 0.51 in *C. sanctus*, 0.53 in *Ch. hengdaoziensis*, and 0.55 in *Y. confucii*); scapula widens caudal to midshaft region, then tapers toward caudal end; proximal part of humeral shaft strongly curved; deltopectoral crest with a relatively small height/length ratio of 0.36 (compared to 0.39 in *E. zhengi*, 0.41 in *Confuciusornis dui*, and larger than 0.6 in *C. sanctus*, *Ch. hengdaoziensis* and *Y. confucii*); extensor process cranially projected nearly half the width of distal articular surface of alular metacarpal*; minor metacarpal slightly bowed caudally, and intermetacarpal space between major and minor metacarpals wider than minor metacarpal; distal third of pubis strongly curved caudally*; proximal end of metatarsal III transversely compressed between metatarsals II and IV*; and metatarsal IV with a lateral flange*. Differs further from *C. dui* in having rostral part of ventral margin of dentary convex, maxilla with a halberd-shaped dorsal process, and lacrimal perforated by a large oval foramen; from *E. zhengi* in having thoracic vertebrae with lateral excavations, and deltopectoral crest perforated; from *Ch. hengdaoziensis* in having clavicular symphysis without a caudal tubercle, alular metacarpal one third the length of major metacarpal (one half in *Ch. hengdaoziensis*), intermediate phalanx of major digit curved, intermediate phalanx of major digit longer than proximal phalanx (opposite condition in *Ch. hengdaoziensis*), and hallux less than half the length of pedal digit II (hallux proportionally longer in *Ch. hengdaoziensis*); and from *Y. confucii* in having

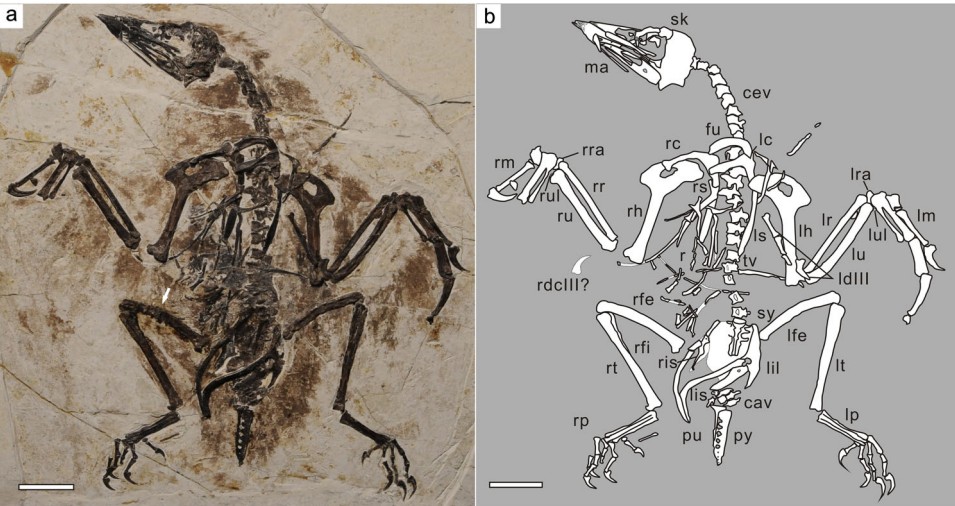

**Fig. 1 *Confuciusornis shifan* holotype (PMoL-AB00178). a** Photograph. **b** Schematic line drawing. *cav* caudal vertebra, *cev* cervical vertebra, *fu* furcula, *ga* gastralia, *lc* left coracoid, *ldIII* left manual digit III, *lfe* left femur, *lh* left humerus, *lil* left ilium, *lis* left ischium, *lm* left manus, *lp* left pes, *lra* left radiale, *lr* left radius, *ls* left scapula, *lt* left tibiotarsus, *lu* left ulna, *lul* left ulnare, *ma* mandible, *pu* pubis, *py* pygostyle, *r* rib, *rc* right coracoid, *rdcIII* claw of right manual digit III, *rfe* right femur, *rfi* right fibula, *rh* right humerus, *ris* right ischium, *rm* right manus, *rp* right pes, *rra* right radiale, *rr* right radius, *rs* right scapula, *rt* right tibiotarsus, *ru* right ulna, *rul* right ulnare, *sk* skull, *sy* synsacrum, *tv* thoracic vertebra. Arrow indicates the position from which the histological section of the right femur was taken. Scale bars: 2 cm.

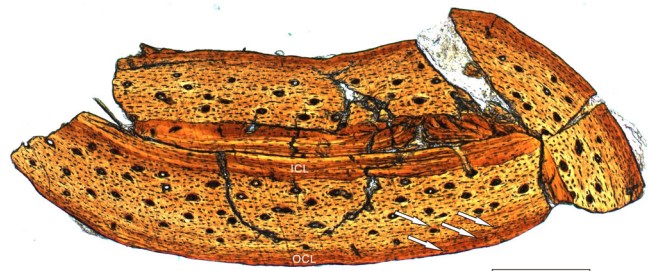

**Fig. 2 Osteohistological section of *Confuciusornis shifan* holotype (PMoL-AB00178).** Sample was taken from the midshaft of the right femur. *ICL* inner circumferential layer, *OCL* outer circumferential layer. Arrows indicate LAGs. Scale bar: 100 μm.

proportionally short forelimb (length ratio of humerus+ulna to femur+tibiotarsus 1.02, compared to 1.19 in *Y. confucii*), major digit claw reduced, hallux less than half the length of pedal digit II, and pedal digit III shorter than tarsometatarsus.

**Ontogenetic assessment.** PMoL-AB00178 is an adult individual, as indicated by a combination of fusion features. All neurocentral sutures are closed without any trace, the sacral vertebrae are completely fused together to form a synsacrum, and nearly all of the compound bones (e.g., the metacarpus, tibiotarsus, and metatarsus) are fully formed, although the pygostyle retains visible intercentral foramina.

To further test this ontogenetic assessment using osteohistological data, the right femur was sectioned at the mid-diaphysis (Fig. 1a). The femoral compact bone shows a triple-layered structure (Fig. 2). The inner circumferential layer (ICL, or endosteal bone) is formed by poorly vascularized lamellar bone with well-organized flattened osteocyte lacunae, which is demarcated from the primary periosteal bone by a distinctly uneven reversal line. The outer circumferential layer (OCL, or external fundamental system) consists of avascular parallel-fibered bone with osteocyte lacunae that are again flattened, but are less regularly arranged than in the ICL. The ICL and OCL each account for about one sixth of the cortical thickness (Fig. 2). The thick middle layer is dominated by fibrolamellar bone with abundant longitudinal primary osteons, and shows no indications of remodeling. One line of arrested growth (LAG) occurs in the middle layer, close to the OCL. The diameter of the vascular canals decreases towards the OCL, as does the abundance of both osteons and simple primary vascular canals. Based on the presence of three additional LAGs in the OCL, in addition to the well-developed nature of the ICL and OCL, this individual is referable to Histology Age Class V (the ontogenetically oldest stage) as defined for *C. sanctus*[13], and at the time of death had already completed its rapid growth phase and transitioned to very slow growth[16].

**Description and comparisons.** PMoL-AB00178 is smaller than most known confuciusornithid specimens, even though it represents one of the few such specimens assignable to the oldest ontogenetic stage. PMoL-AB00178 has an estimated body mass (BM) of 174 g based on a multivariate equation[17], while most other known adult or subadult confuciusornithid specimens have greater estimated body masses (BMs), ranging up to 801 g for the largest known individual of *Confuciusornis*[17,18] (also see Supplementary Table 3). PMoL-AB00178 thus belongs to a truly small-bodied confuciusornithid species. However, *Ch. hengdaoziensis* is even smaller, the only known specimen having an estimated BM of only 138 g (Supplementary Table 3).

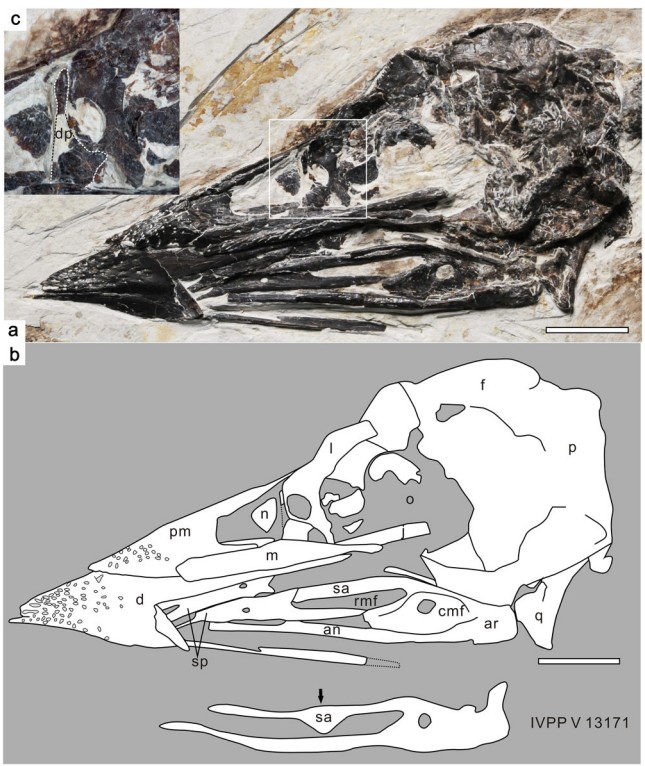

**Fig. 3 Skull and mandible of *Confuciusornis shifan* holotype (PMoL-AB00178). a** Photograph. **b** Schematic line drawing. **c** close-up of the anterior margin of the orbit. *an* angular, *ar* articular, *cmf* caudal mandibular fenestra, *d* dentary, *dp* dorsal process of maxilla, *f* frontal, *j* jugal, *l* lacrimal, *m* maxilla, *n* nasal, *o* orbit, *p* parietal, *pm* premaxilla, *q* quadrate, *rmf* rostral mandibular fenestra, *sa* surangular, *sp* splenial. White rectangle in **a** indicates the region shown in **c**. A line drawing of the posterior half of the mandible of *Confuciusornis sanctus* IVPP V 13171 (drafted based on Fig. 1b in Wang et al.[5]) is presented below the line drawing of the skull and mandible of *Confuciusornis shifan*; arrow indicates the ventral process of the surangular. Scale bars: 1 cm; note that **c** is not to scale.

The skull and the left mandible have been strongly rotated, so that they are exposed in ventrolateral view, whereas most of the right mandible is exposed in medial view (Fig. 3a, b). The rostrum is robust and pointed, with toothless upper and lower jaws as in other confuciusornithids[3–6]. As in *C. sanctus*[3,19], the elongate frontal process of the premaxilla extends dorsocaudally at least far enough to reach the level of the rostral margin of the orbit, and the short maxillary process overlies the premaxillary process of the left maxilla. The premaxillary process of the left maxilla appears conspicuously taller and slightly longer than the jugal process. The jugal process is transversely thickened to form a broad contact with the rostral end of the jugal. Two pieces of bone that are in tight mutual contact, and indeed appear to be partially fused, form the rostral margin of the orbit (Fig. 3c); the smaller, ventrorostrally positioned piece is identified as the dorsal process of the left maxilla, and the larger dorsocaudally positioned piece as the left lacrimal. Although the dorsal process is incomplete, a halberd-shaped outline can be inferred based on the preserved part of this structure and the impression of the missing part. Such a halberd-shaped dorsal process was previously described as an autapomorphy of *C. sanctus*[5], the equivalent process being triangular in *C. dui*, rod-like in *E. zhengi* and rectangular in *Y. confucii*[6]. The lacrimal appears strip-like, and its middle part is perforated by a large oval foramen as suggested by Wang et al.[5] (Fig. 3c). The perforated bone at the rostral margin of the confuciusornithid orbit has been identified

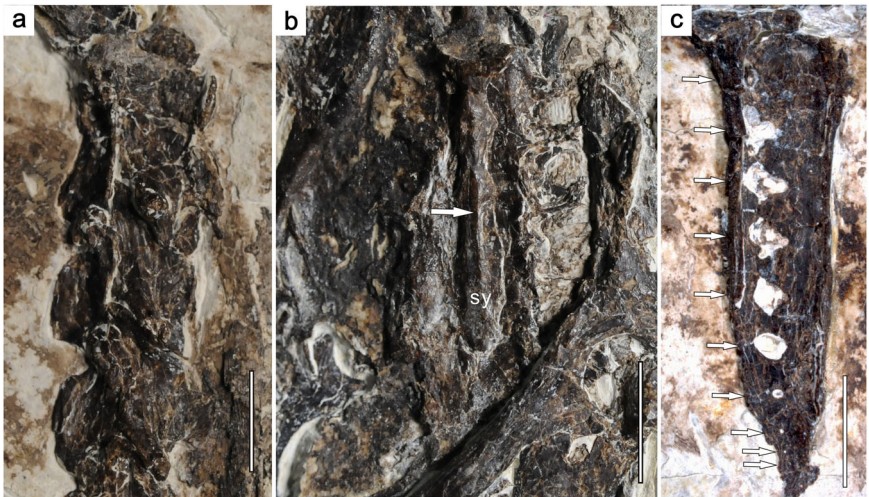

**Fig. 4 Selected axial elements of *Confuciusornis shifan* holotype (PMoL-AB00178). a** cervical vertebrae. **b** synsacrum. **c** pygostyle. *sy* synsacrum. Arrows in **b** and **c** indicate the longitudinal ridge on the ventral surface of the synsacrum and the positions of foramina along the pygostyle, respectively. Also see Supplementary Fig. 1 for a close-up of the distal part of the pygostyle. Scale bars: 0.5 cm in **a**, **c**; 1 cm in **b**.

as the dorsal process of the maxilla[3] or as an ethmoidolacrimal complex[19], but clearly represents the lacrimal in PMoL-AB00178 as in *C. sanctus* IVPP V 13168[5]. The abruptly tapered part of the dorsal process of the maxilla is separated from the rostral margin of the foramen only by a thin strip of the lacrimal. The jugal is represented only by the rostral part of the left element, but appears lower and transversely thicker than the maxilla. The frontal forms a laterally projecting supraorbital flange, the caudal part of which is particularly prominent. The otic process of the right quadrate, which has shifted slightly ventrally so that most of its medial surface is exposed, is much longer than the mandibular process.

The left and right dentaries are almost completely fused with each other along an extended, ventrocaudally inclined mandibular symphysis as in *C. sanctus*[3] (Fig. 3a, b). The transversely compressed ventrocaudal process of the right dentary is visible, but the caudal end of the process is missing. The impression of the missing end suggests that the ventrocaudal process originally contacted the angular laterally, and extended back to the level of the caudal margin of the anterior mandibular fenestra. The right angular and right surangular both appear rod-shaped, and their caudal ends are fused with the articular. The surangular appears to lack a ventral process as in *C. dui*, indicating that the presence of a triangular ventral surangular process in *C. sanctus* is indeed a diagnostic feature of this species[5] (Fig. 3b). The articular prominently projects medially to form a medial articular facet for the quadrate[19]. The left splenial is displaced and partly overlapped by the left dentary, whereas the right splenial is nearly complete and preserved close to its original position. The splenial is plate-like, and forked both rostrally and caudally. The ventral margin of the splenial is thickened along most of its length. The caudodorsal process of the splenial overlaps the medial surface of the surangular. The caudoventral process, which does not contact the angular, protrudes into the rostral half of the rostral mandibular fenestra and gently tapers caudally to contact the ventral process of the surangular. Unlike in *C. sanctus*[3,5] and *Y. confucii*[6], the splenial of *C. shifan* is centrally perforated by an oval foramen (Fig. 3a, b).

The cervical vertebrae are exposed in ventrolateral view, revealing that the cranial and caudal articular surfaces of each centrum are distinctly heterocoelous (Fig. 4a) as previously reported for *Confuciusornis*[5]. The centra of the thoracic series are laterally excavated as in *C. sanctus*, whereas such excavations are

absent in *Y. confucii*[6] (Fig. 1). Each of the two cranialmost thoracic centra bears a prominent, tapered ventral keel. The sacral vertebrae are completely fused with each other to form a synsacrum (Fig. 4b). Based on the number of visible transverse processes, the synsacrum consists of seven vertebrae as in other confuciusornithids[3–6], and the ventral portion of the series of fused centra is transversely compressed to form a longitudinal ridge as in some enantiornithine ornithothoracines[20] (Fig. 4b). By contrast, the synsacrum of *C. sanctus* DNHM D2454 (holotype of the putative species '*Confuciusornis feducciai*') appears dorsoventrally compressed and transversely widened[5]. Five free caudal vertebrae are visible between the synsacrum and the pygostyle (Fig. 1). Additional free caudal vertebrae are likely present, but obscured by the left ischium and some soft tissue. The caudal vertebrae appear significantly shorter than the thoracic ones. The pygostyle of *C. shifan* appears to be exposed in left lateral view (Fig. 4c). The pygostyle is proportionately robust and subequal in length to the tarsometatarsus as in most confuciusornithids[3,21], whereas the pygostyle is considerably longer than the tarsometatarsus in *C. dui*, the length ratio between the two elements being 1.18[21]. The transversely thickened ventral margin of the pygostyle is made up of the co-ossified centra. Both the ventral and dorsal margins gently taper caudally rather than widening caudally as in *C. sanctus* IVPP V 12352 (holotype of the putative species '*Jinzhouornis zhangjiyingia*')[5,21] and *C. dui*[22], and the ventral margin does not bear a longitudinal keel as in *C. sanctus* IVPP V 12352. Some 10 small round foramina form a longitudinal row along the whole lateral furrow of the pygostyle (Fig. 4c, also see Supplementary Fig. 1). The foramina are closer to the ventral margin of the pygostyle than the dorsal margin, and gradually become smaller and more tightly crowded together towards the caudal end of the element. Similar foramina occur in other confuciusornithid specimens, but are fewer in number and confined to the cranial part of the pygostyle[21,23].

The coracoids are fused with the scapulae to form scapulocoracoids. The left and right scapulocoracoids are exposed in lateral and medial views, respectively (Fig. 5a). The long axes of the scapula and coracoid define an angle of 75°. The equivalent angle is similar in *Y. confucii*[6] (65°) and ornithothoracine birds[24] (Figs. 5a and 6), but approximately 90° in *C. sanctus* and *Ch. hengdaoziensis*[3]. A laterally directed glenoid facet is visible on the left scapulocoracoid. The cranial and caudal margins of the glenoid facet project more strongly than the dorsal and ventral

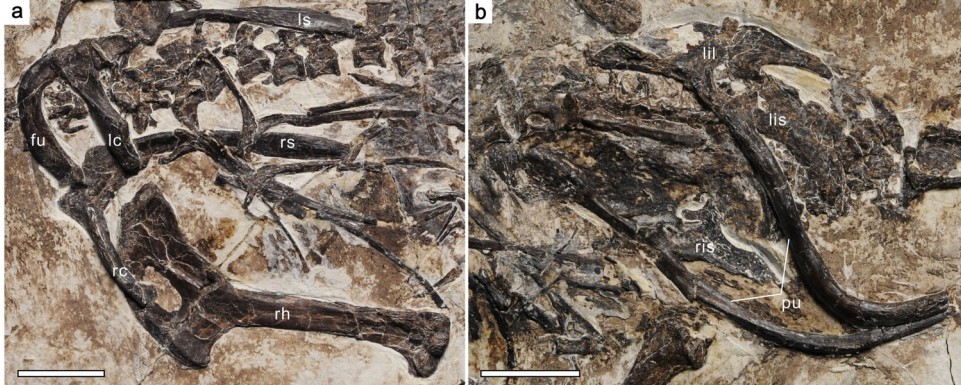

**Fig. 5 Pectoral and pelvic girdles of *Confuciusornis shifan* holotype (PMoL-AB00178). a** Pectoral girdle. **b** pelvic girdle. *fu* furcula, *lc* left coracoid, *lil* left ilium, *lis* left ischium, *ls* left scapula, *pu* pubis, *rc* right coracoid, *rh* right humerus, *ris* right ischium, *rs* right scapula. Scale bars: 1 cm.

ones, implying that the configuration of the facet probably did not strongly limit the elevation and depression of the wing in flapping flight. The acromion of the scapula is well-developed. The shaft of the scapula curves slightly downward; the cranial half of the shaft appears rod-like and slightly compressed dorsoventrally, while the caudal half is more compressed mediolaterally and widens significantly in the dorsoventral direction before tapering to a point as in ornithothoracine birds[25] (Figs. 5a and 6). In other confuciusornithids, the scapula is generally straight, and neither flares partway along its length nor tapers caudally[3]. The coracoid is strut-like and approximately half the length of the scapula. Its proximal end appears dorsoventrally thicker than transversely wide and projects slightly cranially to form an acrocoracoid process as in *Y. confucii*[6], whereas its distal end is compressed dorsoventrally. The acrocoracoid process is not well-developed in *C. sanctus*[3]. The lateral surface of the coracoid is excavated to form a shallow groove, which gradually becomes narrower and shallower distally as in *Y. confucii*[6]. The medial margin of the coracoid is strongly compressed to form a longitudinal keel, except at the proximal end of the bone. The caudal surface of the furcula is mostly exposed. As in other confuciusornithids, the furcula is boomerang-shaped and without a distinct tubercle[3–6] (Figs. 5a and 6). The proximomedial corner of the right clavicular ramus is inflated caudally to form a bulbous structure that contacts the medial face of the proximal part of the right scapulocoracoid; its proximolateral corner is craniocaudally compressed and projects slightly in the lateral direction, forming a fossa. Such a configuration of the proximal end of the clavicular ramus is not seen in other confuciusornithids.

The forelimb is subequal in length to the hindlimb, with a forelimb (humerus+ulna+carpometacarpus)/hindlimb (femur +tibiotarsus+tarsometatarsus) ratio of 1.02 (Fig. 1; Supplementary Table 1). The equivalent ratio is 0.91 in *C. hengdaoziensis*, 0.95 in *E. zhengi*, 1.04 in *C. dui*, 0.98–1.04 in *C. sanctus* and 1.20 in *Y. confucii*[6] (Fig. 6; Supplementary Table 2-2). The proximal part of the ventral edge of the humeral shaft is strongly curved, and the humerus is typical of confuciusornithids in having an expanded and perforated deltopectoral crest. The height/length ratio of the deltopectoral crest is 0.36, and the highest point is at the distal end of the crest; note that the (proximodistal) length of the crest is defined here as the distance between the proximalmost and distalmost points of the crest along the humeral shaft, whereas the (dorsoventral) height of the crest is defined as the total dorsoventral height of the humerus at the point of the crest's greatest dorsal prominence, minus the dorsoventral diameter of the humeral shaft just distal to the crest. The equivalent height/ length ratio is 0.39 in *E. zhengi*[4], 0.41 in *C. dui*[22], and larger than 0.6 in *C. sanctus, Ch. hengdaoziensis*[3] and *Y. confucii*[6], and in all

these taxa the highest point is in the middle of the crest (Figs. 5a and 6). The dorsodistal corner of the deltopectoral crest projects distally, so that the distal margin appears concave. Distally, a large fossa for the brachialis muscle is present proximal to the dorsal and ventral condyles. The dorsal epicondyle is better developed than the ventral epicondyle, and the former projects sharply dorsally.

Both the ulna and the radius are shorter than the humerus, as in other confuciusornithids (Figs. 1 and 6; Supplementary Tables 1, 2-2). A concave ventral cotyle, a flat dorsal cotyle and a concave incisura radialis are clearly visible on the proximal end of the right ulna, whereas a distinct olecranon process is lacking as in *C. sanctus*[3] and *Y. confucii*[6]. The proximal humeral articular facet of the radius is flat, and a well-developed bicipital tubercle is situated on the cranial face of the proximal end of the bone. The bicipital tubercle is small in *C. sanctus*, and absent altogether in *Y. confucii*[6]. Both radialia are in their original positions relative to the radii and carpometacarpi (Fig. 1). The left radiale is exposed in proximopalmar view and the right in distopalmar view. The articular facets for the radius and semilunate carpal are both distinctly concave. The cranial surface of the radiale is convex and forms a small tubercle on the proximal side, and the caudal surface is slightly concave and directed somewhat proximally to articulate with the ulna.

The semilunate carpal is completely fused with metacarpals II and III to form a carpometacarpus (Figs. 1 and 7a, b). The proximal end of the carpometacarpus forms a pulley-like carpal trochlea as in *C. sanctus*[3] and extant birds. A distinct fossa is present in the ventral surface of the proximal end, and the bulbous pisiform process is located distal to the fossa. Metacarpal III is much more slender, and slightly shorter, than metacarpal II, and contact between these metacarpals is limited to their proximal ends. The middle parts of metacarpals II and III are constricted, and the narrowest part of metacarpal III is less than half the width of the narrowest part of metacarpal II, as in other confuciusornithids. However, the intermetacarpal space appears relatively longer and significantly wider than in other confuciusornithids, as in some ornithothoracine birds[25]. The alular metacarpal is about one third the length of metacarpal II as in *C. sanctus*[3] and *E. zhengi*[4], rather than half the length as in *Ch. hengdaoziensis*[3] (Figs. 6 and 7a, b), and is not fused with the carpometacarpus (Fig. 7a, b). The caudal portion of the alular metacarpal extends farther proximally than the cranial portion, and the proximal surface of the metacarpal is concave, forming a cranial carpal fovea. The proximal two thirds of the cranial margin projects cranially to form the extensor process. The craniocaudal width of the process is nearly half that of the distal articular surface of the alular metacarpal (Fig. 7a–c). Such well-developed pisiform and

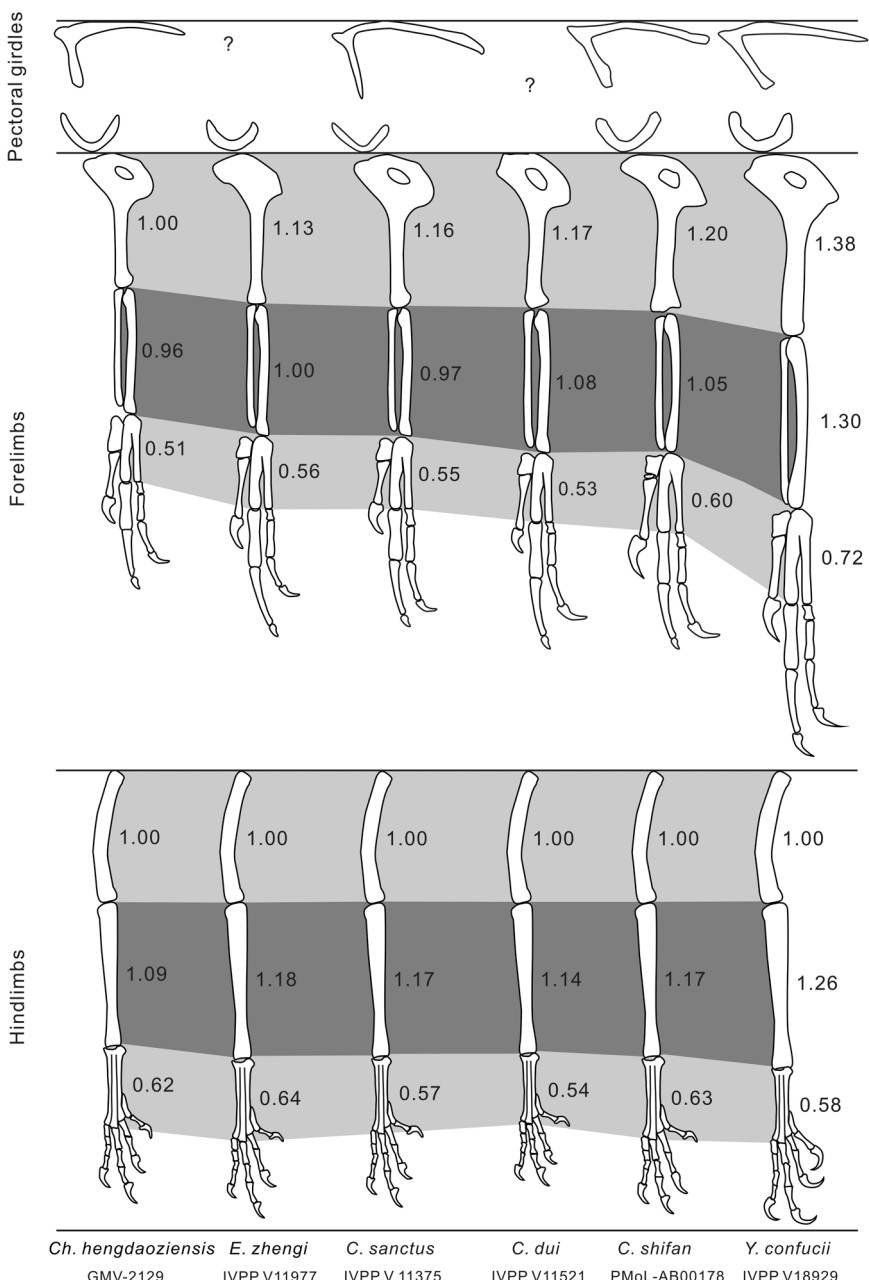

**Fig. 6 Relative proportions of the pectoral girdle and limb elements of the *Confuciusornis shifan* holotype (PMoL-AB00178) and other confuciusornithids.** Different gray shades indicate relative lengths of stylopodial, zeugopodial and metapodial limb segments from top to bottom, respectively. Values near the segments represent ratios of segment length to femoral length. All drawings scaled to a common, arbitrary femoral length.

extensor processes are seen only in the contemporaneous enantiornithine *Xiangornis*[26], a few Early[25,27] and Late Cretaceous ornithuromorph ornithothoracines[28], and extant flying birds (Fig. 7a, b). Unlike in previously known confuciusornithids, the caudal distal condyle of the alular metacarpal is well-developed, and projects more distoventrally than the rest of the metacarpal, as is evident in palmar view. The proximal phalanx of digit II is robust, maintains a constant diameter throughout its entire length, and is slightly shorter than the intermediate phalanx as in other confuciusornithids except *Ch. hengdaoziensis*[3] (Fig. 1), in which the proximal phalanx is slightly longer than the intermediate one[3]. The ungual phalanges of the alular digit and digit III are large and curved, whereas that of digit II is small as in most confuciusornithids. In *Y. confucii*, however, the ungual phalanx of digit II is large[6].

Although the left and right pelvic girdles are fully articulated, only the pubes are relatively completely preserved (Fig. 5b). The right ilium appears relatively low, with a slightly convex dorsal margin and a laterally projecting antitrochanter. The pubis, remarkably, is longer than the femur, being rod-like, slender, and strongly retroverted. Because of a sharp bend in the pubic shaft, the long axis of the pubic symphysis is nearly perpendicular to that of the shaft's proximal portion, a condition resembling that seen in derived ornithuromorph birds[29].

The femoral head, neck and trochanteric crest are well-developed. The proximal tarsals are completely fused with each other and with the tibia, forming a tibiotarsus. Similarly, the proximal ends of metatarsals II–IV are fused with each other and with the distal tarsals, forming a tarsometatarsus. The proximal end of metatarsal III appears transversely compressed between

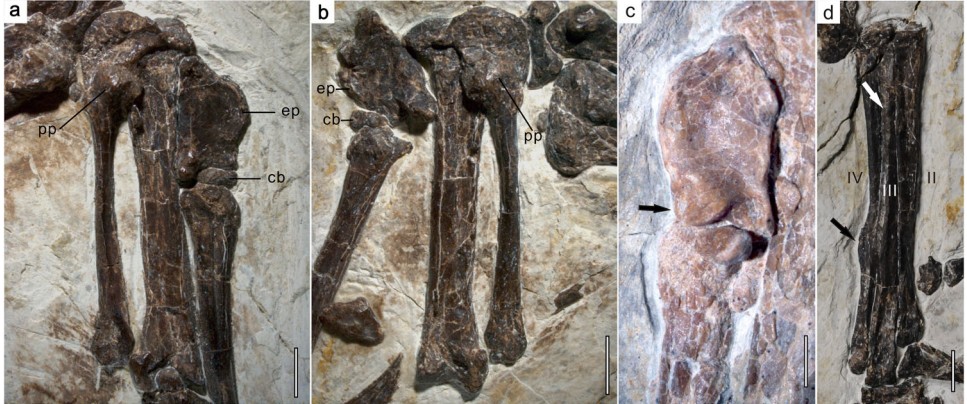

**Fig. 7 Selected limb elements of *Confuciusornis shifan* holotype (PMoL-AB00178). a** left carpometacarpus. **b** right carpometacarpus. **c** left alular metacarpal of the confuciusornithid specimen PMoL-AB00150. **d** right tarsometatarsus. Carpometacarpi are in palmar view, and tarsometatarsus is in cranial view. *cb* cushion-like bone, *ep* extensor process, *pp* pisiform process. Roman numerals in (**d**) identify metatarsals. Black arrow in (**c**) indicates the cranial distal condyle of the alular metacarpal; black and white arrows in (**d**) indicate the ridge-like process on metatarsal IV and the dorsal tubercle on metatarsal III, respectively. Scale bars: 0.25 cm.

metatarsals II and IV in cranial view (Fig. 7d), whereas in other confuciusornithids metatarsal III maintains a constant width along its length[3–6]. An oval tubercle, presumably for insertion of *M. tibialis cranialis*, is located on the dorsal surface near the proximal end as in *C. sanctus*[3], but no similar structure exists on metatarsal II. A lateral flange is present on the distal third of metatarsal IV (Fig. 7d), unlike in other confuciusornithids. In pedal digits III–IV, the penultimate phalanx is longer than the preceding phalanx, as in other confuciusornithids[6]. However, the penultimate phalanx of pedal digit II is subequal in length to the preceding phalanx.

**Secondary epiphyseal ossification**. The most unusual skeletal feature exhibited by PMoL-AB00178 is the presence of a cushion-like small bone between the distal end of the alular metacarpal and the proximal end of the first alular phalanx (Fig. 7a, b). In the left manus, the small bone contacts the cranial portions of the opposing articular surfaces of the metacarpal and phalanx (Fig. 7a); in the right manus, the bone again contacts the cranial portion of the articular surface of the first alular phalanx, but rests against the cranial margin of the alular metacarpal due to slight disarticulation of the alular digit (Fig. 7b). We identify this small bone as an ossification from a secondary ossification center (SOC) within the distal epiphysis of the alular metacarpal, equivalent to the cranial condyle of the distal articular surface seen in other confuciusornithids (Fig. 7c). This identification is based on the following evidence: (1) in the left manus, the small bone remains in apparently natural alignment with the alular metacarpal, but not with the first alular phalanx; (2) the distal articular surface of the alular metacarpal is ginglymoid in other confuciusornithids[6], and the caudal condyle protrudes slightly more distally than the cranial one (Fig. 7c), whereas in PMoL-AB00178 the caudal condyles of both alular metacarpals are slightly more distally prominent than the small bone and typical cranial condyles are absent; and (3) the proximal articular surfaces of both first alular phalanges of PMoL-AB00178 are similar in morphology to those of other confuciusornithids, and a well-developed flexor tubercle is visible on the left alular phalanx, implying that the proximal articulars of the first alular phalanges had well-developed.

## Discussion

Our phylogenetic analysis places *C. shifan* as the sister taxon to *C. sanctus*, and *E. zhengi*, *C. dui*, *Y. confucii* and *Ch. hengdaoziensis*

as successive outgroups to the *C. sanctus* + *C. shifan* clade (Fig. 8). This confuciusornithid phylogeny is largely consistent with those obtained in most recent studies[5,6]. However, the various confuciusornithid subclades recovered in our study are not well supported, as indicated by their low Bremer (1 or 3) and Bootstrap (<30%) values (Fig. 8). It is noteworthy that *C. dui* is placed relatively far from the *C. sanctus* + *C. shifan* clade, and that no features are uniquely shared between *C. sanctus* and *C. shifan*. In fact, the genus *Confuciusornis* has never been well diagnosed. Collectively, the results suggest that additional taxonomic work will be needed to refine the definition and diagnosis of *Confuciusornis*, and in particular to determine whether or not *C. dui* and perhaps even *C. shifan* should be moved out of the genus. However, such work would require in-depth morphological study of a wide range of confuciusornithid material including all reported confuciusornithid specimens, and is consequently beyond the scope of the present paper. Accordingly, we choose not to make any nomenclatural revisions herein.

**Evolution of pygostyles among early birds**. The discovery of *C. shifan* sheds new light on morphological evolution and flight adaptation in early pygostylians, including confuciusornithid birds. The pygostyle is a key diagnostic feature of the clade Pygostylia, and is also an important structure for flight. Among early pygostylians, confuciusornithids, jinguofortisids[30] and the majority of enantiornithines have a proportionally long, robust and rod-shaped pygostyle[21], and this is particularly true of confuciusornithids. Sapeornithids, pengornithid enantiornithines, and ornithuromorphs, by contrast, have a proportionally short, plough-shaped pygostyle[21,23]. The relatively long pygostyles of the former set of taxa have been suggested to consist of a relatively large number of caudal vertebrae[23], but the exact number of caudal vertebrae involved in forming the pygostyle is typically difficult to determine due to their complete coossification. However, a minimum vertebral count can be determined in the case of perforated pygostyles, like that of PMoL-AB00178 (Fig. 4c, also see Supplementary figure S1). Such pygostyles are thought to be the result of incomplete fusion between adjacent neural arches, leaving the intervertebral foramina surrounded, but not completely filled in with new bone[21,23]. Given the presence of 10 preserved foramina, the number of caudal vertebrae forming the pygostyle is estimated to be at least 11 in PMoL-AB00178, greater than the eight pygostyle-forming caudals reported for a subadult of *C. sanctus*[23]. Among other early pygostylians, the number of

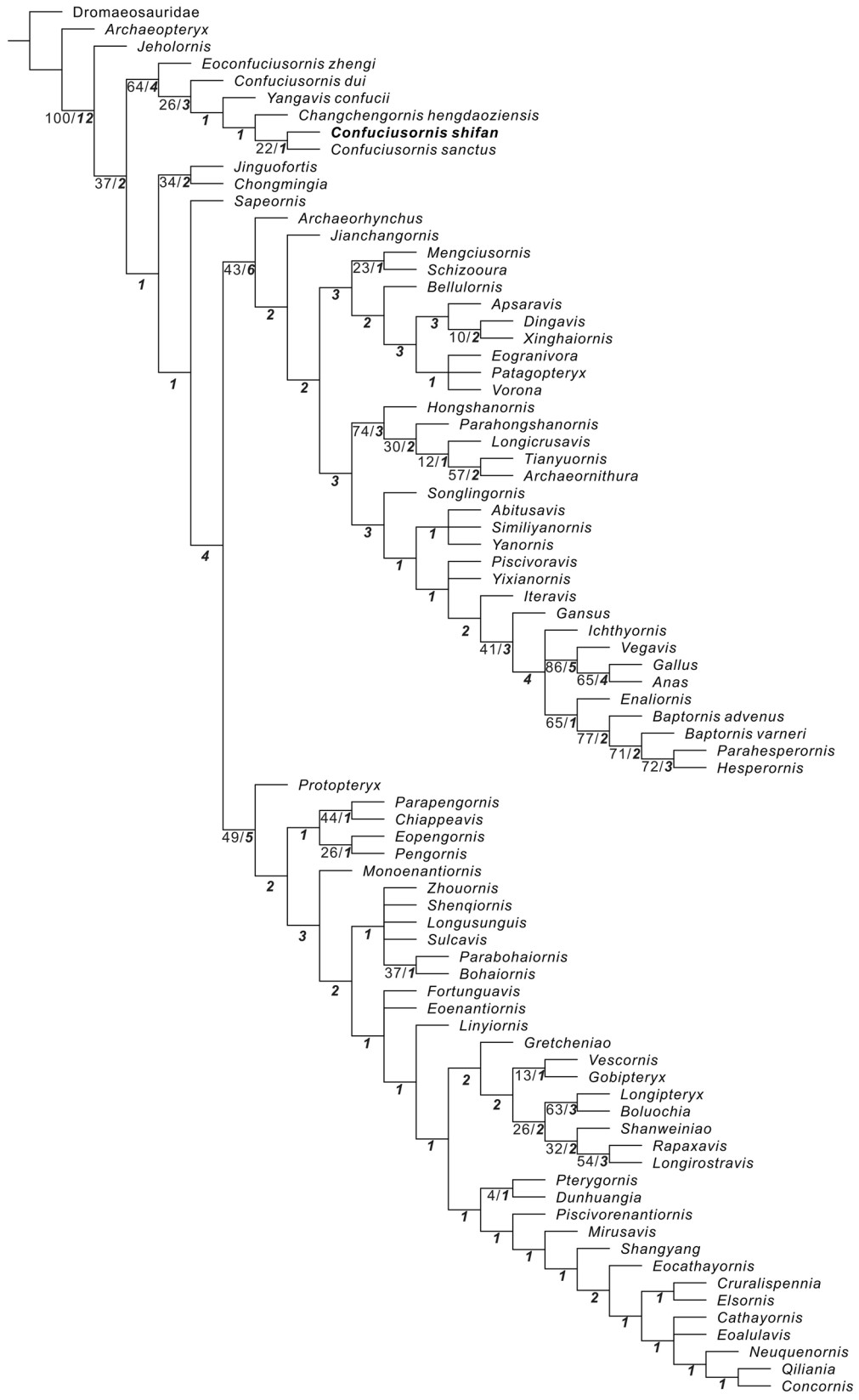

**Fig. 8 Cladogram of Mesozoic birds showing the systematic position of *Confuciusornis shifan*.** The strict consensus of the 192 most parsimonious trees recovered in the phylogenetic analysis performed in this study (length = 1404; consistency index = 0.277; retention index = 0.667). Bootstrap and Bremer values over the minimum threshold are given in normal font and bold italic font, respectively, near the nodes to which they pertain.

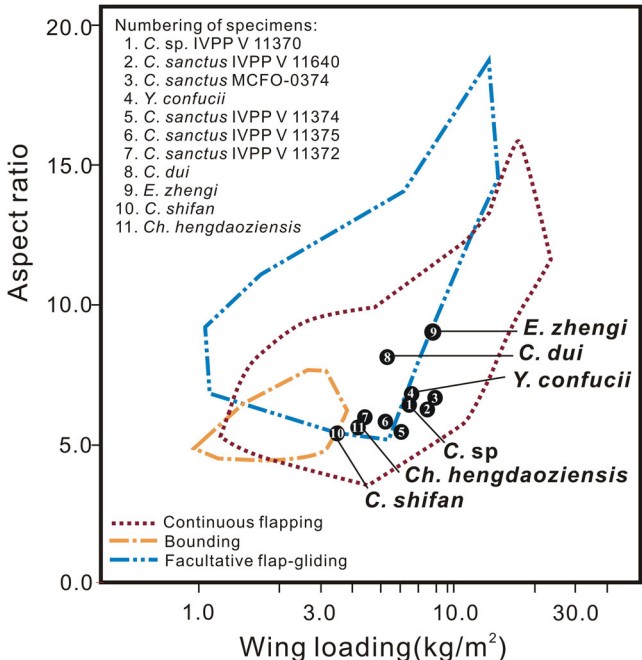

**Fig. 9 Positions of estimated confuciusornithid specimens in a morphospace defined by wing loading (WL) and aspect ratio (AR) of extant birds with particular modes of flight.** Colored lines mark the areas of morphospace occupied by extant birds with particular modes of flight. The image of the morphospace was previously published by Serrano et al.[62], and is used here with permission. Black circles indicate specimens which are holotypes of their respective species, unless only a specimen number is indicated. The specimens are numbered in descending order of estimated body mass.

caudal vertebrae contributing to the pygostyle is known only in *Fukuipteryx*[31], sapeornithids and the early-diverging ornithuromorph *Archaeorhynchus* (IVPP 17075), all of which have 5–6 pygostyle-forming caudals as in most crown-group birds[21,23].

It should be noted that pygostyles are also known in some non-avian theropods. For example, the early-diverging therizinosauroid *Beipiaosaurus* has been reported to have a pygostyle formed by the five distalmost vertebrae[32]; several oviraptorosaurians including *Nomingia* also have their last five caudal vertebrae fused into a pygostyle[33,34]; and the relatively short tails of the scansoriopterygids *Epidexipteryx*[35] and *Ambopteryx*[36] terminate in a pygostyle. In addition, some early birds with a pygostyle, such as *Fukuipteryx*[31] and potentially *Sapeornis*[37], are likely to be earlier-diverging than the long-tailed *Jeholornis*. These observations suggest that pygostyles may have independently evolved multiple times among non-avian theropods and early birds, a scenario consistent with the fact that fusion of the distal caudal vertebrae into a pygostyle is a developmentally easy process[38].

The condition of PMoL-AB00178 has implications for our understanding of not only pygostyle evolution, but also the evolution of the avialan tail as a whole. If the estimated vertebral count for the pygostyle is accurate and PMoL-AB00178 has seven free caudal vertebrae as in *C. sanctus*[3], then a total of 18 caudal vertebrae are present in this confuciusornithid (Figs. 1, 4c). Known early pygostylians, except sapeornithids, normally retain 5–8 free caudal vertebrae: there are seven in confuciusornithids[3], 5–6 in jinguofortisids[30], 6–8 in enantiornithines[20,39], and 5–7 in early ornithuromorphs[25,28,40], similar to the count of 5–6 in crown-group birds[41,42]. Sapeornithids have been reported to have about 12 free caudal vertebrae[43,44], many more than other early pygostylians. Given that 5–6 caudal vertebrae typically form the

pygostyle, the majority of early pygostylians have no more than 14 caudal vertebrae in total. By contrast, confuciusornithids and sapeornithids have about 18 caudal vertebrae, compared to 21–23 free caudal vertebrae in *Archaeopteryx*[45] and 27 in *Jeholornis*[46]. In general, the available data suggest a gradual reduction in caudal vertebral number in early bird evolution and a proportionally much shorter tail in pygostylians than in other early birds, but there are a few exceptions: sapeornithids have a high number of free caudals, and confuciusornithids have a high number of pygostyle-forming ones, as indicated by PMoL-AB00178. In enantiornithines, the proportionally long pygostyle is presumably composed of a relatively high number of vertebrae[23], though no direct evidence is available. Confuciusornithids and most enantiornithines, which have proportionally long rod-like pygostyles, possess ornamental tail feathers[21,47], although long-ipterygids may represent an exception because such feathers have not yet been documented in this enantiornithine subclade[21,48]. In sapeornithids, early ornithuromorphs and pengornithid enantiornithines, by contrast, the short plough-shaped pygostyle supports an aerodynamic rectricial fan[21,48]. The available evidence thus indicates that tail reduction and pygostyle formation involved a complex evolutionary process, and suggests that the pygostyle served different functions in different groups of early birds. The fact that the pygostyle is the only compound bone in PMoL-AB00178 to show incomplete fusion also indicates that the tail may not have played a major role in generating aerodynamic forces during flight.

**Flight adaptation among confuciusornithids**. The flight capabilities of confuciusornithids have been debated in previous studies. Some authors have considered confuciusornithids competent flyers based on a number of morphological features, including long wings with strongly asymmetrical flight feathers, strut-like coracoids, a keeled sternum, and enlarged major manual digits[49], whereas others have suggested more limited flight capabilities[12] based on functional analysis of the morphology of the shoulder joint[7] or the flight feathers[9]. The inference that confuciusornithids were competent flyers has also been supported by some quantitative analyses[50,51]. Nevertheless, there has been broad agreement that confuciusornithids display a more primitive suite of flight-related characters than contemporaneous ornithothoracines[1,7], suggesting inferior flight performance.

Morphological evidence from both *C. shifan* and the recently discovered *Y. confucii* indicates that some variations in flight capability and mode existed within Confuciusornithidae. On one hand, the distantly related *C. shifan* and *Y. confucii* share some derived features related to flight capability, including a sharp angle between the scapula and coracoid, a relatively long coracoid, and a relatively long forelimb (also present in *C. dui* and some *C. sanctus* specimens) (Fig. 6); on the other hand, *Y. confucii* and *C. shifan* each exhibit a flight apparatus specialized to a greater degree and in a different manner than those of other confuciusornithids, or even those of most contemporaneous ornithothoracines. In *Y. confucii*, the main modification is elongation of the forelimb to increase the wing area as in sapeornithids[52] and some ornithothoracine birds, consistent with adaptation for long-distance flight. In *C. shifan*, the main modifications involve reduced body size and refinements to the humerus and carpometacarpus, which to some extent resemble their counterparts in ornithothoracine birds[18,26,53]. *Confuciusornis shifan* is smaller than most other confuciusornithids (Supplementary Table 3), and is notable for the particularly strong development of some skeletal processes to which flight muscles would have attached, including the extensor process of the alular metacarpal and the pisiform process of the

carpometacarpus (Fig. 7a, b). In extant birds, the extensor process is the point of insertion for *M. extensor metacarpi radialis*, the primary muscle involved in extension of the wrist joint, whereas the pisiform process serves for attachment of the retinaculum flexorium and as a pulley changing the direction of the tendon of *M. flexor digitorum profundus*[41], which crosses the wrist and inserts on the distal phalanx of the major digit. A well-developed extensor process and pisiform process can accordingly increase a bird's ability to manipulate and otherwise control the distal part of the wing during flight[54], so the presence of this condition suggests greater flight maneuverability in *C. shifan* than in other confuciusornithids.

A recent study showed that most flight modes seen in extant birds (e.g., continuous flapping, flap-gliding, flap-bounding, thermal soaring) could have been represented among early pygostylians[55]. The flight parameter estimates obtained for confuciusornithids would suggest that they were continuous flappers, but only the holotype of *E. zhengi* and a few specimens of *C. sanctus* were considered. In order to further assess the flight capabilities and strategies of confuciusornithid birds, we used the methods from that analysis[55] and an earlier study that focused on mass estimation[17] (also see Aerodynamic analysis in Methods, below) to estimate BMs and flight parameters for 11 confuciusornithid specimens, including the only known specimens of *Y. confucii* and *C. shifan* (Supplementary Tables 2 and 3). We then plotted our estimated values of two key parameters, namely aspect ratio (AR) and wing loading (WL), on a previously published figure (Fig. 9) showing the overlapping regions of AR-WL morphospace associated with three major flight modes among extant birds[55]. Aspect ratio is the ratio of wing length to wing width, representing a simple measure of wing shape, and WL is the ratio of BM to the total area of the two wings and the part of the trunk between them, representing the load that must be borne by each unit area of the available lift-producing surface. Both metrics are commonly used in assessing the flight capabilities and strategies of flying animals[56].

Our analyses show that BM, AR, and WL vary considerably among confuciusornithid specimens and even among specimens of *C. sanctus* (Supplementary Table 3; Fig. 9), as found in previous analyses[13,55,57]. Interestingly, confuciusornithids are distributed over a large part of the continuous flapping morphospace (Fig. 9), which overlaps extensively on the plot with the morphospaces associated with facultative flap-gliding. *Eoconfuciusornis zhengi*, *Y. confucii*, *C. shifan*, three specimens of *C. sanctus*, and a single specimen identified only as *C.* sp. (IVPP V 11370) plot within the area occupied solely by continuous flappers, and adjacent to the area of overlap with flap-gliders. *Confuciusornis shifan* is also very close to the area of overlap with bounders. *Changchengornis hengdaoziensis* and two specimens of *C. sanctus*, conversely, plot just within the area where flap-gliders overlap with continuous flappers, whereas *C. dui* is close to the center of this zone of overlap. It should be noted that while many of the specimens are positioned near boundaries between flight categories, the parameter estimates that determine their locations all have tight confidence intervals (Supplementary Table 3), suggesting that their referral to particular flight categories is relatively convincing. Among the 11 specimens, *Y. confucii* and *C. dui* possess the proportionally longest forelimb skeleton (Fig. 6) and primary feathers (Supplementary Table 2), respectively, resulting in a high AR, but do not have a particularly high WL (Fig. 9; Supplementary Table 3). *E. zhengi* and *C. shifan* fall at opposite ends of the polygonal area of morphospace occupied by the confuciusornithids, and likely utilized distinct flight strategies that are both seen in extant birds[58–60]. *E. zhengi* is notable for having the highest AR of any confuciusornithid in the analysis, combined with a WL exceeded only by one of the *C. sanctus*

specimens (MCFO-0374), whereas its BM is larger only than those of *Ch. hengdaoziensis* and *C. shifan* (Fig. 9, Supplementary Table 3). Extant birds with high AR and WL are typically medium-sized to large, and are able to generate lift efficiently in fast flight, enabling them to complete long-distance flights with minimal energetic costs[58–60]. By contrast, *C. shifan* has both the lowest AR and the lowest WL of any confuciusornithid in our analysis. Extant birds with low AR and WL are typically small, such as bounding birds, and are able to rapidly take off, and possess high maneuverability in the context of slow short-distance flights[58–60]. Accordingly, our AR and WL estimates suggest that *E. zhengi* and *C. shifan* were well-adapted for fast long-distance flights and for slow short-distance flights, respectively. Our analyses provide further support for the inference that diverse modes of flight adaptation existed among confuciusornithids, and that *C. shifan* was a species characterized by particularly high maneuverability.

More importantly, a comparison under phylogeny may indicate that an evolutionary trend towards decreasing WL and AR values characterized the transition from early-diverging confuciusornithids such as *E. zhengi* and *Y. confucii* to late-diverging ones such as *C. sanctus* and *C. shifan* (Figs. 8 and 9), i.e., that confuciusornithids evolved toward improved flight maneuverability and toward a degree of specialization for slow, short-distance flights like those typically engaged in by extant birds with comparatively low AR and WL values. *Confuciusornis shifan* represents the culmination of this evolutionary trend, at least among currently known confuciusornithids, which is reflected in the proximity of this species to the areas of morphospace associated with flap-gliding and bounding in Fig. 9. Early-diverging confuciusornithids, particularly *Y. confucii*, may instead have been well-adapted for long-distance flight, as suggested by their high AR and the modified shoulder girdle of *Y. confucii* specifically.

Body mass has had an important influence on the evolution of avian flight, and shows a declining trend across the origin of birds and the early evolution of the pygostylia[18,53,61]; BM is also often linked to AR and WL values among extant birds, larger birds tending to have higher AR and WL[58–60]. However, our BM estimates for confuciusornithids under phylogeny do not distinctly show these evolutionary trends (Figs. 8 and 9; Supplementary Table 3). Although *C. shifan* and *Ch. hengdaoziensis*, with relatively low AR and WL, have smaller BMs than other confuciusornithids with higher AR and WL, *E. zhengi* has both the highest AR among confuciusornithids in our study and a very high WL, combined with a smaller BM than any *C. sanctus* specimen (Supplementary Table 3). It is possible that a clearer BM-based trend would be apparent if the analysis excluded immature specimens, but the holotype of *E. zhengi* is thought to represent a subadult[4], and the *C. sanctus* specimens display a large size range (Supplementary Table 3) as in other studies[13,57]. Ontogenetic size variation in the sampled individuals appears to overwhelm any evolutionary trend in body size that may have existed within Confuciusornithidae. Nevertheless, our new finds highlight the diversity of flight adaptation within the earliest-diverging, species-rich pygostylian clade Confuciusornithidae, paralleling the emergence of similar diversity in late-diverging, species-rich pygostylian clades such as ornithothoracines[62]. This is surprising given that confuciusornithids lack a modern pectoral girdle[24] and are inferred to have had a weak downstroke[12]. It is thus important to thoroughly investigate the functional morphology of the flight stroke in confuciusornithids and other early pygostylian birds.

**Evolution of secondary ossification centers**. The discovery of the epiphyseal SOC in *C. shifan* was a truly unexpected finding of this

study. The long bones of tetrapods each fundamentally comprise a central shaft (diaphysis) with expanded articular ends (epiphyses), which are separated from the diaphysis by short intermediate zones known as metaphyses[63]. In extant tetrapods, the long bones are initially cartilaginous and endochondral ossification begins within the diaphysis, at what is termed the primary ossification center (POC). In most tetrapod groups, ossification spreads gradually from the POC throughout the diaphysis and metaphyses, and potentially into the epiphyses, leaving only a relatively thin cap of hyaline cartilage over the endochondral bone. In mammals and most lizards, however, SOCs appear within the epiphyses, and are separated from the metaphyses by growth plates[63–65]. Each growth plate constantly generates new hyaline cartilage between itself and the metaphysis, so that the skeletal element grows longitudinally. Meanwhile, cartilage is gradually converted to bone within the epiphyses, as well as in the proximal and distal ends of the diaphysis. As skeletal maturity approaches, the growth plate becomes thin and finally disappears, causing complete fusion of the epiphyses to the metaphyses and, in turn, an end to longitudinal growth[63]. The timing of complete fusion varies among species, and among the different long bones of a single individual[63,66]. Additional ossification centers may appear within apophyses ("traction epiphyses"), and eventually fuse with the long bones to serve as protruding attachment sites for tendons and ligaments[67,68]. Furthermore, ossification centers may appear within the tendons or ligaments themselves, forming bones called sesamoids[69,70]. These may either remain embedded in the tendons/ligaments with no direct connection to the rest of the skeleton, as in the case of the mammalian patella, or fuse into adjacent long bones as apophyseal projections[67–70].

Presently, the only recognized true epiphyseal ossification in birds is one that forms at the proximal end of the tibiotarsus in many taxa, and may be plesiomorphic at least for crown-group birds[68,71–75]. Furthermore, the tarsal elements and some carpal elements of extant birds fuse to adjacent long bones in a manner superficially resembling the fusion of epiphyseal ossifications to metaphyses, and apophyseal ossifications are also known in some species[72,74].

Epiphyseal SOCs have been demonstrated to provide additional stiffness to articular portions of bones that require reinforcement, for example as a result of stresses associated with support and locomotion in the terrestrial environment, and to facilitate the formation of topographically complex joint surfaces[65,76,77]. To our knowledge, however, epiphyseal SOCs do not occur in non-avian archosauriforms, including large, undoubtedly terrestrial dinosaurs that would have experienced substantial locomotor stresses. Epiphyseal SOCs probably evolved independently in mammals, birds and lepidosaurs[69], for reasons potentially relating to patterns of skeletal growth as well as to mechanical factors. Furthermore, we are unaware of any previous report of an epiphyseal SOC in a non-ornithuromorph bird. The proximal tibial center appears to have been present in hesperornithiforms based on differences between adult and juvenile tibiotarsi of the Early Cretaceous genus *Enaliornis*[78], but it is surprising that the holotype of *C. shifan* possesses an epiphyseal SOC anywhere in the skeleton, let alone at a location—the cranial portion of the distal end of the alular metacarpal—where no such feature occurs even in crown-group birds.

The SOC in the distal epiphysis of the alular metacarpal of *C. shifan* (Fig. 7a, b) is presumably an autapomorphic feature, whose presence may be explicable on the basis of a combination of growth strategy and functional demands on the wing. A recent study has demonstrated that epiphyseal SOCs tend to evolve as a response to high mechanical stress[77]. For example, epiphyseal SOCs are developed only in the legs and thumbs of newborn bats, which are used from birth to cling to the mother or the roost[77].

In extant birds the alular metacarpal plays an important role in aerial maneuvers[58,59], and the wings of *C. shifan* appear better-adapted for high-maneuverability flight than those of other confuciusornithids. As a result, the distal end of the alular metacarpal of *C. shifan* may well have begun to experience considerable stresses as soon as a developing juvenile began to fly. Furthermore, the histological features of *C. shifan* suggest a slower growth rate than that of other confuciusornithids, based on Amprino's rule[79] (Fig. 2). For example, *C. shifan* has longitudinally oriented vascular canals in the middle layer of the femoral compact bone (Fig. 2), whereas concentrically oriented canals are more numerous in other confuciusornithids[13]. If the spread of ossification from the primary centers of the long bones into the epiphyses was also slow, then the establishment of an epiphyseal SOC might have been necessary to reinforce the distal end of the alular metacarpal at a comparatively early ontogenetic stage, when the juvenile was beginning to fly but skeletal growth was still incomplete. The holotype of *C. shifan*, however, clearly represents a mature individual in which the caudal distal condyle of the alular metacarpal had fully ossified by the time of death, presumably via the normal mechanism of spread from the diaphyseal POC.

## Methods

**Histological preparation of *C. shifan* (PMoL-AB00178).** We took a bone sample from near the mid-diaphysis of the right femur of PMoL-AB00178 (Fig. 1a), and prepared a thin cross-section using standard procedures[80]. The sample was embedded in one-component resin (EXAKT Technovit 7200), which was then hardened in a light polymerization device (EXAKT 520). A thin cross-section was cut using a high-precision circular saw (EXAKT 300CP). The section was ground down using the EXAKT 400CS grinding system until the desired optical transparency was obtained. The histological section was examined under a polarized light microscope (ZEISS Axio Imager 2 Pol), and photographed with a ZEISS AxioCam 705 digital microscope camera.

**Phylogenetic analysis.** To investigate the systematic position of *C. shifan* relative to other Mesozoic birds, a phylogenetic analysis was carried out using the most comprehensive data matrix currently available for Mesozoic birds[81], with *C. shifan* added in based on scorings from PMoL-AB00178 (Supplementary character scorings for cladistic analysis). The revised matrix consists of 81 taxa and 280 characters. The matrix was analyzed using TNT v1.5[82] with default settings. All characters were equally weighted, with 35 characters ordered; the dataset was analyzed using the "New Technology search" methods of sectorial search, ratchet, tree drift, and tree fusion, with default settings; and the minimum-length tree was found in 10 replicates to recover as many tree islands as possible. The recovered trees were then used as the basis for a traditional TBR search. Zero-length branches were collapsed. Decay indices (Bremer support values) were calculated using the Bremer script embedded in TNT, and absolute Bootstrap frequencies were calculated using 1000 pseudoreplicates in TNT with default settings.

**Aerodynamic analysis.** In order to assess the flight capabilities and strategies of confuciusornithid birds, we estimated values of several flight parameters for 11 confuciusornithid individuals, including the holotypes of *C. dui*, *Ch. hengdaoziensis*, *E. zhengi*, *Y. confucii* and *C. shifan* (see Supplementary Tables 2 and 3). Specifically, we estimated the body mass (BM), wingspan (B), and lift surface area ($S_L$) of these specimens following the methods of Serrano et al.[17,55], and calculated a confidence interval for each estimate based on the mean percentage prediction error (|%MPE|) obtained by Serrano et al.[17,55] for the multivariate equation with which the estimate was generated. We then used these estimates as a basis for calculating aspect ratio (AR, $AR = B^2/S_L$) and wing loading (WL, $WL = BM/S_L$) values (see Supplementary Table 3) for the confuciusornithids. Almost all the anatomical data needed for these estimates were obtained by measuring the specimens directly with digital calipers (±0.1 mm), by taking measurements from high-resolution images using tpsDig 2.17[83] (available at: https://sbmorphometrics.org/), or from measurements published in the literature; but in a few cases, primary feather length had to be estimated from the combined length of the main segments of the wing skeleton (see Supplementary Table 2 and accompanying text for details).

**Nomenclatural acts.** This published work and the nomenclatural acts it contains have been registered in ZooBank, the proposed online registration system for the International Code of Zoological Nomenclature (ICZN). The ZooBank LSID (Life Science Identifiers) can be resolved and the associated information viewed through any standard web browser by appending the LSID to the prefix http://zoobank.org/. The LSID for this publication is 945DD1B0-C3A8-49BC-993E-40A4F857083F.

**Reporting summary**. Further information on research design is available in the Nature Portfolio Reporting Summary linked to this article.

## Data availability

All data are publicly available in this article and its supplementary file, and other data are available from the corresponding author on reasonable request. The specimen PMoL-AB00178 is housed at the Paleontological Museum of Liaoning (Shenyang).

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

## Acknowledgements

We thank Ge Sun for help during the course of the work, Shurui Yang and Jingqi Wang for preparing the specimen described in this paper, Fei Liang, Shuchong Bai, and Wenju Xie for photography, Shukang Zhang and Jinkai Jiang for assistance with preparing the histological section, and Francisco José Serrano for valuable comments on the aerodynamic analysis. This work was supported by grants from NSFC (41688103, 42072030, 42288201) and LZD201701, a Discovery Grant from the Natural Sciences and Engineering Research Council of Canada (RGPIN-2017-06246), and start-up funding awarded by the University of Alberta to C.S.

## Author contributions

D.-Y.H., R.-F.W., M.-S.Z., and X.X. designed the research plan. R.-F.W., D.-Y.H., C.S., S.-Y.W., and Q.Z. performed the analysis. R.-F.W., D.-Y.H., C.S., X.X. wrote the manuscript.

## Competing interests

The authors declare no competing interests.
