## [Peer Review File · Communications Biology]

Reviewers' comments:

Reviewer #1 (Remarks to the Author):

The non-ornithothoracine confuciusornithids are truly an interesting clade of fossil birds. They are the most ubiquitous in the number of specimens and one of the most speciose among Cretaceous avialan families. Therefore, the clade gives us a rare window into the past about the speciation, morphological evolution, and ecology of basal birds during the time when numerous evolutionary experiments on flight adaptation were taking place among avialans.

In this regard, I feel it very interesting that the clade still contains previously unknown species, and authors successfully present lines of evidence to support that it is indeed a new species of the confuciusornithids. Furthermore, the manuscript provides new information about flight adaptation in the clade in terms of development of pygostyle and strategy of limb ossification.

Therefore, I think the paper deserves publication and specialists in the field would certainly benefit from the new morphological and evolutionary information provided from the manuscript. Confuciusornithids are one of well known fossil birds for general public (perhaps second to famed Archaeopteryx), and an addition of a new species with multiple evolutionary implications would be certainly welcomed.

With that being said, I recommend that the manuscript be accepted with minor revisions. I am mostly convinced and satisfied with fossil evidence that the authors provide, but I generally find that the discussion could be expanded considering the implications that the specimen might have. The authors conduct a phylogenetic analysis and present evolutionary relationships of confuciusornithid species. They also investigate confuciusornithid flight adaptation in terms of pygostyle development and wing forms. Therefore, the paper would be more interesting if it includes a discussion about the potential course of evolution for flight among confuciusornithids. Also, one of the main argument of the paper is the finding of epiphyseal SOC. While I am not concerned about the discussion of the feature, I feel it deserves additional evidence to support that the bone element indeed comes from epiphyseal SOC. It is such an unusual finding and the identification needs to be done carefully.

As a supplementary file, I included annotated manuscript PDF. Please refer to it for my specific comments and suggestions.

Finally, I am happy that our knowledge about these Early Cretaceous birds keep being added, as manuscripts like this keep coming out. The present one surely contributes to the study of avian evolution and I look forward to the paper published in the near future.

Reviewer #2 (Remarks to the Author):

Wang et al. have made an excellent description of a new confuciusornithid taxon, as well as making novel comments on the evolution of flight in Confuciusornithidae both with qualitative description of a small bone unique among avialans and some quantitative measurements shown in extant birds to be informative of flight style. The descriptive aspects of the work are extensive and go above and beyond, e.g. using histology to establish maturity of the specimen with a higher confidence than most fossil bird descriptions. There are, however, three key areas I believe need revision:

1. Assigning the specimen to the genus *Confuciusornis* is not well-justified. To my knowledge the genus currently does not have a diagnosis, and several works (including this one) have found the genus to be paraphyletic. Thus I recommend simply erecting a new genus for this specimen.
2. Differentiation from *Confuciusornis dui* has not been completely established. The specimen shares some key features described in this species, and from published material I have not been able to confirm *C. dui* lacks their proposed autapomorphies for *C. shifan*. I believe the authors are correct that they are distinct, but I do not think the manuscript has shown that conclusively.

3. Uncertainties in the quantitative section should be addressed more. Body mass and primary feather length are both estimated, and I know the former equations have a $\sim\pm 20\%$ uncertainty range. I would like to see discussion of how these uncertainties affect the results.

I have also commented directly on the PDF some more minor revisions, mostly key citations that should be added or ways phrasing and figures could be made clearer.

I believe the conclusions of this work will be of interest to a broad range of readers including paleontologists, evolutionary biologists, and developmental biologists. However, I will note that the description currently contains an abundance of field-specific anatomical terms that will likely not be accessible to a general audience.

Overall, I recommend this manuscript for publication given the minor revisions above.

Reviewer #3 (Remarks to the Author):

This paper describes a new confuciusornithid bird from China and discusses its flight ability. Overall I think it's a worthwhile and interesting contribution and worth publishing with minor revisions.

My main disagreement is with how the results are framed. With respect to flight adaptation, this is complex- what is a "higher level of flight adaptation"?

A duck has much shorter, smaller wings than an albatross. But the duck is highly adapted for fast flapping flight, whereas the albatross is highly adapted as a dynamic soarer. They are highly adapted for different kinds of flight. A pheasant has short, small wings, but it has exceptional ability for short bursts of flight and extremely high power output, it is far better at a quick takeoff than an albatross, which must run to get airborne. They are all highly specialized just in different ways.

Or as the Daoist philosopher Zhuang Zhou put it back in the Warring States period: which is more useful: the long legs of a crane, or the short legs of a duck? One is adapted for wading, one for swimming; both serve their purpose for their respective owners. It's all an issue of tradeoffs. They're better for some things and worse for others.

So I think it's a bit subjective; I think what you can definitely argue is that there is a high diversity of wing design here: the Confuciusornithidae are clearly doing a range of different things, just like modern birds that differ in wing proportions, flight style, foraging, etc. I think it would make more sense to emphasize the diversity of flight adaptation in the clade, rather than necessarily trying to argue that they're evolving to become overall better fliers, which isn't impossible, but seems harder to demonstrate.

In retrospect it's easy to identify various taxa as transitional and closer or further along the path to extant birds, to see evolution in directional terms. But of course at the time, Confuciusornithidae were one of many different competing lineages- enantiornithes, ornithurae, etc.- and just like them, they were diversifying in different directions based on local selective pressures, and undergoing radiations; it's interesting to see this; I think it would be useful to emphasize this part of the story more.

The sesamoid was less interesting to me, I feel that section could be condensed.

A new confuciusornithid bird with a secondary epiphyseal ossification reveals
phylogenetic changes in confuciusornithid flight mode

Renfei Wang^{1,2}, Dongyu Hu^{2*}, Meisheng Zhang¹, Shiyong Wang^{3,4}, Qi Zhao^{3,4}, Corwin
Sullivan^{5,6} & Xing Xu^{3,4*}

¹ College of Earth Sciences, Jilin University, Changchun 130061, China

² Shenyang Normal University, Paleontological Museum of Liaoning, Key Laboratory
for Evolution of Past Life in Northeast Asia, Liaoning Province, 253 North Huanghe
Street, Shenyang 110034, China

³ Key Laboratory of Vertebrate Evolution and Human Origins, Institute of Vertebrate
Paleontology and Paleoanthropology, Chinese Academy of Sciences, Beijing 100044,
China

⁴ Center for Excellence in Life and Paleoenvironment, Chinese Academy of Sciences,
Beijing 100044, China

⁵ Department of Biological Sciences, University of Alberta, Edmonton, AB T6G 2E9,
Canada

⁶ Philip J. Currie Dinosaur Museum, Wembley, AB T0H 3S0, Canada

Correspondence and requests for materials should be addressed to D.H. (email:
hudongyu@synu.edu.cn) and X.X. (email: xuxing@ivpp.ac.cn)

The confuciusornithids are the earliest known beaked birds, and ~~also~~ constitute
the only species-rich and **morphologically diverse** clade of Early Cretaceous birds that
existed prior to the cladogenesis of two major avialan groups Enantiornithes and
Ornithuromorpha. Here, we report a new confuciusornithid species from the Lower
Cretaceous of western Liaoning, northeastern China. Compared to other
confuciusornithids and most contemporaneous enantiornithines and ornithuromorphs,
this new species and the recently reported *Yangavis confucii* show a higher level of
flight adaptation, ~~but~~ the wings of the two taxa differ from one another in many
respects. Aerodynamic analyses of the known confuciusornithids indicate
considerable variations of flight capability and style across the clade, and to a lesser
degree through ontogeny, and specifically suggest both a trend towards improved
flight capability and a change in flight strategy in confuciusornithid evolution. Most
significantly, this new confuciusornithid differs from other Mesozoic birds in having a
secondary epiphyseal ossification, located in the alular digit. This highly unusual
feature may reflect the functional demands of flight when skeletal growth was still
incomplete. The new find strikingly exemplifies the morphological, developmental
and functional diversity of the first beaked birds.

**Introduction**

Confuciusornithidae is a clade of Early Cretaceous pygostylian birds known from
the Jehol Biota of East Asia¹, representing the earliest known toothless, beaked birds.
Five genera and eleven species, recovered from the Dabeigou, Yixian and Jiufotang
formations (~135~120 Ma), have been described and assigned to this family, though
the validity of some species is questionable²⁻⁶. Confuciusornithids are the only
species-rich, **morphologically diverse** avialan clade known to have existed prior to the
cladogenesis of the major groups Enantiornithes and Ornithuromorpha, and are
represented by thousands of exceptionally well preserved specimens that collectively
provide rich information on confuciusornithid morphology, taxonomy, flight ability,
growth, diet and ecology^{3,5,7-13}. Here, we report a new confuciusornithid species,
*Confuciusornis shifan* sp. nov., from the Jiufotang Formation. *C.* *shifan* differs from

other confuciusornithids in a number of morphological and developmental features,
which have implications for understanding confuciusornithid taxic diversity,
morphological disparity, development, and flight behavior.

**Results**

**Systematic paleontology.**

Aves Linnaeus, 1758

Pygostylia Chiappe, 2002

Confuciusornithidae Hou et al., 1995

*Confuciusornis* Hou et al., 1995

*Confuciusornis shifan* sp. nov.

[revised manuscript text omitted]

The left and right dentaries are almost completely fused with each other along an
extended, ventrocaudally inclined mandibular symphysis as in *C. sanctus*³ (Fig. 3a,b).
The transversely compressed ventrocaudal process of the right dentary is visible, but
the caudalmost end of the process is missing. In the intact mandible, the ventrocaudal
process likely contacts the angular laterally, and extends back to the level of the
caudal margin of the anterior mandibular fenestra based on the position of the
impression of the missing end. The right angular and right surangular both appear
rod-shaped, and their caudal ends fuse into the articular. The surangular appears to
lack a ventral process as in *C. dui*, indicating that the presence of a triangular ventral

surangular process in *C. sanctus* is indeed a diagnostic feature of this species⁵ (Fig.
3b). The articular prominently projects medially to form a medial articular facet for
the quadrate¹⁶. In contrast to the displaced and partly overlapped left splenial, the right
splenial is nearly complete and preserved close to its original position. The splenial is
plate-like and rostrally forked, and protrudes into the rostral half of the rostral
mandibular fenestra. The dorsal margin of the splenial overlaps the medial surface of
the surangular, and the ventral margin, which does not contact the angular, is
thickened along most of its length and gently tapers caudally to contact the ventral
process of the surangular. Unlike in other confuciusornithids, the splenial of *C. shifan*
is centrally perforated by an oval foramen (Fig. 3a,b).

The cervical vertebrae are exposed in ventrolateral view, ~~making it possible to see~~
that the cranial and caudal articular surfaces of each centrum are distinctly
heterocoelous (Fig. 4a) as previously reported for *Confuciusornis*⁵. The centra of the
thoracic series are laterally excavated as in *C. sanctus*, whereas such excavations are
absent in *Y. confucii*⁶ (Fig. 1). ~~The two cranialmost centra each bear~~ a prominent,
tapered ventral keel. The sacral vertebrae are completely fused with each other to
form a synsacrum (Fig. 4b). Based on the number of visible transverse processes, the
synsacrum consists of seven vertebrae as in other confuciusornithids³⁻⁶, and in ventral
view the total centra are transversely compressed to form a longitudinal ridge as in
some enantiornithines¹⁷ (Fig. 4b). However, the synsacrum of *C. sanctus* DNHM
D2454 (holotype of the putative species *Confuciusornis feducciai*) appears
dorsoventrally compressed and transversely widened⁵. Five free caudal vertebrae are
visible between the synsacrum and the pygostyle (Fig. 1), whereas additional caudal
vertebrae are likely overlaid by the left ischium and some soft tissue. The caudal
vertebrae appear significantly shorter than the thoracic ones. The pygostyle of *C.*
*shifan* appears to be exposed in left lateral view (Fig. 4c). The pygostyle is
proportionately robust and subequal in length to the tarsometatarsus, as is generally
the case in confuciusornithids^{3,18}. The transversely thickened ventral margin of the
pygostyle is distinctly made up of the co-ossified centra, which are thinned caudally.
The ventral surface does not form a longitudinal keel as in *C. sanctus* IVPP V 12352

(holotype of the putative species *Jinzhouornis zhangjiyingia*)^{5,18}. Some 11 small round
foramina form a longitudinal row along the whole lateral furrow of the pygostyle (Fig.
4c). The foramina are closer to the ventral margin of the pygostyle than the dorsal
margin, and gradually become smaller and more tightly crowded together towards the
caudal end of the series. Similar foramina occur in other confuciusornithine
specimens, but are fewer in number and only present on the cranial part of the
pygostyle^{18,19}.

The coracoids are fused with the scapulae to form scapulocoracoids. The left and
right scapulocoracoids are exposed in lateral and medial views, respectively (Fig. 5a).
The long axes of the scapula and coracoid define an acute angle as in *Y. confucii*⁶ and
more advanced birds (Fig. 5a, 6). This angle is approximately 90° in *C. sanctus*³. A
laterally directed glenoid facet is visible in the left scapulocoracoid. The cranial and
caudal margins of the glenoid facet project more strongly than the dorsal and ventral
ones, implying that the configuration of the facet probably did not strongly limit the
elevation and depression of the wing in flapping flight. The acromion of the scapula is
well-developed. The shaft of the scapula curves slightly downward; the cranial half of
the shaft appears rod-like and slightly compressed dorsoventrally, and while the
caudal half is more compressed mediolaterally and widens significantly in the
dorsoventral direction before tapering to a point as in more advanced birds²⁰ (Fig. 5a,
6). In other confuciusornithids, the scapula is generally straight, and neither flares
partway along its length nor tapers caudally³. The coracoid is strut-like and
approximately half the length of the scapula. Its proximal end appears dorsoventrally
thicker than transversely wide and projects slightly cranially to form an acrocoracoid
process as in *Y. confucii*⁶, whereas its distal end is compressed dorsoventrally. The
acrocoracoid process is not well-developed in *C. sanctus*³. The lateral surface of the
coracoid is excavated to form a shallow groove, which gradually becomes narrower
and shallower distally as in *Y. confucii*⁶. The medial margin of the coracoid is strongly
compressed to form a longitudinal keel, except at the proximal end of the bone. The
furcula is mostly exposed in caudal view. As in other confuciusornithids, the furcula is
boomerang-shaped and without a distinct tubercle³⁻⁶ (Fig. 5a, 6). The proximomedial

corner of the right clavicular ramus is inflated caudally to form a bulbous structure
that contacts the medial face of the proximal end of the right scapulocoracoid, and
meanwhile the proximolateral corner is more craniocaudally compressed than the rest
of the ramus and slightly projects laterally, to form a distinct impression on the caudal
surface of the corner. Similar structure is not seen in other confuciusornithids.

The forelimb is subequal in length to the hindlimb, with a forelimb (humerus +
ulna) / hindlimb (femur + tibiotarsus) ratio of 1.02 (Fig. 1; Supplementary table S1).
The equivalent ratio is 0.93 in *C. hengdaoziensis*, 0.96 in *E. zhengi*, 1.06 in *C. dui*,
0.91~1.08 in *C. sanctus* and 1.19 in *Y. confucii*⁶ (Fig. 6). The humerus is typical of
confuciusornithids in having an expanded and perforated deltopectoral crest. However,
the crest is less prominent overall than those of other confuciusornithids except *C. dui*⁵,
and the point of the greatest prominence is at the distal end of the crest rather than in the
middle as in other confuciusornithids³⁻⁶ (Fig. 5a, 6). The dorsodistal corner of the crest
projects distally, so that the distal margin appears concave. Distally, a large fossa for the
brachialis muscle is present proximal to the dorsal and ventral condyles. The dorsal
epicondyle is better developed than the ventral epicondyle, and the former projects
sharply dorsally. Both the ulna and the radius are shorter than the humerus, as in other
confuciusornithids (Fig. 1, 6; Supplementary table S1, 2). A concave ventral cotyle, a
flat dorsal cotyla and a concave incisura radialis are clearly visible on the proximal
end of the right ulna, whereas a distinctly projecting olecranon process is lacking. The
proximal humeral articular facet of the radius is flat and the bicipital tubercle is
situated on the cranial face of the proximal end of the radius. Both radialis are in their
original positions relative to the radii and carpometacarpi (Fig. 1). The left radiale are
exposed in palmar and obliquely proximal view and the right in palmar and obliquely
distal view. The articular facets for the radius and semilunate carpal are both
significantly concave. The cranial surface of the radiale is convex and forms a small
tubercle on the proximal side, and the caudal surface is slightly concave and directed
somewhat proximally to articulate with the ulna.

The semilunate carpal is completely fused with metacarpals II and III to form a
carpometacarpus (Fig. 1, 7a,b). Metacarpal III is much more slender, and slightly

shorter, than metacarpal II, and contact between these metacarpals is limited to their
proximal ends as in enantiornithines²¹. The middle parts of metacarpals II and III are
constricted, and the narrowest part of metacarpal III is less than half the width of the
narrowest part of metacarpal II as in other confuciusornithids. However, the
intermetacarpal space appears relatively longer and significantly wider than in other
confuciusornithids, as in some advanced birds²⁰. The proximal end of the
carpometacarpus forms a pulley-like carpal trochlea as in *C. sanctus*³ and extant birds.
A distinct fossa is present in the region where the semilunate carpal is fused to
metacarpals II and III. The bump-like pisiform process is located near the proximal
end of metacarpal II. The alular metacarpal is about one third the length of metacarpal
II as in *C. sanctus*³ and *E. zhengi*⁴, rather than half the length as in *Ch.*
*hengdaoziensis*³ (Fig. 6, 7a,b). It is not fused with the carpometacarpus (Fig. 7a,b).
The caudal portion of the alular metacarpal extends farther proximally than the cranial
portion so that the proximal surface of the metacarpal is concave, forming a cranial
carpal fovea. The proximal two thirds of the cranial margin projects cranially to form
the extensor process. The craniocaudal width of the process is nearly half that of the
distal articular surface of the alular metacarpal (Fig. 7a,b,c). Such a well-developed
extensor process is not seen in contemporaneous basal birds, or in most
enantiornithines and ornithuromorphs^{20,21}. Unlike in previously known
confuciusornithids, the caudal distal condyle of the alular metacarpal is
well-developed, and projected more distoventrally than the rest of the metacarpal,
evident in palmar view. The proximal phalanx of digit II is robust, maintains a
constant diameter throughout its entire length, and is slightly shorter than the
intermediate phalanx as in other confuciusornithids except *Ch. hengdaoziensis*³(Fig.
1). In *Ch. hengdaoziensis*, the proximal phalanx is slightly longer than the
intermediate one³. The ungual phalanges of the alular digit and digit III are large and
curved, whereas that of digit II is small as in most confuciusornithids. In *Y. confucii*,
however, the ungual phalanx of digit II is large⁶.

Among the fully articulated left and right pelvic girdles, only the pubes are
relatively completely preserved (Fig. 5b). The right ilium appears relatively low, with

a slightly convex dorsal margin and a laterally projected antitrochanter. The pubis,
remarkably, is longer than the femur, being rod-like, slender, and strongly retroverted.
Because of a sharp bend in the pubic shaft, the long axis of the pubic symphysis is
nearly perpendicular to that of the shaft's proximal portion, a condition resembling
that seen in derived ornithuromorph birds.

The femoral head, neck and trochanteric crest are well-developed. The proximal
tarsals are completely fused with each other and with the tibia, forming a tibiotarsus.
The proximal ends of metatarsals II-IV are fused with each other and with the distal
tarsals, forming a tarsometatarsus. The proximal end of metatarsal III appears
transversely compressed between metatarsals II and IV in cranial view (Fig. 7d),
whereas this metatarsal in other confuciusornithids does not change in width along its
length³⁻⁶. An oval tubercle presumably for insertion of *M. tibialis cranialis* is located
on the dorsal surface near the proximal end as in *C. sanctus*³, but similar structure
does not exist on metatarsal II. A ridge-like process is present on the lateral margin of
the distal third of metatarsal IV (Fig. 7d), but it is not seen in other confuciusornithids.
In pedal digits III-IV, the penultimate phalanx is longer than the preceding phalanx, as
in other confuciusornithids⁶. However, the penultimate phalanx of pedal digit II is
subequal in length to the preceding phalanx.

*Secondary epiphyseal ossification* The most unusual skeletal feature exhibited by
PMoL-AB00178 is the presence of a cushion-like small bone between the distal end
of the alular metacarpal and the proximal end of the first alular phalanx (Fig. 7a,b). In
the left manus, the small bone contacts the cranial portions of the opposing articular
surfaces of the metacarpal and phalanx (Fig. 7a); in the right manus, the small bone
again contacts the cranial portion of the proximal articular surface of the first alular
phalanx, but rests against the cranial margin of the left alular metacarpal due to the
slight disarticulation of the alular digit (Fig. 7b). We identify this small bone as an
ossification that arose from a secondary ossification center within the distal epiphysis
of the alular metacarpal and is equivalent to the medial condyle seen on the
metacarpal's distal articular surface in other confuciusornithids (Fig. 7c), based on the

following evidence: (1) in the right manus, the small bone remains in apparently
natural alignment with the alular metacarpal, but not with the first alular phalanx; (2)
in other confuciusornithids the distal articular surface of the alular metacarpal is
ginglymoid⁶, and the caudal condyle protrudes slightly more distally than the cranial
one (Fig. 7c), whereas in both alular metacarpals of PMoL-AB00178 the caudal
condyle is distally prominent and cranial condyle is absent; and (3) in both first alular
phalanges of PMoL-AB00178, the proximal articular surface is similar in morphology
to those of other confuciusornithids, and a well-developed flexor tubercle is visible on
the left alular phalanx.

Discussion

Our phylogenetic analysis places *C. shifan* as the sister taxon to *C. sanctus*, and *E.*
*zhengi*, *C. dui*, *Y. confucii* and *Ch. hengdaoziensis* as successive outgroups to the *C.*
*sanctus* + *C. shifan* clade. However, the various confuciusornithid subclades involved
in this arrangement are not well supported, as indicated by their low Bremer and
Bootstrap values (Fig. 8).

The discovery of *C. shifan* sheds new light on morphological evolution and flight
adaptation in confuciusornithid birds. Among early pygostylians, both
confuciusornithid and enantiornithines have relatively large pygostyles¹⁸, although the
exact number of caudal vertebrae involved in the pygostyle is typically difficult to
determine due to their complete coossification. Perforated pygostyles, such as that of
PMoL-AB00178, are thought to be the result of incomplete fusion between adjacent
neural arches, and the foramina are finally enclosed^{18,19}. The pygostyle of
PMoL-AB00178 then presumably exemplifies the developmental stage at which
fusion among the neural arches was partial rather than complete. Given the presence
of 11 preserved foramina, the number of caudal vertebrae within the pygostyle is
estimated to be 12. If the estimate for the pygostyle is accurate and PMoL-AB00178
has seven free caudal vertebrae as in *C. sanctus*³, then 19 caudal vertebrae are present
in the tail as a whole (Fig. 1, 4c), a total only slightly less than the count of 21-23 free
caudal vertebrae reported for the long bony tail of *Archaeopteryx*²². This comparison

implies that the transition in birds from a long bony tail like that plesiomorphically
present in reptiles to a short one ending in a pygostyle was driven primarily by
shortening of the individual caudal vertebrae, rather than by a reduction in vertebral
count. The fact that the pygostyle is the only compound bone in PMoL-AB00178 to
show incomplete fusion also indicates that the tail may not have played a major role in
generating aerodynamic forces during flight.

The flight capability of confuciusornithids has been debated in previous studies.
Some authors have considered confuciusornithids competent flyers based on a number
of morphological features, including long wings with strongly asymmetrical flight
feathers, strut-like coracoids, a keeled sternum, and enlarged major manual digits²³,
whereas others have suggested more limited flight capabilities¹² based on functional
analysis of the morphology of the shoulder joint⁷ or the flight feathers⁹. Nevertheless,
there has been broad agreement that confuciusornithids display a more primitive suite
of flight-related characters than contemporaneous enantiornithines and
ornithuromorphs^{1,7}, suggesting inferior flight performance. However, morphological
evidence from both *C. shifan* and the recently discovered *Y. confucii* indicates that
some variation in flight capability existed within Confuciusornithidae, as both these
species appear to have been unusually well-adapted for flight, albeit in different ways.
*Y. confucii* and *C. shifan* both exhibit a more specialized flight apparatus than is
present in other confuciusornithids, or even in most contemporaneous enantiornithines
and ornithuromorphs. In *Y. confucii*, the main modification is elongation of the
forelimb to increase the wing area as in sapeornithids²⁴ and some enantiornithine and
ornithuromorph birds. In *C. shifan* the main modifications involve refinements to the
pectoral girdle, humerus and carpometacarpus, which to some extent resemble their
counterparts in enantiornithine and ornithuromorph birds. In particular, some skeletal
processes to which flight muscles would have attached are particularly well developed.
A few such features, including the extensor process of the alular metacarpal and the
pisiform process of the carpometacarpus, are seen only in the contemporaneous
enantiornithine *Xiangornis*²¹, a few Early and Late Cretaceous ornithuromorphs²⁵, and
extant flying birds (Fig. 7a,b). In extant birds, the extensor process is the point of

insertion for *M. extensor metacarpi radialis*, the primary muscle involved in extension
of the wrist joint, and the pisiform process serves for attachment of the retinaculum
flexorium, and as a pulley changing the direction of the tendon of *M. flexor digitorum*
*profundus*²⁶, which crosses the wrist and inserts on the distal phalanx of the major
digit. A well-developed extensor process and pisiform process can accordingly
increase manipulating and controlling capabilities of the distal part of the wing during
flight²⁷, collectively suggesting greater flight capability in *C. shifan* than in other
confuciusornithids.

A recent study showed that most flight modes seen in extant birds (e.g.,
continuous flapping, flap-gliding, flap-bounding, thermal soaring) could ~~potentially~~
have been represented among early birds²⁸. The flight parameter estimates obtained
for confuciusornithids in that analysis would suggest that they were continuous
flappers, but only the holotype of *E. zhengi* and a few specimens of *C. sanctus* were
considered. In order to further assess the flight capabilities and strategies of
confuciusornithid birds, we followed the methods of Serrano et al. (2017)²⁸ (also see
Aerodynamic analysis in Methods below) to estimate flight parameters for 11
confuciusornithid specimens, including the only known specimens of *Y. confucii* and
*C. shifan* (Supplementary table S2, 3), and plotted estimates of two key parameters for
these confuciusornithids, namely aspect ratio (AR) and wing loading (WL), on a
previously published figure showing the regions of AR-WL morphospace associated
with the main flight modes in extant birds²⁹. ~~AR~~ is the ratio of wing length to wing
width, representing a simple measure of wing shape, and WL is the ratio of body mass
to the total area of the two wings and the part of the trunk between them, representing
the load that must be borne by each unit area of the available lift-producing surface.
Both metrics are commonly used in assessing the flight capabilities and strategies of
flying animals. Our analyses show that both AR and WL vary considerably among
confuciusornithid specimens and even among specimens of *C. sanctus* (Supplemental
table S3; Fig. 9), as found ~~by previous researchers~~²⁸. Though two specimens of *C.*
*sanctus* fall slightly into the area of morphospace where extant facultative flap-gliders
overlap with continuous flappers, most confuciusornithids plot at least narrowly

outside the flap-gliding area, and within the area occupied solely by continuous
flappers (Fig. 9). Interestingly, confuciusornithids are distributed throughout a large
part of the continuous flapping morphospace, and two ends represented by *E. zhengi*
and *Y. confucii* with the highest WL and AR, and *C. shifan* with the lowest WL and
AR, corresponding to two distinct flight strategies seen in the extant birds;
medium-to-large birds tend to have high AR and WL to increase lift efficiency and
flight speed, and minimize energy costs for long flights, whereas small birds and
particularly small forest-dwelling birds tend to have lower AR and WL to increase
maneuverability during shorter and slower flights³⁰⁻³². *C. shifan*, with its low values of
both AR and WL, lies well within continuous flapping morphospace but adjacent to
the areas of overlap with flap-gliders and flap-bounders.

Most importantly, our analyses under phylogeny may indicate an evolutionary
trend affecting confuciusornithid flight, involving a tendency for WL and AR to
decrease from early-diverging confuciusornithids such as *E. zhengi* and *Y. confucii* to
late-diverging ones such as *C. sanctus* and *C. shifan*. This suggests that the evolution
of improved flight ability in confuciusornithids was likely associated with an
ecological shift to a more densely vegetated environment. *C. shifan* represents the
culmination of this evolutionary trend, at least among currently known
confuciusornithids, which is reflected in the proximity of this species to flap-bounding
morphospace in Figure 9. *Y. confucii* may independently become more adapted for
open environment or long flight. The modified nature of their wings and shoulder
girdles further support this inference.

[revised manuscript text omitted]

The SOC in the distal epiphysis of the alular metacarpal of *C. shifan* (Fig. 7a,b) is
presumably an autapomorphic feature, whose appearance may be explicable on the
basis of a combination of growth strategy and functional demands on the wing. A
recent study has demonstrated that epiphyseal SOC's tend to evolve as a response to
high mechanical stress⁴⁷, and in *C. shifan* the distal end of the alular metacarpal may
well have begun to experience considerable stresses as soon as a developing juvenile
began to fly. In extant birds the alular metacarpal plays an important role in aerial
maneuvers, and the wings of *C. shifan* appear better-adapted than those of other
confuciusornithids for high-maneuverability flight in relatively closed environments.
As a result, the distal end of the alular metacarpal might have been more frequently
subject to large stresses in *C. shifan* than in most confuciusornithids. Furthermore, the
histological features of *C. shifan* suggest a slower growth rate than that of other
confuciusornithids, based on Amprino's rule⁴⁹. For example, *C. shifan* has
longitudinally oriented vascular canals in the middle layer of the femoral compact
bone (Fig. 2), whereas concentrically oriented canals are more numerous in other
confuciusornithids¹³. If the spread of ossification from the primary centers of the long
bones into the epiphyses was also slow, then the establishment of an epiphyseal SOC
might have been necessary to reinforce the distal end of the alular metacarpal at a
comparatively early ontogenetic stage, when the juvenile was beginning to fly but

skeletal growth was still incomplete. The holotype of *C. shifan*, however, clearly
represents a mature individual in which the caudal distal condyle of the alular
metacarpal had fully ossified by the time of death, presumably via the normal
mechanism of spread from the diaphyseal POC. We infer that the secondary
epiphyseal ossification would have fused with the rest of the alular metacarpal to
remodel a cranial distal condyle of more normal appearance, had the individual lived
longer.

**Methods**

**Histological preparation of *C. shifan* (PMoL-AB00178).** We took the bone sample
near the mid-diaphysis of the right femur of PMoL-AB00178 (Fig. 1a). Cross-section
was prepared using standard procedures⁵⁰. The sample was embedded in
one-component resin (EXAKT Technovit 7200), which was then hardened in a light
polymerization device (EXAKT 520). Thin cross-section was cut using an accurate
circular saw (EXAKT 300CP). The section was ground down using the EXAKT
400CS grinding system until the desired optical transparency was obtained. The
histological section was examined under a polarized light microscope (ZEISS Axio
Imager 2 Pol), and photographed with a ZEISS AxioCam 705 digital microscope
camera.

**Phylogenetic analysis.** To investigate the systematic position of *C. shifan* relative to
other Mesozoic birds, a phylogenetic analysis was carried out using the most
comprehensive data matrix currently available for Mesozoic birds⁵¹, with *C. shifan*
added in based on scorings from PMoL-AB00178. The revised matrix consists of 81
taxa and 280 characters. The matrix was analyzed using TNT v1.5 (22)⁵² with default
settings. All characters were equally weighted, with 35 characters ordered; the dataset
was analyzed using the “New Technology search” methods with sectorial search,
ratchet, tree drift, and tree fusion with default settings; the minimum-length tree was
found in 10 replicates to recover as many tree islands as possible. The recovered trees
were then used as the basis for a traditional TBR search. Zero-length branches were

collapsed. Decay indices (Bremer support values) were calculated using the Bremer
script embedded in TNT, and absolute Bootstrap frequencies were calculated using
1000 pseudoreplicates in TNT with default settings.

**Aerodynamic analysis.** In order to assess the flight capabilities and strategies of
confuciusornithid birds, we estimated values of several flight parameters for 11
confuciusornithid individuals, including the holotypes of *C. dui*, *Ch. hengdaoziensis*, *E.*
*zhengi*, *Y. confucii* and *C. shifan* (see Supplementary tables S2, 3). Specifically, we
estimated the body mass (BM), wingspan (B), and lift surface area (SL) of these
specimens following the methods of Seranno et al. (2015, 2017)^{15,28}, and subsequently
used these estimates as a basis for calculating aspect ratio (AR, $AR = B^2/SL$) and wing
loading (WL, $WL = BM/SL$) values (see Supplementary table S3). The anatomical
data needed for these estimates were obtained by measuring the specimens directly
with digital calipers, or alternatively from high-resolution images using tpsDig 2.17⁵³
(available at: <https://sbmorphometrics.org/>) from measurements published in the
literature, or from estimated measurements based on the methods of Serrano et
al.2018²⁹ (see Supplementary table S2 for details).

[revised manuscript text omitted]

Figure 1. Photograph (a) and line drawing (b) of *Confuciusornis shifan* holotype
 (PMoL-AB00178). Arrow indicates the sampling position for the histological section.

Abbreviations: cav, caudal vertebra; cev, cervical vertebra; fu, furcula; ga, gastralia; lc,
 left coracoid; ldIII, left manual digit III; lfe, left femur; lh, left humerus; lil, left ilium;
 lis, left ischium; lm, left manus; lp, left pes; lra, left radiale; lr, left radius; ls, left
 scapula; lt, left tibiotarsus; lu, left ulna; lul, left ulnare; pu, pubis; py, pygostyle; r, rib;
 rc, right coracoid; rdcIII, caw of right manual digit III; rfe, right femur; rfi, right fibula;
 rh, right humerus; ris, right ischium; rm, right manus; rp, right pes; rra, right radiale; rr,
 right radius; rs, right scapula; rt, right tibiotarsus; ru, right ulna; rul, right ulnare; sk,
 skull; sy, synsacrum; tv, thoracic vertebra. Scale bars: 2 cm.

Figure 2. Photograph of a midshaft histological section of the right femur of the
*Confuciusornis shifan* holotype (PMoL-AB00178). White arrows indicate LAGs.

Abbreviations: ICL, inner circumferential layer; OCL, outer circumferential layer.

Scale bar: 100 μ m.

Figure 3. Photograph of the skull and mandible of the *Confuciusornis shifan* holotype

(PMoL-AB00178) in left ventrolateral view (a) with corresponding line drawing (b),

a close-up of the anterior margin of the orbit (c). White rectangle in (a) indicates the

region of (c). A line drawing of the posterior half of the mandible of *Confuciusornis*

*sanctus* IVPP V 13171 in Wang et al., 2018⁵ is appended below the line drawing of

the skull and mandible of the *Confuciusornis shifan*, and the arrow indicates the

position of the ventral process of the surangular. Abbreviations: an, angular; ar,

articular; cmf, caudal mandibular fenestra; d, dentary; f, frontal; j, jugal; l, lacrimal; m,

maxilla; o, orbit; rmf, rostral mandibular fenestra; n, nasal; p, parietal; pm, premaxilla;

q, quadrate; sa, surangular; sp, splenial. Scale bars: 1 cm; note that (c) is not to scale.

Figure 4. Photographs of the cervical vertebrae (a), synsacrum (b) and pygostyle (c)

of the *Confuciusornis shifan* holotype (PMoL-AB00178). Abbreviation: sy,

synsacrum. Arrows in (c) indicate the positions of foramina along the pygostyle. Scale

bars: 0.5 cm in (a) and (c); 1 cm in (b).

Figure 5. Photographs of the pectoral (a) and pelvic (b) girdle of the *Confuciusornis*
*shifan* holotype (PMoL-AB00178). Abbreviation: fu, furcula; lc, left coracoid; lil, left
ilium; lis, left ischium; ls, left scapula; pu, pubis; rc, right coracoid; rh, right humerus;
ris, right ischium; rs, right scapula. Scale bars: 1 cm.

Figure 6. Line drawings of the pectoral girdle, forelimb, hindlimb of the
 *Confuciusornis shifan* holotype (PMoL-AB00178), and other confuciusornithids, with
 shading to indicate lengths of stylopodial, zeugopodial and metapodial limb segments.
 Values near the segments represent ratios of segment length to femoral length. All
 drawings scaled to a common, arbitrary femoral length.

Figure 7. Photographs of the left (a) and right (b) carpometacarpi and right
 tarsometatarsus (d) of *Confuciusornis shifan* holotype (PMoL-AB00178), and the left
 alular metacarpal (c) of the confuciusornithid specimen PMoL-AB00150.
 Carpometacarpi in palmar view, tarsometatarsus in cranial view. Black and white
 arrows in (d) indicate the ridge-like process on metatarsal IV and the dorsal tubercle
 on metatarsal III, respectively. Abbreviations: cb, cushion-like bone; ep, extensor
 process; pp, pisiform process. Roman numerals in (d) identify metatarsals. Scale bars:
 0.25 cm.

Figure 8. Cladogram of Mesozoic birds showing the systematic position of
*Confuciusornis shifan*, and representing the strict consensus of the 192 most
parsimonious trees recovered in the phylogenetic analysis performed in this study
(length = 1404; consistency index = 0.277; retention index = 0.667). Bootstrap and
Bremer values are given in normal font and in bold italic font, respectively, near the
nodes to which they pertain.

Figure 9. Positions of 11 confuciusornithid specimens in a previously published
 morphospace defined by wing loading (WL) and aspect ratio (AR), showing the areas
 of morphospace occupied by extant birds with particular modes of flight²⁹. Specimens
 are indicated by black circles, and numbered in descending order of estimated AR.
 Specimens are holotypes of their respective species unless a number is indicated.

A new confuciusornithid bird with a secondary epiphyseal ossification reveals
phylogenetic changes in confuciusornithid flight mode

Renfei Wang^{1,2}, Dongyu Hu^{2*}, Meisheng Zhang¹, Shiyong Wang^{3,4}, Qi Zhao^{3,4}, Corwin
Sullivan^{5,6} & Xing Xu^{3,4*}

¹ College of Earth Sciences, Jilin University, Changchun 130061, China

² Shenyang Normal University, Paleontological Museum of Liaoning, Key Laboratory
for Evolution of Past Life in Northeast Asia, Liaoning Province, 253 North Huanghe
Street, Shenyang 110034, China

³ Key Laboratory of Vertebrate Evolution and Human Origins, Institute of Vertebrate
Paleontology and Paleoanthropology, Chinese Academy of Sciences, Beijing 100044,
China

⁴ Center for Excellence in Life and Paleoenvironment, Chinese Academy of Sciences,
Beijing 100044, China

⁵ Department of Biological Sciences, University of Alberta, Edmonton, AB T6G 2E9,
Canada

⁶ Philip J. Currie Dinosaur Museum, Wembley, AB T0H 3S0, Canada

Correspondence and requests for materials should be addressed to D.H. (email:
hdongyu@synu.edu.cn) and X.X. (email: xuxing@ivpp.ac.cn)

The confuciusornithids are the earliest known beaked birds, and also constitute
the only species-rich and morphologically diverse clade of Early Cretaceous birds that
existed prior to the cladogenesis of two major avialan groups Enantiornithes and
Ornithuromorpha. Here we report a new confuciusornithid species from the Lower
Cretaceous of western Liaoning, northeastern China. Compared to other
confuciusornithids and most contemporaneous enantiornithines and ornithuromorphs,
this new species and the recently reported *Yangavis confucii* show a higher level of
flight adaptation, but the wings of the two taxa differ from one another in many
respects. Aerodynamic analyses of the known confuciusornithids indicate
considerable variations of flight capability and style across the clade, and to a lesser
degree through ontogeny, and specifically suggest both a trend towards improved
flight capability and a change in flight strategy in confuciusornithid evolution. Most
significantly, this new confuciusornithid differs from other Mesozoic birds in having a
secondary epiphyseal ossification, located in the alular digit. This highly unusual
feature may reflect the functional demands of flight when skeletal growth was still
incomplete. The new find strikingly exemplifies the morphological, developmental
and functional diversity of the first beaked birds.

Introduction

Confuciusornithidae is a clade of Early Cretaceous pygostylian birds known from
the Jehol Biota of East Asia¹, representing the earliest known toothless, beaked birds.
Five genera and eleven species, recovered from the Dabeigou, Yixian and Jiufotang
formations (~135~120 Ma), have been described and assigned to this family, though
the validity of some species is questionable²⁻⁶. Confuciusornithids are the only
species-rich, morphologically diverse avialan clade known to have existed prior to the
cladogenesis of the major groups Enantiornithes and Ornithuromorpha, and are
represented by thousands of exceptionally well preserved specimens that collectively
provide rich information on confuciusornithid morphology, taxonomy, flight ability,
growth, diet and ecology^{3,5,7-13}. Here, we report a new confuciusornithid species,
*Confuciusornis shifan* sp. nov., from the Jiufotang Formation. *C. shifan* differs from

other confuciusornithids in a number of morphological and developmental features,
which have implications for understanding confuciusornithid **taxic** diversity,
morphological disparity, development, and flight behavior.

**Results**

**Systematic paleontology.**

**Aves Linnaeus, 1758**

Pygostylia Chiappe, 2002

Confuciusornithidae Hou et al. 1995

*Confuciusornis* Hou et al., 1995

*Confuciusornis shifan* sp. nov.

[revised manuscript text omitted]

The left and right dentaries are almost completely fused with each other along an
extended, ventrocaudally inclined mandibular symphysis as in *C. sanctus*³ (Fig. 3a,b).
The transversely compressed ventrocaudal process of the right dentary is visible, but
the caudalmost end of the process is missing. In the intact mandible, the ventrocaudal
process likely contacts the angular laterally, and extends back to the level of the
caudal margin of the anterior mandibular fenestra based on the position of the
impression of the missing end. The right angular and right surangular both appear
rod-shaped, and their caudal ends fuse into the articular. The surangular appears to
lack a ventral process as in *C. dui*, indicating that the presence of a triangular ventral

surangular process in *C. sanctus* is indeed a diagnostic feature of this species⁵ (Fig.
3b). The articular prominently projects medially to form a medial articular facet for
the quadrate¹⁶. In contrast to the displaced and partly overlapped left splenial, the right
splenial is nearly complete and preserved close to its original position. The splenial is
plate-like and rostrally forked, and protrudes into the rostral half of the rostral
mandibular fenestra. The dorsal margin of the splenial overlaps the medial surface of
the surangular, and the ventral margin, which does not contact the angular, is
thickened along most of its length and gently tapers caudally to contact the ventral
process of the surangular. Unlike **in other confuciusornithids**, the splenial of *C. shifan*
is centrally perforated by an oval foramen (Fig. 3a,b).

The cervical vertebrae are exposed in ventrolateral view, making it possible to see
that the cranial and caudal articular surfaces of each centrum are distinctly
heterocoelous (Fig. 4a) as previously reported for *Confuciusornis*⁵. The centra of the
thoracic series are laterally excavated as in *C. sanctus*, whereas such excavations are
absent in *Y. confucii*⁶ (Fig. 1). The **two cranialmost centra** each bear a prominent,
tapered ventral keel. The sacral vertebrae are completely fused with each other to
form a synsacrum (Fig. 4b). Based on the number of visible transverse processes, the
synsacrum consists of seven vertebrae as in other confuciusornithids³⁻⁶, and in ventral
view the total centra are transversely compressed to form a longitudinal ridge as in
some enantiornithines¹⁷ (Fig. 4b). However, the synsacrum of *C. sanctus* DNHM
D2454 (holotype of the putative species ***Confuciusornis feducciai***) appears
dorsoventrally compressed and transversely widened⁵. Five free caudal vertebrae are
visible between the synsacrum and the pygostyle (Fig. 1), whereas additional caudal
vertebrae are likely overlaid by the left ischium and some soft tissue. The caudal
vertebrae appear significantly shorter than the thoracic ones. The pygostyle of *C.*
*shifan* appears to be exposed in left lateral view (Fig. 4c). The pygostyle is
proportionately robust and subequal in length to the tarsometatarsus, as is generally
the case in confuciusornithids^{3,18}. The transversely thickened ventral margin of the
pygostyle is distinctly made up of the co-ossified centra, which are thinned caudally.
The ventral surface does not form a longitudinal keel as in *C. sanctus* IVPP V 12352

(holotype of the putative species *Jinzhouornis zhangjiyingia*)^{5,18}. Some 11 small round
foramina form a longitudinal row along the whole lateral furrow of the pygostyle (Fig.
4c). The foramina are closer to the ventral margin of the pygostyle than the dorsal
margin, and gradually become smaller and more tightly crowded together towards the
caudal end of the series. Similar foramina occur in other *confuciusornithine*
specimens, but are fewer in number and only present on the cranial part of the
pygostyle^{18,19}.

The coracoids are fused with the scapulae to form scapulocoracoids. The left and
right scapulocoracoids are exposed in lateral and medial views, respectively (Fig. 5a).
The long axes of the scapula and coracoid define an acute angle as in *Y. confucii*⁶ and
*more advanced* birds (Fig. 5a, 6). This angle is approximately 90° in *C. sanctus*³. A
laterally directed glenoid facet is visible in the left scapulocoracoid. The cranial and
caudal margins of the glenoid facet project more strongly than the dorsal and ventral
ones, implying that the configuration of the facet probably did not strongly limit the
elevation and depression of the wing in flapping flight. The acromion of the scapula is
well-developed. The shaft of the scapula curves slightly downward; the cranial half of
the shaft appears rod-like and slightly compressed dorsoventrally, and while the
caudal half is more compressed mediolaterally and widens significantly in the
dorsoventral direction before tapering to a point as in *more advanced* birds²⁰ (Fig. 5a,
6). In other *confuciusornithids*, the scapula is generally straight, and neither flares
partway along its length nor tapers caudally³. The coracoid is strut-like and
approximately half the length of the scapula. Its proximal end appears dorsoventrally
thicker than transversely wide and projects slightly cranially to form an acrocoracoid
process as in *Y. confucii*⁶, whereas its distal end is compressed dorsoventrally. The
acrocoracoid process is not well-developed in *C. sanctus*³. The lateral surface of the
coracoid is excavated to form a shallow groove, which gradually becomes narrower
and shallower distally as in *Y. confucii*⁶. The medial margin of the coracoid is strongly
compressed to form a longitudinal keel, except at the proximal end of the bone. The
furcula is mostly exposed in caudal view. As in other *confuciusornithids*, the furcula is
boomerang-shaped and without a distinct tubercle³⁻⁶ (Fig. 5a, 6). The proximomedial

corner of the right clavicular ramus is inflated caudally to form a bulbous structure
that contacts the medial face of the proximal end of the right scapulocoracoid. and
meanwhile the proximolateral corner is more craniocaudally compressed than the rest
of the ramus and slightly projects laterally, to form a distinct impression on the caudal
surface of the corner. Similar structure is not seen in other confuciusornithids.

The forelimb is subequal in length to the hindlimb, with a forelimb (humerus +
ulna) / hindlimb (femur + tibiotarsus) ratio of 1.02 (Fig. 1; Supplementary table S1).
The equivalent ratio is 0.93 in *C. hengdaoziensis*, 0.96 in *E. zhengi*, 1.06 in *C. dui*,
0.91~1.08 in *C. sanctus* and 1.19 in *Y. confucii*⁶ (Fig. 6). The humerus is typical of
confuciusornithids in having an expanded and perforated deltopectoral crest. However,
the crest is less prominent overall than those of other confuciusornithids except *C. dui*⁵,
and the point of the greatest prominence is at the distal end of the crest rather than in the
middle as in other confuciusornithids³⁻⁶ (Fig. 5a, 6). The dorsodistal corner of the crest
projects distally, so that the distal margin appears concave. Distally, a large fossa for the
brachialis muscle is present proximal to the dorsal and ventral condyles. The dorsal
epicondyle is better developed than the ventral epicondyle, and the former projects
sharply dorsally. Both the ulna and the radius are shorter than the humerus, as in other
confuciusornithids (Fig. 1, 6; Supplementary table S1, 2). A concave ventral cotyle, a
flat dorsal cotyla and a concave incisura radialis are clearly visible on the proximal
end of the right ulna, whereas a distinctly projecting olecranon process is lacking. The
proximal humeral articular facet of the radius is flat and the bicipital tubercle is
situated on the cranial face of the proximal end of the radius. Both radialia are in their
original positions relative to the radii and carpometacarpi (Fig. 1). The left radiale are
exposed in palmar and obliquely proximal view and the right in palmar and obliquely
distal view. The articular facets for the radius and semilunate carpal are both
significantly concave. The cranial surface of the radiale is convex and forms a small
tubercle on the proximal side, and the caudal surface is slightly concave and directed
somewhat proximally to articulate with the ulna.

The semilunate carpal is completely fused with metacarpals II and III to form a
carpometacarpus (Fig. 1, 7a,b). Metacarpal III is much more slender, and slightly

shorter, than metacarpal II, and contact between these metacarpals is limited to their
proximal ends as **in enantiornithines**²¹. The middle parts of metacarpals II and III are
constricted, and the narrowest part of metacarpal III is less than half the width of the
narrowest part of metacarpal II as in other confuciusornithids. However, the
intermetacarpal space appears relatively longer and significantly wider than in other
confuciusornithids, as in some **advanced birds**²⁰. The proximal end of the
carpometacarpus forms a pulley-like carpal trochlea as in *C. sanctus*³ and extant birds.
A distinct fossa is present in the region where the semilunate carpal is fused to
metacarpals II and III. The bump-like pisiform process is located near the proximal
end of metacarpal II. The alular metacarpal is about one third the length of metacarpal
II as in *C. sanctus*³ and *E. zhengi*⁴, rather than half the length as in *Ch.*
*hengdaoziensis*³ (Fig. 6, 7a,b). It is not fused with the carpometacarpus (Fig. 7a,b).
The caudal portion of the alular metacarpal extends farther proximally than the cranial
portion so that the proximal surface of the metacarpal is concave, forming a cranial
carpal fovea. The proximal two thirds of the cranial margin projects cranially to form
the extensor process. The craniocaudal width of the process is nearly half that of the
distal articular surface of the alular metacarpal (Fig. 7a,b,c). Such a well-developed
extensor process is not seen in contemporaneous basal birds, or **in most**
**enantiornithines** and **ornithuromorphs**^{20,21}. Unlike in previously known
confuciusornithids, the caudal distal condyle of the alular metacarpal is
well-developed, and projected more distoventrally than the rest of the metacarpal,
evident in palmar view. The proximal phalanx of digit II is robust, maintains a
constant diameter throughout its entire length, and is slightly shorter than the
intermediate phalanx as in other confuciusornithids except *Ch. hengdaoziensis*³(Fig.
1). In *Ch. hengdaoziensis*, the proximal phalanx is slightly longer than the
intermediate one³. The ungual phalanges of the alular digit and digit III are large and
curved, whereas that of digit II is small as in most confuciusornithids. In *Y. confucii*,
however, the ungual phalanx of digit II is large⁶.

Among the fully articulated left and right pelvic girdles, only the pubes are
relatively completely preserved (Fig. 5b). The right ilium appears relatively low, with

a slightly convex dorsal margin and a laterally projected antitrochanter. The pubis,
remarkably, is longer than the femur, being rod-like, slender, and strongly retroverted.
Because of a sharp bend in the pubic shaft, the long axis of the pubic symphysis is
nearly perpendicular to that of the shaft's proximal portion, a condition resembling
that seen in derived ornithuromorph birds.

The femoral head, neck and trochanteric crest are well-developed. The proximal
tarsals are completely fused with each other and with the tibia, forming a tibiotarsus.
The proximal ends of metatarsals II-IV are fused with each other and with the distal
tarsals, forming a tarsometatarsus. The proximal end of metatarsal III appears
transversely compressed between metatarsals II and IV in cranial view (Fig. 7d),
whereas this metatarsal in other confuciusornithids does not change in width along its
length³⁻⁶. An oval tubercle presumably for insertion of *M. tibialis cranialis* is located
on the dorsal surface near the proximal end as in *C. sanctus*³, but similar structure
does not exist on metatarsal II. A ridge-like process is present on the lateral margin of
the distal third of metatarsal IV (Fig. 7d), but it is not seen in other confuciusornithids.
In pedal digits III-IV, the penultimate phalanx is longer than the preceding phalanx, as
in other confuciusornithids⁶. However, the penultimate phalanx of pedal digit II is
subequal in length to the preceding phalanx.

*Secondary epiphyseal ossification* The most unusual skeletal feature exhibited by
PMoL-AB00178 is the presence of a cushion-like small bone between the distal end
of the alular metacarpal and the proximal end of the first alular phalanx (Fig. 7a,b). In
the left manus, the small bone contacts the cranial portions of the opposing articular
surfaces of the metacarpal and phalanx (Fig. 7a); in the right manus, the small bone
again contacts the cranial portion of the proximal articular surface of the first alular
phalanx, but rests against the cranial margin of the left alular metacarpal due to the
slight disarticulation of the alular digit (Fig. 7b). We identify this small bone as an
ossification that arose from a secondary ossification center within the distal epiphysis
of the alular metacarpal and is equivalent to the medial condyle seen on the
metacarpal's distal articular surface in other confuciusornithids (Fig. 7c), based on the

following evidence: (1) in the right manus, the small bone remains in apparently
natural alignment with the alular metacarpal, but not with the first alular phalanx; (2)
in other confuciusornithids the distal articular surface of the alular metacarpal is
ginglymoid⁶, and the caudal condyle protrudes slightly more distally than the cranial
one (Fig. 7c), whereas in both alular metacarpals of PMoL-AB00178 the caudal
condyle is distally prominent and cranial condyle is absent; and (3) in both first alular
phalanges of PMoL-AB00178, the proximal articular surface is similar in morphology
to those of other confuciusornithids, and a well-developed flexor tubercle is visible on
the left alular phalanx.

Discussion

Our phylogenetic analysis places *C. shifan* as the sister taxon to *C. sanctus*, and *E.*
*zhengi*, *C. dui*, *Y. confucii* and *Ch. hengdaoziensis* as successive outgroups to the *C.*
*sanctus* + *C. shifan* clade. However, the various confuciusornithid subclades involved
in this arrangement are not well supported, as indicated by their low Bremer and
Bootstrap values (Fig. 8).

The discovery of *C. shifan* sheds new light on morphological evolution and flight
adaptation in confuciusornithid birds. Among early pygostylians, both
confuciusornithid and enantiornithines have relatively large pygostyles¹⁸, although the
exact number of caudal vertebrae involved in the pygostyle is typically difficult to
determine due to their complete coossification. Perforated pygostyles, such as that of
PMoL-AB00178, are thought to be the result of incomplete fusion between adjacent
neural arches, and the foramina are finally enclosed^{18,19}. The pygostyle of
PMoL-AB00178 then presumably exemplifies the developmental stage at which
fusion among the neural arches was partial rather than complete. Given the presence
of 11 preserved foramina, the number of caudal vertebrae within the pygostyle is
estimated to be 12. If the estimate for the pygostyle is accurate and PMoL-AB00178
has seven free caudal vertebrae as in *C. sanctus*³, then 19 caudal vertebrae are present
in the tail as a whole (Fig. 1, 4c), a total only slightly less than the count of 21-23 free
caudal vertebrae reported for the long bony tail of *Archaeopteryx*²². This comparison

implies that the transition in birds from a long bony tail like that plesiomorphically
present in reptiles to a short one ending in a pygostyle was driven primarily by
shortening of the individual caudal vertebrae, rather than by a reduction in vertebral
count. The fact that the pygostyle is the only compound bone in PMoL-AB00178 to
show incomplete fusion also indicates that the tail may not have played a major role in
generating aerodynamic forces during flight.

The flight capability of confuciusornithids has been debated in previous studies.
Some authors have considered confuciusornithids competent flyers based on a number
of morphological features, including long wings with strongly asymmetrical flight
feathers, strut-like coracoids, a keeled sternum, and enlarged major manual digits²³,
whereas others have suggested more limited flight capabilities¹² based on functional
analysis of the morphology of the shoulder joint⁷ or the flight feathers⁹. Nevertheless,
there has been broad agreement that confuciusornithids display a more primitive suite
of flight-related characters than contemporaneous enantiornithines and
ornithuromorphs^{1,7}, suggesting inferior flight performance. However, morphological
evidence from both *C. shifan* and the recently discovered *Y. confucii* indicates that
some variation in flight capability existed within Confuciusornithidae, as both these
species appear to have been unusually well-adapted for flight, albeit in different ways.
*Y. confucii* and *C. shifan* both exhibit a more specialized flight apparatus than is
present in other confuciusornithids, or even in most contemporaneous enantiornithines
and ornithuromorphs. In *Y. confucii* the main modification is elongation of the
forelimb to increase the wing area as in sapeornithids²⁴ and some enantiornithine and
ornithuromorph birds. In *C. shifan* the main modifications involve refinements to the
pectoral girdle, humerus and carpometacarpus, which to some extent resemble their
counterparts in enantiornithine and ornithuromorph birds. In particular, some skeletal
processes to which flight muscles would have attached are particularly well developed.
A few such features, including the extensor process of the alular metacarpal and the
pisiform process of the carpometacarpus, are seen only in the contemporaneous
enantiornithine *Xiangornis*²¹, a few Early and Late Cretaceous ornithuromorphs²⁵, and
extant flying birds (Fig. 7a,b). In extant birds, the extensor process is the point of

insertion for M. extensor metacarpi radialis, the primary muscle involved in extension
of the wrist joint, and the pisiform process serves for attachment of the retinaculum
flexorium, and as a pulley changing the direction of the tendon of M. flexor digitorum
profundus²⁶, which crosses the wrist and inserts on the distal phalanx of the major
digit. A well-developed extensor process and pisiform process can accordingly
increase manipulating and controlling capabilities of the distal part of the wing during
flight²⁷, collectively suggesting greater flight capability in *C. shifan* than in other
confuciusornithids.

A recent study showed that most flight modes seen in extant birds (e.g.,
continuous flapping, flap-gliding, flap-bounding, thermal soaring) could potentially
have been represented among early birds²⁸. The flight parameter estimates obtained
for confuciusornithids in that analysis would suggest that they were continuous
flappers, but only the holotype of *E. zhengi* and a few specimens of *C. sanctus* were
considered. In order to further assess the flight capabilities and strategies of
confuciusornithid birds, we followed the methods of Serrano et al. (2017)²⁸ (also see
Aerodynamic analysis in Methods below) to estimate flight parameters for 11
confuciusornithid specimens, including the only known specimens of *Y. confucii* and
*C. shifan* (Supplementary table S2, 3), and plotted estimates of two key parameters for
these confuciusornithids, namely aspect ratio (AR) and wing loading (WL), on a
previously published figure showing the regions of AR-WL morphospace associated
with the main flight modes in extant birds²⁹. AR is the ratio of wing length to wing
width, representing a simple measure of wing shape, and WL is the ratio of body mass
to the total area of the two wings and the part of the trunk between them, representing
the load that must be borne by each unit area of the available lift-producing surface.
Both metrics are commonly used in assessing the flight capabilities and strategies of
flying animals. Our analyses show that both AR and WL vary considerably among
confuciusornithid specimens and even among specimens of *C. sanctus* (Supplemental
table S3; Fig. 9), as found by previous researchers²⁸. Though two specimens of *C.*
*sanctus* fall slightly into the area of morphospace where extant facultative flap-gliders
overlap with continuous flappers, most confuciusornithids plot at least narrowly

outside the flap-gliding area, and within the area occupied solely by continuous
flappers (Fig. 9). Interestingly, confuciusornithids are distributed throughout a large
part of the continuous flapping morphospace, and two ends represented by *E. zhengi*
and *Y. confucii* with the highest WL and AR, and *C. shifan* with the lowest WL and
AR, corresponding to two distinct flight strategies seen in the extant birds;
medium-to-large birds tend to have high AR and WL to increase lift efficiency and
flight speed, and minimize energy costs for long flights, whereas small birds and
particularly small forest-dwelling birds tend to have lower AR and WL to increase
maneuverability during shorter and slower flights³⁰⁻³². *C. shifan*, with its low values of
both AR and WL, lies well within continuous flapping morphospace but adjacent to
the areas of overlap with flap-gliders and flap-bounders.

Most importantly, our analyses under phylogeny may indicate an evolutionary
trend affecting confuciusornithid flight, involving a tendency for WL and AR to
decrease from early-diverging confuciusornithids such as *E. zhengi* and *Y. confucii* to
late-diverging ones such as *C. sanctus* and *C. shifan*. This suggests that the evolution
of improved flight ability in confuciusornithids was likely associated with an
ecological shift to a more densely vegetated environment. *C. shifan* represents the
culmination of this evolutionary trend, at least among currently known
confuciusornithids, which is reflected in the proximity of this species to flap-bounding
morphospace in Figure 9. *Y. confucii* may independently become more adapted for
open environment or long flight. The modified nature of their wings and shoulder
girdles further support this inference.

[revised manuscript text omitted]

The SOC in the distal epiphysis of the alular metacarpal of *C. shifan* (Fig. 7a,b) is
presumably an autapomorphic feature, whose appearance may be explicable on the
basis of a combination of growth strategy and functional demands on the wing. A
recent study has demonstrated that epiphyseal SOC's tend to evolve as a response to
high mechanical stress⁴⁷, and in *C. shifan* the distal end of the alular metacarpal may
well have begun to experience considerable stresses as soon as a developing juvenile
began to fly. In extant birds the alular metacarpal plays an important role in aerial
maneuvers, and the wings of *C. shifan* appear better-adapted than those of other
confuciusornithids for high-maneuverability flight in relatively closed environments.
As a result, the distal end of the alular metacarpal might have been more frequently
subject to large stresses in *C. shifan* than in most confuciusornithids. Furthermore, the
histological features of *C. shifan* suggest a slower growth rate than that of other
confuciusornithids, based on Amprino's rule⁴⁹. For example, *C. shifan* has
longitudinally oriented vascular canals in the middle layer of the femoral compact
bone (Fig. 2), whereas concentrically oriented canals are more numerous in other
confuciusornithids¹³. If the spread of ossification from the primary centers of the long
bones into the epiphyses was also slow, then the establishment of an epiphyseal SOC
might have been necessary to reinforce the distal end of the alular metacarpal at a
comparatively early ontogenetic stage, when the juvenile was beginning to fly but

skeletal growth was still incomplete. The holotype of *C. shifan*, however, clearly
represents a mature individual in which the caudal distal condyle of the alular
metacarpal had fully ossified by the time of death, presumably via the normal
mechanism of spread from the diaphyseal POC. We infer that the secondary
epiphyseal ossification would have fused with the rest of the alular metacarpal to
remodel a cranial distal condyle of more normal appearance, had the individual lived
longer.

**Methods**

**Histological preparation of *C. shifan* (PMoL-AB00178).** We took the bone sample
near the mid-diaphysis of the right femur of PMoL-AB00178 (Fig. 1a). Cross-section
was prepared using standard procedures⁵⁰. The sample was embedded in
one-component resin (EXAKT Technovit 7200), which was then hardened in a light
polymerization device (EXAKT 520). Thin cross-section was cut using an accurate
circular saw (EXAKT 300CP). The section was ground down using the EXAKT
400CS grinding system until the desired optical transparency was obtained. The
histological section was examined under a polarized light microscope (ZEISS Axio
Imager 2 Pol), and photographed with a ZEISS AxioCam 705 digital microscope
camera.

**Phylogenetic analysis.** To investigate the systematic position of *C. shifan* relative to
other Mesozoic birds, a phylogenetic analysis was carried out using the most
comprehensive data matrix currently available for Mesozoic birds⁵¹, with *C. shifan*
added in based on scorings from PMoL-AB00178. The revised matrix consists of 81
taxa and 280 characters. The matrix was analyzed using TNT v1.5 (22)⁵² with default
settings. All characters were equally weighted, with 35 characters ordered; the dataset
was analyzed using the “New Technology search” methods with sectorial search,
ratchet, tree drift, and tree fusion with default settings; the minimum-length tree was
found in 10 replicates to recover as many tree islands as possible. The recovered trees
were then used as the basis for a traditional TBR search. Zero-length branches were

collapsed. Decay indices (Bremer support values) were calculated using the Bremer
script embedded in TNT, and absolute Bootstrap frequencies were calculated using
1000 pseudoreplicates in TNT with default settings.

**Aerodynamic analysis.** In order to assess the flight capabilities and strategies of
confuciusornithid birds, we estimated values of several flight parameters for 11
confuciusornithid individuals, including the holotypes of *C. dui*, *Ch. hengdaoziensis*, *E.*
*zhengi*, *Y. confucii* and *C. shifan* (see Supplementary tables S2, 3). Specifically, we
estimated the body mass (BM), wingspan (B), and lift surface area (SL) of these
specimens following the methods of Seranno et al. (2015, 2017)^{15,28}, and subsequently
used these estimates as a basis for calculating aspect ratio (AR, $AR = B^2/SL$) and wing
loading (WL, $WL = BM/SL$) values (see Supplementary table S3). The anatomical
data needed for these estimates were obtained by measuring the specimens directly
with digital calipers, or alternatively from high-resolution images using tpsDig 2.17⁵³
(available at: <https://sbmorphometrics.org/>) from measurements published in the
literature, or from estimated measurements based on the methods of Serrano et
al. 2018²⁹ (see Supplementary table S2 for details).

[revised manuscript text omitted]

Figure 1. Photograph (a) and line drawing (b) of *Confuciusornis shifan* holotype
 (PMoL-AB00178). Arrow indicates the sampling position for the histological section.

Abbreviations: cav, caudal vertebra; cev, cervical vertebra; fu, furcula; ga, gastralia; lc,
 left coracoid; ldIII, left manual digit III; lfe, left femur; lh, left humerus; lil, left ilium;
 lis, left ischium; lm, left manus; lp, left pes; lra, left radiale; lr, left radius; ls, left
 scapula; lt, left tibiotarsus; lu, left ulna; lul, left ulnare; pu, pubis; py, pygostyle; r, rib;
 rc, right coracoid; rdcIII, caw of right manual digit III; rfe, right femur; rfi, right fibula;
 rh, right humerus; ris, right ischium; rm, right manus; rp, right pes; rra, right radiale; rr,
 right radius; rs, right scapula; rt, right tibiotarsus; ru, right ulna; rul, right ulnare; sk,
 skull; sy, synsacrum; tv, thoracic vertebra. Scale bars: 2 cm.

Figure 2. Photograph of a midshaft histological section of the right femur of the
*Confuciusornis shifan* holotype (PMoL-AB00178). White arrows indicate LAGs.

Abbreviations: ICL, inner circumferential layer; OCL, outer circumferential layer.

Scale bar: 100 μ m.

Figure 3. Photograph of the skull and mandible of the *Confuciusornis shifan* holotype

(PMoL-AB00178) in left ventrolateral view (a) with corresponding line drawing (b),

a close-up of the anterior margin of the orbit (c). White rectangle in (a) indicates the

region of (c). A line drawing of the posterior half of the mandible of *Confuciusornis*

*sanctus* IVPP V 13171 in Wang et al., 2018⁵ is appended below the line drawing of

the skull and mandible of the *Confuciusornis shifan*, and the arrow indicates the

position of the ventral process of the surangular. Abbreviations: an, angular; ar,

articular; cmf, caudal mandibular fenestra; d, dentary; f, frontal; j, jugal; l, lacrima; m,

maxilla; o, orbit; rmf, rostral mandibular fenestra; n, nasal; p, parietal; pm, premaxilla;

q, quadrate; sa, surangular; sp, splenial. Scale bars: 1 cm; note that (c) is not to scale.

Figure 4. Photographs of the cervical vertebrae (a), **synsacrum** (b) and pygostyle (c)

of the *Confuciusornis shifan* holotype (PMoL-AB00178). Abbreviation: sy,

synsacrum. Arrows in (c) indicate the positions of foramina along the pygostyle. Scale

bars: 0.5 cm in (a) and (c); 1 cm in (b).

Figure 5. Photographs of the pectoral (a) and pelvic (b) girdle of the *Confuciusornis*
*shifan* holotype (PMoL-AB00178). Abbreviation: fu, furcula; lc, left coracoid; lil, left
ilium; lis, left ischium; ls, left scapula; pu, pubis; rc, right coracoid; rh, right humerus;
ris, right ischium; rs, right scapula. Scale bars: 1 cm.

**Figure 6.** Line drawings of the pectoral **girdle**, **forelimb**, **hindlimb** of the
 *Confuciusornis shifan* holotype (PMoL-AB00178), and other confuciusornithids, with
 shading to indicate lengths of stylopodial, zeugopodial and metapodial limb segments.
 Values near the segments represent ratios of segment length to femoral length. All
 drawings scaled to a common, arbitrary femoral length.

Figure 7. Photographs of the left (a) and right (b) carpometacarpi and right
 tarsometatarsus (d) of *Confuciusornis shifan* holotype (PMoL-AB00178), and the left
 alular metacarpal (c) of the confuciusornithid specimen PMoL-AB00150.
 Carpometacarpi in palmar view, tarsometatarsus in cranial view. Black and white
 arrows in (d) indicate the ridge-like process on metatarsal IV and the dorsal tubercle
 on metatarsal III, respectively. Abbreviations: cb, cushion-like bone; ep, extensor
 process; pp, pisiform process. Roman numerals in (d) identify metatarsals. Scale bars:
 0.25 cm.

Figure 9. Positions of 11 confuciusornithid specimens in a previously published
 morphospace defined by wing loading (WL) and aspect ratio (AR), showing the areas
 of morphospace occupied by extant birds with particular modes of flight²⁹. Specimens
 are indicated by black circles, and numbered in descending order of estimated AR.
 Specimens are holotypes of their respective species unless a number is indicated.

Reviewer # 1

If species-rich relative to others, the group would naturally be morphologically more diverse, and it seems to me that this phrase is redundant. Removing it would not interfere with the uniqueness of the family.

As commented above, if species-rich relative to others, the group would naturally be morphologically more diverse, and it seems to me that this phrase is redundant. Removing it would not interfere with the uniqueness of the family.

Following the reviewer's suggestion, we have removed the phrase "morphologically diverse".

Can you be more specific about age? It would be a critical piece of information in addressing evolution and diversification of confuciusornithids through Yixian and Jiufotang time.

We have added more specific information on the geological age of the specimen.

The distalmost three foramina are hard to see in the fig. 4c. Authors might want to include an additional image to clearly show that these three are present.

In order to capture a clearer image, we cleaned the distalmost end of the pygostyle more thoroughly than before. Unfortunately, we found that what we had originally considered to be the two distalmost foramina in the pygostyle were actually fossae. They may represent foramina that had been filled in with bone, based on their spacing relative to each other and to the foramina proximal to them, but we nevertheless decided not to count them. Furthermore, the one proximalmost foramen of the pygostyle was not counted in the original manuscript. Therefore, the revised number of foramina is ten.

The sentence is too long and difficult to follow. Please separate it into two or more concise sentences.

This sentence, and the others flagged with similar annotations by the reviewer, have been either reworded for greater readability or essentially removed from the manuscript as a result of changes to the scientific argumentation.

I think this is an interesting observation and the discussion deserves expansion. The number of distal caudals forming pygostyle is also observable in *Fukuipteryx* and I would appreciate its comparison to the one in *C. shifan* so that the evolution of pygostyle in fossil birds can be highlighted further. Rashid

et al. (2018) should also help as it describes development of chicken pygostyle from the embryo to young adult, which potentially demonstrates evolutionary pathway of the distal caudals.

Pygostyles have been found in several non-avian theropods and there has been a debate whether or not these non-avian pygostyles are precursor of avian pygostyles, originating deep in the maniraptoran phylogeny. Comparison of the number of distal caudals contributing the pygostyles between non-avian and avian theropods may help to answer this question.

We have expanded our comparisons among avian pygostyles and our discussion of pygostyle formation in various taxa, including some non-avian ones.

I understand that *C. shifan*, as well as *Y. confucii*, might have been more adapted for flight, as authors argue. However, I feel it is more interesting if they could include a discussion about the evolution of flight capability among confuciusornithids, taking the phylogeny of the group into account.

In their study, *Y. confucii* is not most derived among the group, yet displaying better flight adaptation than *C. sanctus* and *Ch. hengdaoziensis*. While *C. shifan* is the most derived among the group with *C. sanctus*, the latter does not exhibit notable flight adaptation. Are there any shared morphological factors in *Y. confucii* and *C. shifan* that might explain their adaptation to flight in spite of their positions in the confuciusornithid phylogeny?

We have further stressed the flight-related features shared by *Y. confucii* and *C. shifan*, and have also reframed our discussion of the evolution of flight capability and adaptation among confuciusornithids to argue that different confuciusornithids were well-adapted to different styles of flight. In particular, *Y. confucii* and *C. shifan* appear to have been well suited to fast, long-distance flight and slow, short-distance flight, respectively. From this perspective, it is not surprising that two species occupying disparate parts of the confuciusornithid phylogenetic tree evolved different types of aerial competence.

This is an interesting argument, and I would recommend to put flight apparatus features into consideration. As authors suggest in the lines 378-409, flight apparatus is variously obtained among confuciusornithids, some more morphologically adapted and others less. Not only making an argument about flight capability in terms of WL and AR with respect to the phylogeny, authors could look at the flight apparatus and see if such trends in WL and AR coincide with presence (or numbers of) flight-adapting apparatus.

We have examined confuciusornithid flight apparatus evolution, but have not been able to identify trends in. However, we have also added some discussion of this issue to the revised ms.

One thing that I am concerned is that the authors jump into a suggestion that the evolutionary flight adaptation among confuciusornithids is linked to an ecological shift toward densely vegetated environment. I see that small extant birds would have lower AR and WL to increase maneuverability for short flights, particularly in forest-dwelling ones. However, for confuciusornithids, it is difficult to determine whether such lower AR and WL is for forest-dwelling habitat or for other reasons. Is there any geological evidence that Jehol became more vegetated during the evolution of confuciusornithids, or any additional fossil evidence suggests that late-diverging members were more adapted to the densely vegetated environment (diet, etc.? If so, authors need to support their argument with the additional evidence. If not, "ecological shift to a more densely vegetated area" sounds a little too much speculation. For example, if lower AR and WL is more typical for smaller birds, decreasing AR and WL among confuciusornithids toward more derived members may simply reflect decreased adult body size through the evolution of the family.

We have deleted the suggestion that confuciusornithids adapted to a more densely vegetated environment in the course of their evolution, as we agree that evidence for an increase in vegetation over the temporal range of the clade is lacking. We can not identify a trend of decreasing body size paralleling the decrease in WL and AR, but we have added some discussion of this topic.

From lines 358-453, authors discuss the degrees of flight adaptation of confuciusornithids and their evolutionary trend. For readers to know where the discussion is heading to, it is recommended that the whole part be sub-headed such as "Flight Adaptation among Confuciusornithids".

We have added sub-headings to the discussion as suggested.

Authors might want to show further evidence to support that the extra elements in *C. shifan* in fact represent epiphyseal SOC. How do they compare to modern epiphyseal SOC in crown aves morphologically?

Crown birds uniformly lack epiphyseal SOC in the manus, and accordingly do not provide pertinent comparative information. However, we feel that the evidence mentioned in our original manuscript strongly supports the inference that the "extra" element is indeed derived from an epiphyseal SOC, partly because no obvious alternative explanation is available. Even so, we have still adduced further evidence as suggested by another reviewer.

spell out SOC.

(The term secondary ossification center first appears in the manuscript in the line 339. Authors may add (SOC) here and keep using it throughout the rest of the paper).

This advice has been followed.

Reviewer # 2

I would say jeholornithiforms fit this definition as-is. I suggest "pygostylian birds" or "short-tailed birds" or modifying the claim.

Following the referee's suggestion, we have revised the definition to indicate that confuciusornithids constitute the only pygostylian clade to have existed prior to Ornithothoraces.

This sounds like the paper is summarizing past work, make clear this is novel.

We have revised this part of the abstract.

Seems too much jargon for a key point of the abstract. Perhaps "an extra cushion-like bone in the first digit of the wing".

We have revised the text as suggested.

I suggest placing the citations after each item in the list for easier reader reference.

Suggestion followed.

Also relevant to cite here:

Navalón, G., Meng, Q., Marugán-Lobón, J., Zhang, Y., Wang, B., Xing, H., Liu, D. and Chiappe, L.M., 2018. Diversity and evolution of the Confuciusornithidae: Evidence from a new 131-million-year-old specimen from the Huajiying Formation in NE China. *Journal of Asian Earth Sciences*, 152, pp.12-22.

Falk, A., O'Connor, J., Wang, M. and Zhou, Z., 2019. On the preservation of the beak in *Confuciusornis* (Aves: Pygostylia). *Diversity*, 11(11), p.212.

Miller, C.V., Pittman, M., Kaye, T.G., Wang, X., Bright, J.A. and Zheng, X., 2020. Disassociated rhamphotheca of fossil bird *Confuciusornis* informs early beak reconstruction, stress regime, and developmental patterns. *Communications biology*, 3(1), pp. 519

Zheng, X., O'Connor, J., Wang, Y., Wang, X., Xuwei, Y., Zhang, X. and Zhou, Z., 2020. New information on the keratinous beak of *Confuciusornis* (Aves: Pygostylia) from two new specimens. *Frontiers in Earth Science*, p.367.

Marugán-Lobón, J. and Chiappe, L. M., 2022. Ontogenetic niche shifts in the Mesozoic bird *Confuciusornis sanctus*. *Current Biology*.

Also current citation 16, "Cranial morphology of the Early Cretaceous bird Confuciusornis"

We have added most of these references.

You use Avialae earlier in the manuscript, which implies you use Aves to refer to crown birds (not containing Confuciusornithidae). This seems consistent with the use of avian in the Discussion. Either change this line to Avialae Gauthier, 1986 or change avialan usage.

Either way the usage of Aves/Avialae/Neornithes should be explicitly stated in the Introduction. Pittman et al. 2020 is a good recent reference for usage discussion.

Gauthier, J., 1986. Saurischian monophyly and the origin of birds. *Memoirs of the California Academy of sciences*, 8, pp.1-55.

Pittman, M., O'Connor, J., Field, D.J., Turner, A.H., Ma, W. and Makovicky, P., 2020. Chapter 1: Pennaraptoran Systematics. *Bull Am Mus Nat Hist*, pp.7-36.

We have switched "Aves" to "Avialae" here. We think the usages of Avialae, Pygostylia are familiar enough that these don't need to be explained in the Introduction.

Is bed information available? See e.g.

Lijun, Z., Yajun, Y., Lidong, Z., Shengzhe, G., Wuli, W. and Shaolin, Z., 2007. Precious Fossil-bearing Beds of the Lower Cretaceous Jiufotang Formation in Western Liaoning Province, China. *Acta Geologica Sinica-English Edition*, 81(3), pp.357-364.

We have added more precise stratigraphic information.

The diagnosis shows that PMoL-AB00178 represents a unique confuciusornithid, but not that it belongs in the genus *Confuciusornis*. As best I can tell after Wang et al 2019 (current ref 5) there is no genus diagnosis for *Confuciusornis*, and your phylogeny finds the genus to be paraphyletic.

Thus I suggest referring this specimen to a new genus. Alternatively, you could create a new genus diagnosis for *Confuciusornis* and more thoroughly discuss the recovery of *C. dui* as an outgroup in your phylogeny.

We appreciate the reviewer's suggestion. Although we have not identified any features that are uniquely shared by *C. sanctus* and *C. shifan*, these species are quite similar to each other within the diversity of confuciusornithids, e.g., in the curvature of the ventral margin of the rostral end of the dentary and length ratio between the elements of fore- and hind-limbs. We think it is appropriate to provisionally place *C. shifan* in *Confuciusornis*, as explained in our responses to this reviewer's main review, but we have followed the reviewer's suggestion to discuss our phylogenetic results more thoroughly.

Please give approximate values of angles and ratios as in Wang and Zhou 2019 (current ref 6). If this is impossible given the journal's word limits then provide them in the supplement.

We have added the relevant values.

Are you confident this is the original margin? From the photograph it looks like it could be a break in a straighter margin.

Are there any other differences you can find?

Given the odd state of the dentary (last comment), the variation in alular claw size in *C. sanctus* (e.g. LPM-0229, p. 30 in Chiappe and Meng 2016), ventral process of surangular absent (previously autapomorphy of *C. dui* in Wang et al 2019), apparent transverse compression of proximal metatarsal III (proposed synapomorphy of *C. shifan*) in *C. dui* from Hou et al 1999's fig. 2d drawing, and generally poor state of published photos of the lost *C. dui* holotype I think it is important to make extra clear what separates this specimen from *C. dui*.

The state of the dentary is not really odd in our estimation, and we are confident that the rostral end of the dentary is exposed in ventrolateral view and is unbroken. The mandibular symphysis is slightly expanded in the ventral direction as in *C. sanctus*.

Based on Hou et al 1999's fig. 2d drawing, together with photos of the *C. dui* holotype (and casts thereof) in two Chinese books and several papers, we think the metatarsals of *C. dui* are evidently similar in morphology to those of *C. sanctus*, i.e., metatarsal III does not change in transverse width along its length, and metatarsal IV does not bear a lateral flange. In *C. shifan*, by contrast, the proximal end of metatarsal III is compressed and a lateral flange is present on metatarsal IV, clearly separating *C. shifan* from *C. dui*. Even so, following the reviewer's suggestion, we have noted additional features that distinguish these two species.

In the Description and Comparisons the dorsal process of the maxilla is said to differ, though I cannot tell where it is supposed to be in Figure 3. The dorsal process seems to be broken off with indistinct shards in the area it should be.

Although the left dorsal process is partly broken, the outline of the whole structure is clear based on the preserved portion and the impression of the missing one. We have added a line drawing with a dotted line to indicate the outline of the process in Fig.3c.

Should cite previous work calculating their body mass as well, e.g. current citation 15 and Miller and Pittman 2021

Miller, C.V. and Pittman, M., 2021. The diet of early birds based on modern and fossil evidence and a new framework for its reconstruction. *Biological Reviews*, 96(5), pp.2058-2112.

We have added citations to both references.

I cannot see this in Fig 3, suggest an inset and/or line drawing.

We have added a line drawing in Fig. 3c to indicate the outline of the dorsal process.

In how many confuciusornithids is the splenial known? I believe I have only seen it in *C. sanctus* and *Yangavis*.

Of the thoracic series or of the entire vertebral column?

I believe this should be in 'single quotes' for this journal.

confuciusornithid? confusiornithiform? Not clear what clade this refers to, as I do not believe there is a Confuciusornithinae defined.

? As in ornithothoracine birds? Extant birds? Later-diverging confuciusornithids?

Of the pectoral girdle? If so state explicitly.

We have made the requested changes and clarifications.

If the information is known, this could serve as an interesting point of comparison to other confuciusornithids (in addition to the metacarpal comments currently in the Discussion). Differences in articulations and muscle

attachment sites in the wing could support the paper's point regarding diverse flight styles in the family. Same for radius below.

Unfortunately, little information is available regarding articular surfaces and muscle attachment sites on the ulna and radius among confuciusornithids.

Is this only seen there, or is it in other avialans as well?

Vague phrasing. Discussion states explicitly which do have a similar process, that detail should be here and the discussion more general.

We have dropped the comparative statement with regard to contact between the proximal ends of the metacarpals, and added more detail with respect to the processes of the carpometacarpus.

Fascinating! May be worth mentioning the work of Mayr (2017) in the Discussion regarding this.

Mayr, G., 2017. Evolution of avian breeding strategies and its relation to the habitat preferences of Mesozoic birds. *Evolutionary Ecology*, 31(1), pp.131-141.

Green comments are aspects I believe would make the paper more impactful to discuss briefly (like the current section on tail vertebrae), but I understand may be beyond the scope of this work.

As implied by the referee, getting into the biological implications of the pelvic morphology is beyond the scope of our paper. We are planning a separate, detailed study of the pelvic girdle, as we feel that fusion between the pubes may be absent or minimal. Assuming we can confirm that this is the case, we shall discuss this feature in a subsequent paper.

I've seen a similar process on some extant raptorial bird tarsometatarsi, may be worth looking into.

We have not been able to find a similar process on any extant raptorial bird tarsometatarsi.

Per my Diagnosis comment, I think this topology requires further comment (e.g. what apomorphies drive the topology and make *Confuciusornis* paraphyletic) if the new taxon is kept as a species of *Confuciusornis*. Said discussion would be useful regardless of the change.

We have expanded our discussion of our phylogenetic results.

But your histological data finds the specimen to be highly mature, so wouldn't that mean it is at a later developmental stage than most confuciusornithids with complete pygostyle fusion? Could this be an example of paedomorphosis?

Our histological data indicate the specimen to be highly mature, but also suggest a slower skeletal growth rate than in *C. sanctus*, which may explain why the pygostyle remains unfused even at a late ontogenetic stage.

Is this a new hypothesis, or has this or alternative explanations been made for tail shortening before? Should signpost more clearly if the former, cite relevant works if the latter.

Having looked further into the issue of pygostyle evolution, we describe a more nuanced and complex pattern in the revised version of the manuscript.

Would be interesting to compare to past work on tail's aerodynamic function in early birds, e.g.:

O'Connor, J.K., Wang, X., Zheng, X., Hu, H., Zhang, X. and Zhou, Z., 2016. An enantiornithine with a fan-shaped tail, and the evolution of the rectricial complex in early birds. *Current Biology*, 26(1), pp.114-119.

Wang, M., O'Connor, J.K., Zhao, T., Pan, Y., Zheng, X., Wang, X. and Zhou, Z., 2021. An Early Cretaceous enantiornithine bird with a pintail. *Current Biology*, 31(21), pp.4845-4852.

We have added the requested comparisons.

Some quantitative analyses have also recovered them as generally strong fliers, and should be mentioned. Off the top of my head:

Dececchi, T.A. and Larsson, H.C., 2011. Assessing arboreal adaptations of bird antecedents: testing the ecological setting of the origin of the avian flight stroke. *PloS one*, 6(8), p.e22292.

Pei, R., Pittman, M., Goloboff, P.A., Dececchi, T.A., Habib, M.B., Kaye, T.G., Larsson, H.C., Norell, M.A., Brusatte, S.L. and Xu, X., 2020. Potential for powered flight neared by most close avialan relatives, but few crossed its thresholds. *Current Biology*, 30(20), pp.4033-4046.

We have added citations to the suggested references.

You also mention that *C. shifan* is unusually small in the Description section. Miller and Pittman 2021 (page 13) and Torres et al 2021 both note a reduction in body mass at the Ornithothoraces node, which would make sense as another flight adaptation *C. shifan* may have converged upon.

Miller, C.V. and Pittman, M., 2021. The diet of early birds based on modern and fossil evidence and a new framework for its reconstruction. *Biological Reviews*, 96(5), pp.2058-2112.

Torres, C.R., Norell, M.A. and Clarke, J.A., 2021. Bird neurocranial and body mass evolution across the end-Cretaceous mass extinction: The avian brain shape left other dinosaurs behind. *Science Advances*, 7(31), p.eabg7099.

We have added some discussion of whether size has played an important role in *confuciusornithid* flight evolution, and we conclude that there is no clear trend.

Serrano's multivariate mass equations have a ~20% confidence interval, how does this affect the results?

We now clearly acknowledge the uncertainty in the multivariate equations, in terms of mean percentage error ($|\%MPE|$, which does indeed approach $\pm 20\%$).

Pittman et al. 2020 would be good citation here

Pittman, M., Heers, A.M., Serrano, F.J., Field, D.J., Habib, M.B., Dececchi, T.A., Kaye, T.G. and Larsson, B.C., 2020. Methods of studying early theropod flight.

We have added citations to the suggested references.

Add citations; the phylogeny in this paper is noted as poorly-supported, but it seems the respective early- and late-divergence of these taxa are consistent in most recent studies.

The consistency with previous studies is now noted in our discussion of our phylogenetic results.

Presumably because birds in densely vegetated environments tend to have lower AR and WL?

Citation needed if so. The trend in Fig. 9 to me looks like the later-diverging *confuciusornithids* have more flap-gliding affinity, and I tend to associate more

gliding with more open areas (e.g. sea birds, vultures, migratory passerines, etc.)

We've removed the reference to densely vegetated environments, so this is now a moot point.

From here to line 496 seems more like Introduction than Discussion. I believe the information is necessary, but it breaks the flow of the paper by not addressing the fossil for an extended period.

This text is indeed background information, but its relevance would be unclear if it were moved to the Introduction. Even though it does slightly break the flow of the paper, as noted by the reviewer, we believe that it is more appropriately placed here than in the Introduction.

This study specifically points out SOCs' role in the digit I of bats, I presume serving a similar role to the alular digit in birds. Should be mentioned.

The revised version of the paper explicitly notes that the cited reference is about bats.

Again, citation needed discussing flight maneuverability in closed vs open environments.

The question of closed vs open environments has now been dropped from the paper, in favour of simply discussing different flight styles.

The logic of this is a little unclear to me. The earlier Discussion notes that SOCs may or may not end up fusing with the POCs. I suggest deleting as this seems unnecessarily speculative.

Citation error or is this part of the version number?

The table caption is very thorough, excellent work!
I will say it is a bit dense to read, if the journal allows I suggest adding line breaks or bullet points. Maybe even an additional table of measurements and specimens with the source at the intersection.

From the caption it looks like L_{prim} was estimated for most taxa; as with body mass, how does the uncertainty affect the AR and WL values?

Same problem of year and superscript is in Table S2 caption.

We have removed the sentence here about the epiphyseal structure fusing with the rest of the alular metacarpal, fixed the citation errors (good catch on the reviewer's part!), reformatted the text above Table S2, and more fully explained our method for estimating L_{prim} (and the relatively low uncertainty thereof).

Also the supplement does not have a works cited section for the references it cites.

We have added a references section.

This area is discussed minimally in the paper, why is it emphasized here?

Although the paper doesn't devote a lot of space to the anterior margin of the orbit, some important structures are located here, so in our view the close-up is necessary.

Please label the keel. It is a proposed autapomorphy and I am not sure what you are referring to.

The keel is the longitudinal ridge extending along the synsacrum, and we think this should be clear enough. We have added a label to indicate it.

Figure 7. Why the strange order? Makes more sense to me to remove tarsometatarsus mention from first sentence, and have a second sentence "Also pictured is the right tarsometatarsus (d) of *Confuciusornis shifan* holotype (PMoL-AB00178)"

We have reordered the content of this caption.

Should I be able to see a III? If so I cannot.

We have labelled metatarsal III.

Figure 9. Graph currently requires a lot of looking around to understand, would make sense to make fossil points different shapes or colors based on species.

Also the current red and green are indistinct to colorblindness, lightening or darkening should fix (lots of colorblind-friendly palettes online)

We have revised the colour scheme in the graph, and changed the labelling of individual points so that all specimens that are not *C. sanctus* are labelled.

Do we have any ontogenetic data on the *C. sanctus* specimens? I wonder if their flight strategy may have changed through life, especially given the recent proposition of niche partitioning:

Marugán-Lobón, J. and Chiappe, L. M., 2022. Ontogenetic niche shifts in the Mesozoic bird *Confuciusornis sanctus*. *Current Biology*.

Unfortunately, we have any ontogenetic data on the *C. sanctus* specimens. Whether or not their flight strategy may have changed through life is a very good issue to next step research.

A new confuciusornithid bird with a secondary epiphyseal ossification reveals
phylogenetic changes in confuciusornithid flight mode

Renfei Wang^{1,2}, Dongyu Hu^{2*}, Meisheng Zhang¹, Shiyong Wang^{3,4}, Qi Zhao^{3,4}, Corwin
Sullivan^{5,6} & Xing Xu^{3,4*}

¹ College of Earth Sciences, Jilin University, Changchun 130061, China

² Shenyang Normal University, Paleontological Museum of Liaoning, Key Laboratory
for Evolution of Past Life in Northeast Asia, Liaoning Province, 253 North Huanghe
Street, Shenyang 110034, China

³ Key Laboratory of Vertebrate Evolution and Human Origins, Institute of Vertebrate
Paleontology and Paleoanthropology, Chinese Academy of Sciences, Beijing 100044,
China

⁴ Center for Excellence in Life and Palaeoenvironment, Chinese Academy of Sciences,
Beijing 100044, China

⁵ Department of Biological Sciences, University of Alberta, Edmonton, AB T6G 2E9,
Canada

⁶ Philip J. Currie Dinosaur Museum, Wembley, AB T0H 3S0, Canada

Correspondence and requests for materials should be addressed to D.H. (email:
hudongyu@synu.edu.cn) and X.X. (email: xuxing@ivpp.ac.cn)

The confuciusornithids are the earliest known beaked birds, and also constitute
the only species-rich and morphologically diverse clade of Early Cretaceous birds that
existed prior to the cladogenesis of two major avialan groups Enantiornithes and
Ornithuromorpha. Here, we report a new confuciusornithid species from the Lower
Cretaceous of western Liaoning, northeastern China. Compared to other
confuciusornithids and most contemporaneous enantiornithines and ornithuromorphs,
this new species and the recently reported *Yangavis confucii* show a higher level of
flight adaptation, but the wings of the two taxa differ from one another in many
respects. Aerodynamic analyses of the known confuciusornithids indicate
considerable variations of flight capability and style across the clade, and to a lesser
degree through ontogeny, and specifically suggest both a trend towards improved
flight capability and a change in flight strategy in confuciusornithid evolution. Most
significantly, this new confuciusornithid differs from other Mesozoic birds in having a
secondary epiphyseal ossification, located in the alular digit. This highly unusual
feature may reflect the functional demands of flight when skeletal growth was still
incomplete. The new find strikingly exemplifies the morphological, developmental
and functional diversity of the first beaked birds.

Introduction

Confuciusornithidae is a clade of Early Cretaceous pygostylian birds known from
the Jehol Biota of East Asia¹, representing the earliest known toothless, beaked birds.
Five genera and eleven species, recovered from the Dabeigou, Yixian and Jiufotang
formations (~135~120 Ma), have been described and assigned to this family, though
the validity of some species is questionable²⁻⁶. Confuciusornithids are the only
species-rich, morphologically diverse avialan clade known to have existed prior to the
cladogenesis of the major groups Enantiornithes and Ornithuromorpha, and are
represented by thousands of exceptionally well preserved specimens that collectively
provide rich information on confuciusornithid morphology, taxonomy, flight ability,
growth, diet and ecology^{3,5,7-13}. Here, we report a new confuciusornithid species,
*Confuciusornis shifan* sp. nov., from the Jiufotang Formation. *C.* *shifan* differs from

other confuciusornithids in a number of morphological and developmental features,
which have implications for understanding confuciusornithid taxic diversity,
morphological disparity, development, and flight behavior.

**Results**

**Systematic paleontology.**

Aves Linnaeus, 1758

Pygostylia Chiappe, 2002

Confuciusornithidae Hou et al., 1995

*Confuciusornis* Hou et al., 1995

*Confuciusornis shifan* sp. nov.

[revised manuscript text omitted]

The left and right dentaries are almost completely fused with each other along an
extended, ventrocaudally inclined mandibular symphysis as in *C. sanctus*³ (Fig. 3a,b).
The transversely compressed ventrocaudal process of the right dentary is visible, but
the caudalmost end of the process is missing. In the intact mandible, the ventrocaudal
process likely contacts the angular laterally, and extends back to the level of the
caudal margin of the anterior mandibular fenestra based on the position of the
impression of the missing end. The right angular and right surangular both appear
rod-shaped, and their caudal ends fuse into the articular. The surangular appears to
lack a ventral process as in *C. dui*, indicating that the presence of a triangular ventral

surangular process in *C. sanctus* is indeed a diagnostic feature of this species⁵ (Fig.
3b). The articular prominently projects medially to form a medial articular facet for
the quadrate¹⁶. In contrast to the displaced and partly overlaped left splenial, the right
splenial is nearly complete and preserved close to its original position. The splenial is
plate-like and rostrally forked, and protrudes into the rostral half of the rostral
mandibular fenestra. The dorsal margin of the splenial overlaps the medial surface of
the surangular, and the ventral margin, which does not contact the angular, is
thickened along most of its length and gently tapers caudally to contact the ventral
process of the surangular. Unlike in other confuciusornithids, the splenial of *C. shifan*
is centrally perforated by an oval foramen (Fig. 3a,b).

The cervical vertebrae are exposed in ventrolateral view, ~~making it possible to see~~
that the cranial and caudal articular surfaces of each centrum are distinctly
heterocoelous (Fig. 4a) as previously reported for *Confuciusornis*⁵. The centra of the
thoracic series are laterally excavated as in *C. sanctus*, whereas such excavations are
absent in *Y. confucii*⁶ (Fig. 1). ~~The two cranialmost centra each bear~~ a prominent,
tapered ventral keel. The sacral vertebrae are completely fused with each other to
form a synsacrum (Fig. 4b). Based on the number of visible transverse processes, the
synsacrum consists of seven vertebrae as in other confuciusornithids³⁻⁶, and in ventral
view the total centra are transversely compressed to form a longitudinal ridge as in
some enantiornithines¹⁷ (Fig. 4b). However, the synsacrum of *C. sanctus* DNHM
D2454 (holotype of the putative species *Confuciusornis feducciai*) appears
dorsoventrally compressed and transversely widened⁵. Five free caudal vertebrae are
visible between the synsacrum and the pygostyle (Fig. 1), whereas additional caudal
vertebrae are likely overlaid by the left ischium and some soft tissue. The caudal
vertebrae appear significantly shorter than the thoracic ones. The pygostyle of *C.*
*shifan* appears to be exposed in left lateral view (Fig. 4c). The pygostyle is
proportionately robust and subequal in length to the tarsometatarsus, as is generally
the case in confuciusornithids^{3,18}. The transversely thickened ventral margin of the
pygostyle is distinctly made up of the co-ossified centra, which are thinned caudally.
The ventral surface does not form a longitudinal keel as in *C. sanctus* IVPP V 12352

(holotype of the putative species *Jinzhouornis zhangjiyingia*)^{5,18}. **Some 11 small round**
**foramina form a longitudinal row along the whole lateral furrow of the pygostyle (Fig.**
**4c).** The foramina are closer to the ventral margin of the pygostyle than the dorsal
margin, and gradually become smaller and more tightly crowded together towards the
caudal end of the series. Similar foramina occur in other confuciusornithine
specimens, but are fewer in number and only present on the cranial part of the
pygostyle^{18,19}.

The coracoids are fused with the scapulae to form scapulocoracoids. The left and
right scapulocoracoids are exposed in lateral and medial views, respectively (Fig. 5a).
The long axes of the scapula and coracoid define an acute angle as in *Y. confucii*⁶ and
more advanced birds (Fig. 5a, 6). This angle is approximately 90° in *C. sanctus*³. A
laterally directed glenoid facet is visible in the left scapulocoracoid. The cranial and
caudal margins of the glenoid facet project more strongly than the dorsal and ventral
ones, implying that the configuration of the facet probably did not strongly limit the
elevation and depression of the wing in flapping flight. The acromion of the scapula is
well-developed. The shaft of the scapula curves slightly downward; the cranial half of
the shaft appears rod-like and slightly compressed dorsoventrally, and while the
caudal half is more compressed mediolaterally and widens significantly in the
dorsoventral direction before tapering to a point as in more advanced birds²⁰ (Fig. 5a,
6). In other confuciusornithids, the scapula is generally straight, and neither flares
partway along its length nor tapers caudally³. The coracoid is strut-like and
approximately half the length of the scapula. Its proximal end appears dorsoventrally
thicker than transversely wide and projects slightly cranially to form an acrocoracoid
process as in *Y. confucii*⁶, whereas its distal end is compressed dorsoventrally. The
acrocoracoid process is not well-developed in *C. sanctus*³. The lateral surface of the
coracoid is excavated to form a shallow groove, which gradually becomes narrower
and shallower distally as in *Y. confucii*⁶. The medial margin of the coracoid is strongly
compressed to form a longitudinal keel, except at the proximal end of the bone. The
furcula is mostly exposed in caudal view. As in other confuciusornithids, the furcula is
boomerang-shaped and without a distinct tubercle³⁻⁶ (Fig. 5a, 6). **The proximomedial**

corner of the right clavicular ramus is inflated caudally to form a bulbous structure
that contacts the medial face of the proximal end of the right scapulocoracoid, and
meanwhile the proximolateral corner is more craniocaudally compressed than the rest
of the ramus and slightly projects laterally, to form a distinct impression on the caudal
surface of the corner. Similar structure is not seen in other confuciusornithids.

The forelimb is subequal in length to the hindlimb, with a forelimb (humerus +
ulna) / hindlimb (femur + tibiotarsus) ratio of 1.02 (Fig. 1; Supplementary table S1).
The equivalent ratio is 0.93 in *C. hengdaoziensis*, 0.96 in *E. zhengi*, 1.06 in *C. dui*,
0.91~1.08 in *C. sanctus* and 1.19 in *Y. confucii*⁶ (Fig. 6). The humerus is typical of
confuciusornithids in having an expanded and perforated deltopectoral crest. However,
the crest is less prominent overall than those of other confuciusornithids except *C. dui*⁵,
and the point of the greatest prominence is at the distal end of the crest rather than in the
middle as in other confuciusornithids³⁻⁶ (Fig. 5a, 6). The dorsodistal corner of the crest
projects distally, so that the distal margin appears concave. Distally, a large fossa for the
brachialis muscle is present proximal to the dorsal and ventral condyles. The dorsal
epicondyle is better developed than the ventral epicondyle, and the former projects
sharply dorsally. Both the ulna and the radius are shorter than the humerus, as in other
confuciusornithids (Fig. 1, 6; Supplementary table S1, 2). A concave ventral cotyla, a
flat dorsal cotyla and a concave incisura radialis are clearly visible on the proximal
end of the right ulna, whereas a distinctly projecting olecranon process is lacking. The
proximal humeral articular facet of the radius is flat and the bicapital tubercle is
situated on the cranial face of the proximal end of the radius. Both radialia are in their
original positions relative to the radii and carpometacarpi (Fig. 1). The left radiale are
exposed in palmar and obliquely proximal view and the right in palmar and obliquely
distal view. The articular facets for the radius and semilunate carpal are both
significantly concave. The cranial surface of the radiale is convex and forms a small
tubercle on the proximal side, and the caudal surface is slightly concave and directed
somewhat proximally to articulate with the ulna.

The semilunate carpal is completely fused with metacarpals II and III to form a
carpometacarpus (Fig. 1, 7a,b). Metacarpal III is much more slender, and slightly

shorter, than metacarpal II, and contact between these metacarpals is limited to their
proximal ends as in enantiornithines²¹. The middle parts of metacarpals II and III are
constricted, and the narrowest part of metacarpal III is less than half the width of the
narrowest part of metacarpal II as in other confuciusornithids. However, the
intermetacarpal space appears relatively longer and significantly wider than in other
confuciusornithids, as in some advanced birds²⁰. The proximal end of the
carpometacarpus forms a pulley-like carpal trochlea as in *C. sanctus*³ and extant birds.
A distinct fossa is present in the region where the semilunate carpal is fused to
metacarpals II and III. The bump-like pisiform process is located near the proximal
end of metacarpal II. The alular metacarpal is about one third the length of metacarpal
II as in *C. sanctus*³ and *E. zhengi*⁴, rather than half the length as in *Ch.*
*hengdaoziensis*³ (Fig. 6, 7a,b). It is not fused with the carpometacarpus (Fig. 7a,b).
The caudal portion of the alular metacarpal extends farther proximally than the cranial
portion so that the proximal surface of the metacarpal is concave, forming a cranial
carpal fovea. The proximal two thirds of the cranial margin projects cranially to form
the extensor process. The craniocaudal width of the process is nearly half that of the
distal articular surface of the alular metacarpal (Fig. 7a,b,c). Such a well-developed
extensor process is not seen in contemporaneous basal birds, or in most
enantiornithines and ornithuromorphs^{20,21}. Unlike in previously known
confuciusornithids, the caudal distal condyle of the alular metacarpal is
well-developed, and projected more distoventrally than the rest of the metacarpal,
evident in palmar view. The proximal phalanx of digit II is robust, maintains a
constant diameter throughout its entire length, and is slightly shorter than the
intermediate phalanx as in other confuciusornithids except *Ch. hengdaoziensis*³(Fig.
1). In *Ch. hengdaoziensis*, the proximal phalanx is slightly longer than the
intermediate one³. The ungual phalanges of the alular digit and digit III are large and
curved, whereas that of digit II is small as in most confuciusornithids. In *Y. confucii*,
however, the ungual phalanx of digit II is large⁶.

Among the fully articulated left and right pelvic girdles, only the pubes are
relatively completely preserved (Fig. 5b). The right ilium appears relatively low, with

a slightly convex dorsal margin and a laterally projected antitrochanter. The pubis,
remarkably, is longer than the femur, being rod-like, slender, and strongly retroverted.
Because of a sharp bend in the pubic shaft, the long axis of the pubic symphysis is
nearly perpendicular to that of the shaft's proximal portion, a condition resembling
that seen in derived ornithuromorph birds.

The femoral head, neck and trochanteric crest are well-developed. The proximal
tarsals are completely fused with each other and with the tibia, forming a tibiotarsus.
The proximal ends of metatarsals II-IV are fused with each other and with the distal
tarsals, forming a tarsometatarsus. The proximal end of metatarsal III appears
transversely compressed between metatarsals II and IV in cranial view (Fig. 7d),
whereas this metatarsal in other confuciusornithids does not change in width along its
length³⁻⁶. An oval tubercle presumably for insertion of M. tibialis cranialis is located
on the dorsal surface near the proximal end as in *C. sanctus*³, but similar structure
does not exist on metatarsal II. A ridge-like process is present on the lateral margin of
the distal third of metatarsal IV (Fig. 7d), but it is not seen in other confuciusornithids.
In pedal digits III-IV, the penultimate phalanx is longer than the preceding phalanx, as
in other confuciusornithids⁶. However, the penultimate phalanx of pedal digit II is
subequal in length to the preceding phalanx.

*Secondary epiphyseal ossification* The most unusual skeletal feature exhibited by
PMoL-AB00178 is the presence of a cushion-like small bone between the distal end
of the alular metacarpal and the proximal end of the first alular phalanx (Fig. 7a,b). In
the left manus, the small bone contacts the cranial portions of the opposing articular
surfaces of the metacarpal and phalanx (Fig. 7a); in the right manus, the small bone
again contacts the cranial portion of the proximal articular surface of the first alular
phalanx, but rests against the cranial margin of the left alular metacarpal due to the
slight disarticulation of the alular digit (Fig. 7b). We identify this small bone as an
ossification that arose from a secondary ossification center within the distal epiphysis
of the alular metacarpal and is equivalent to the medial condyle seen on the
metacarpal's distal articular surface in other confuciusornithids (Fig. 7c), based on the

following evidence: (1) in the right manus, the small bone remains in apparently
natural alignment with the alular metacarpal, but not with the first alular phalanx; (2)
in other confuciusornithids the distal articular surface of the alular metacarpal is
ginglymoid⁶, and the caudal condyle protrudes slightly more distally than the cranial
one (Fig. 7c), whereas in both alular metacarpals of PMoL-AB00178 the caudal
condyle is distally prominent and cranial condyle is absent; and (3) in both first alular
phalanges of PMoL-AB00178, the proximal articular surface is similar in morphology
to those of other confuciusornithids, and a well-developed flexor tubercle is visible on
the left alular phalanx.

Discussion

Our phylogenetic analysis places *C. shifan* as the sister taxon to *C. sanctus*, and *E.*
*zhengi*, *C. dui*, *Y. confucii* and *Ch. hengdaoziensis* as successive outgroups to the *C.*
*sanctus* + *C. shifan* clade. However, the various confuciusornithid subclades involved
in this arrangement are not well supported, as indicated by their low Bremer and
Bootstrap values (Fig. 8).

The discovery of *C. shifan* sheds new light on morphological evolution and flight
adaptation in confuciusornithid birds. Among early pygostylians, both
confuciusornithid and enantiornithines have relatively large pygostyles¹⁸, although the
exact number of caudal vertebrae involved in the pygostyle is typically difficult to
determine due to their complete coossification. Perforated pygostyles, such as that of
PMoL-AB00178, are thought to be the result of incomplete fusion between adjacent
neural arches, and the foramina are finally enclosed^{18,19}. The pygostyle of
PMoL-AB00178 then presumably exemplifies the developmental stage at which
fusion among the neural arches was partial rather than complete. Given the presence
of 11 preserved foramina, the number of caudal vertebrae within the pygostyle is
estimated to be 12. If the estimate for the pygostyle is accurate and PMoL-AB00178
has seven free caudal vertebrae as in *C. sanctus*³, then 19 caudal vertebrae are present
in the tail as a whole (Fig. 1, 4c), a total only slightly less than the count of 21-23 free
caudal vertebrae reported for the long bony tail of *Archaeopteryx*²². This comparison

implies that the transition in birds from a long bony tail like that plesiomorphically
present in reptiles to a short one ending in a pygostyle was driven primarily by
shortening of the individual caudal vertebrae, rather than by a reduction in vertebral
count. The fact that the pygostyle is the only compound bone in PMoL-AB00178 to
show incomplete fusion also indicates that the tail may not have played a major role in
generating aerodynamic forces during flight.

The flight capability of confuciusornithids has been debated in previous studies.
Some authors have considered confuciusornithids competent flyers based on a number
of morphological features, including long wings with strongly asymmetrical flight
feathers, strut-like coracoids, a keeled sternum, and enlarged major manual digits²³,
whereas others have suggested more limited flight capabilities¹² based on functional
analysis of the morphology of the shoulder joint⁷ or the flight feathers⁹. Nevertheless,
there has been broad agreement that confuciusornithids display a more primitive suite
of flight-related characters than contemporaneous enantiornithines and
ornithuromorphs^{1,7}, suggesting inferior flight performance. However, morphological
evidence from both *C. shifan* and the recently discovered *Y. confucii* indicates that
some variation in flight capability existed within Confuciusornithidae, as both these
species appear to have been unusually well-adapted for flight, albeit in different ways.
*Y. confucii* and *C. shifan* both exhibit a more specialized flight apparatus than is
present in other confuciusornithids, or even in most contemporaneous enantiornithines
and ornithuromorphs. In *Y. confucii*, the main modification is elongation of the
forelimb to increase the wing area as in sapeornithids²⁴ and some enantiornithine and
ornithuromorph birds. In *C. shifan* the main modifications involve refinements to the
pectoral girdle, humerus and carpometacarpus, which to some extent resemble their
counterparts in enantiornithine and ornithuromorph birds. In particular, some skeletal
processes to which flight muscles would have attached are particularly well developed.
A few such features, including the extensor process of the alular metacarpal and the
pisiform process of the carpometacarpus, are seen only in the contemporaneous
enantiornithine *Xiangornis*²¹, a few Early and Late Cretaceous ornithuromorphs²⁵, and
extant flying birds (Fig. 7a,b). In extant birds, the extensor process is the point of

insertion for *M. extensor metacarpi radialis*, the primary muscle involved in extension
of the wrist joint, and the pisiform process serves for attachment of the retinaculum
flexorium, and as a pulley changing the direction of the tendon of *M. flexor digitorum*
*profundus*²⁶, which crosses the wrist and inserts on the distal phalanx of the major
digit. A well-developed extensor process and pisiform process can accordingly
increase manipulating and controlling capabilities of the distal part of the wing during
flight²⁷, collectively suggesting greater flight capability in *C. shifan* than in other
confuciusornithids.

A recent study showed that most flight modes seen in extant birds (e.g.,
continuous flapping, flap-gliding, flap-bounding, thermal soaring) could ~~potentially~~
have been represented among early birds²⁸. The flight parameter estimates obtained
for confuciusornithids in that analysis would suggest that they were continuous
flappers, but only the holotype of *E. zhengi* and a few specimens of *C. sanctus* were
considered. In order to further assess the flight capabilities and strategies of
confuciusornithid birds, we followed the methods of Serrano et al. (2017)²⁸ (also see
Aerodynamic analysis in Methods below) to estimate flight parameters for 11
confuciusornithid specimens, including the only known specimens of *Y. confucii* and
*C. shifan* (Supplementary table S2, 3), and plotted estimates of two key parameters for
these confuciusornithids, namely aspect ratio (AR) and wing loading (WL), on a
previously published figure showing the regions of AR-WL morphospace associated
with the main flight modes in extant birds²⁹. ~~AR~~ is the ratio of wing length to wing
width, representing a simple measure of wing shape, and WL is the ratio of body mass
to the total area of the two wings and the part of the trunk between them, representing
the load that must be borne by each unit area of the available lift-producing surface.
Both metrics are commonly used in assessing the flight capabilities and strategies of
flying animals. Our analyses show that both AR and WL vary considerably among
confuciusornithid specimens and even among specimens of *C. sanctus* (Supplemental
table S3; Fig. 9), as found ~~by previous researchers~~²⁸. Though two specimens of *C.*
*sanctus* fall slightly into the area of morphospace where extant facultative flap-gliders
overlap with continuous flappers, most confuciusornithids plot at least narrowly

outside the flap-gliding area, and within the area occupied solely by continuous
flappers (Fig. 9). Interestingly, confuciusornithids are distributed throughout a large
part of the continuous flapping morphospace, and two ends represented by *E. zhengi*
and *Y. confucii* with the highest WL and AR, and *C. shifan* with the lowest WL and
AR, corresponding to two distinct flight strategies seen in the extant birds;
medium-to-large birds tend to have high AR and WL to increase lift efficiency and
flight speed, and minimize energy costs for long flights, whereas small birds and
particularly small forest-dwelling birds tend to have lower AR and WL to increase
maneuverability during shorter and slower flights³⁰⁻³². *C. shifan*, with its low values of
both AR and WL, lies well within continuous flapping morphospace but adjacent to
the areas of overlap with flap-gliders and flap-bounders.

Most importantly, our analyses under phylogeny may indicate an evolutionary
trend affecting confuciusornithid flight, involving a tendency for WL and AR to
decrease from early-diverging confuciusornithids such as *E. zhengi* and *Y. confucii* to
late-diverging ones such as *C. sanctus* and *C. shifan*. This suggests that the evolution
of improved flight ability in confuciusornithids was likely associated with an
ecological shift to a more densely vegetated environment. *C. shifan* represents the
culmination of this evolutionary trend, at least among currently known
confuciusornithids, which is reflected in the proximity of this species to flap-bounding
morphospace in Figure 9. *Y. confucii* may independently become more adapted for
open environment or long flight. The modified nature of their wings and shoulder
girdles further support this inference.
The discovery of the epiphyseal SOC in *C. shifan* was a truly unexpected finding
of this study. The long bones of tetrapods each fundamentally comprise a central shaft
(diaphysis) with expanded articular ends (epiphyses), which are separated from the
diaphysis by short intermediate zones known as metaphyses³³. In extant tetrapods, the
long bones are initially cartilaginous and endochondral ossification begins within
the diaphysis, at what is termed the primary ossification center (POC). In most
tetrapod groups, ossification spreads gradually from the POC throughout the diaphysis
and metaphyses, and potentially into the epiphyses, leaving only a relatively thin cap

of hyaline cartilage over the endochondral bone. In mammals and most lizards,
however, secondary ossification centers (SOCs) appear within the epiphyses, and are
separated from the metaphyses by growth plates³³⁻³⁵. Each growth plate constantly
generates new hyaline cartilage between itself and the metaphysis, so that the skeletal
element grows longitudinally. Meanwhile, cartilage is gradually converted to bone
within the epiphyses, as well as in the proximal and distal ends of the diaphysis. As
skeletal growth concludes, the growth plate becomes thin and finally disappears,
causing complete fusion of the epiphyses to the metaphyses and, in turn, an end to
longitudinal growth³³. The timing of complete fusion varies among species, and
among the different long bones of a single individual^{33,36}. Additional ossification
centers may appear within apophyses, which typically form protruding features that
serve as attachment sites for tendons and ligaments^{37,38}. Furthermore, ossification
centers may appear within the tendons or ligaments themselves, forming bones called
sesamoids^{39,40}. In subsequent ontogeny these may either remain embedded in the
tendons/ligaments with no direct connection to the rest of the skeleton, as in the case
of the mammalian patella, or fuse into adjacent long bones as apophyseal projections
(“traction epiphyses”)³⁷⁻⁴⁰

Presently, the only recognized true epiphyseal ossification in birds is one that
forms at the proximal end of the tibiotarsus in many taxa, and may be plesiomorphic
at least for crown-group birds^{41,42}. This epiphyseal ossification may have evolved
from a pre-existing sesamoid as a “traction epiphysis”^{38,43,44}, or may represent a true
secondary epiphyseal ossification comparable to those present in mammals and most
lizards⁴⁵. Furthermore, the tarsal elements and some carpal elements of extant birds
fuse to adjacent long bones in a manner superficially resembling the fusion of
epiphyseal ossifications to metaphyses, and apophysis-like ossifications are also
known in some species^{42,44}.

Epiphyseal SOC's have been demonstrated to provide additional stiffness to
articular portions of bones that require reinforcement, for example as a result of
stresses associated with support and locomotion in the terrestrial environment, and to
facilitate the formation of topographically complex joint surfaces^{35,46,47}. To our

knowledge, however, epiphyseal SOC's do not occur in non-avian archosauriforms,
including in large, undoubtedly terrestrial dinosaurs that would have experienced
substantial locomotor stresses. Epiphyseal SOC's probably evolved independently in
mammals, birds and lepidosaurs³⁹, for reasons potentially relating to patterns of
skeletal growth as well as to mechanical factors. Furthermore, we are unaware of any
previous report of an epiphyseal SOC in a non-ornithuromorph bird. The proximal
tibial center appears to have been present in hesperornithiforms based on differences
between adult and juvenile tibiotarsi of the Early Cretaceous genus *Enaliornis*⁴⁸, but it
is surprising that the holotype of *C. shifan* possesses an epiphyseal SOC anywhere in
the skeleton, let alone at a location – the cranial portion of the distal end of the alular
metacarpal – where no such feature occurs even in crown-group birds.

The SOC in the distal epiphysis of the alular metacarpal of *C. shifan* (Fig. 7a,b) is
presumably an autapomorphic feature, whose appearance may be explicable on the
basis of a combination of growth strategy and functional demands on the wing. A
recent study has demonstrated that epiphyseal SOC's tend to evolve as a response to
high mechanical stress⁴⁷, and in *C. shifan* the distal end of the alular metacarpal may
well have begun to experience considerable stresses as soon as a developing juvenile
began to fly. In extant birds the alular metacarpal plays an important role in aerial
maneuvers, and the wings of *C. shifan* appear better-adapted than those of other
confuciusornithids for high-maneuverability flight in relatively closed environments.
As a result, the distal end of the alular metacarpal might have been more frequently
subject to large stresses in *C. shifan* than in most confuciusornithids. Furthermore, the
histological features of *C. shifan* suggest a slower growth rate than that of other
confuciusornithids, based on Amprino's rule⁴⁹. For example, *C. shifan* has
longitudinally oriented vascular canals in the middle layer of the femoral compact
bone (Fig. 2), whereas concentrically oriented canals are more numerous in other
confuciusornithids¹³. If the spread of ossification from the primary centers of the long
bones into the epiphyses was also slow, then the establishment of an epiphyseal SOC
might have been necessary to reinforce the distal end of the alular metacarpal at a
comparatively early ontogenetic stage, when the juvenile was beginning to fly but

skeletal growth was still incomplete. The holotype of *C. shifan*, however, clearly
represents a mature individual in which the caudal distal condyle of the alular
metacarpal had fully ossified by the time of death, presumably via the normal
mechanism of spread from the diaphyseal POC. We infer that the secondary
epiphyseal ossification would have fused with the rest of the alular metacarpal to
remodel a cranial distal condyle of more normal appearance, had the individual lived
longer.

**Methods**

**Histological preparation of *C. shifan* (PMoL-AB00178).** We took the bone sample
near the mid-diaphysis of the right femur of PMoL-AB00178 (Fig. 1a). Cross-section
was prepared using standard procedures⁵⁰. The sample was embedded in
one-component resin (EXAKT Technovit 7200), which was then hardened in a light
polymerization device (EXAKT 520). Thin cross-section was cut using an accurate
circular saw (EXAKT 300CP). The section was ground down using the EXAKT
400CS grinding system until the desired optical transparency was obtained. The
histological section was examined under a polarized light microscope (ZEISS Axio
Imager 2 Pol), and photographed with a ZEISS AxioCam 705 digital microscope
camera.

**Phylogenetic analysis.** To investigate the systematic position of *C. shifan* relative to
other Mesozoic birds, a phylogenetic analysis was carried out using the most
comprehensive data matrix currently available for Mesozoic birds⁵¹, with *C. shifan*
added in based on scorings from PMoL-AB00178. The revised matrix consists of 81
taxa and 280 characters. The matrix was analyzed using TNT v1.5 (22)⁵² with default
settings. All characters were equally weighted, with 35 characters ordered; the dataset
was analyzed using the “New Technology search” methods with sectorial search,
ratchet, tree drift, and tree fusion with default settings; the minimum-length tree was
found in 10 replicates to recover as many tree islands as possible. The recovered trees
were then used as the basis for a traditional TBR search. Zero-length branches were

collapsed. Decay indices (Bremer support values) were calculated using the Bremer
script embedded in TNT, and absolute Bootstrap frequencies were calculated using
1000 pseudoreplicates in TNT with default settings.

**Aerodynamic analysis.** In order to assess the flight capabilities and strategies of
confuciusornithid birds, we estimated values of several flight parameters for 11
confuciusornithid individuals, including the holotypes of *C. dui*, *Ch. hengdaoziensis*, *E.*
*zhengi*, *Y. confucii* and *C. shifan* (see Supplementary tables S2, 3). Specifically, we
estimated the body mass (BM), wingspan (B), and lift surface area (SL) of these
specimens following the methods of Seranno et al. (2015, 2017)^{15,28}, and subsequently
used these estimates as a basis for calculating aspect ratio (AR, $AR = B^2/SL$) and wing
loading (WL, $WL = BM/SL$) values (see Supplementary table S3). The anatomical
data needed for these estimates were obtained by measuring the specimens directly
with digital calipers, or alternatively from high-resolution images using tpsDig 2.17⁵³
(available at: <https://sbmorphometrics.org/>) from measurements published in the
literature, or from estimated measurements based on the methods of Serrano et
al.2018²⁹ (see Supplementary table S2 for details).

[revised manuscript text omitted]

Figure 1. Photograph (a) and line drawing (b) of *Confuciusornis shifan* holotype
 (PMoL-AB00178). Arrow indicates the sampling position for the histological section.

Abbreviations: cav, caudal vertebra; cev, cervical vertebra; fu, furcula; ga, gastralium; lc,
 left coracoid; ldIII, left manual digit III; lfe, left femur; lh, left humerus; lil, left ilium;
 lis, left ischium; lm, left manus; lp, left pes; lra, left radiale; lr, left radius; ls, left
 scapula; lti, left tibiotarsus; lu, left ulna; lul, left ulnare; pu, pubis; py, pygostyle; r, rib;
 rc, right coracoid; rdcIII, caw of right manual digit III; rfe, right femur; rfi, right fibula;
 rh, right humerus; ris, right ischium; rm, right manus; rp, right pes; rra, right radiale; rr,
 right radius; rs, right scapula; rti, right tibiotarsus; ru, right ulna; rul, right ulnare; sk,
 skull; sy, synsacrum; tv, thoracic vertebra. Scale bars: 2 cm.

Figure 2. Photograph of a midshaft histological section of the right femur of the
*Confuciusornis shifan* holotype (PMoL-AB00178). White arrows indicate LAGs.

Abbreviations: ICL, inner circumferential layer; OCL, outer circumferential layer.

Scale bar: 100µm.

Figure 3. Photograph of the skull and mandible of the *Confuciusornis shifan* holotype

(PMoL-AB00178) in left ventrolateral view (a) with corresponding line drawing (b),

a close-up of the anterior margin of the orbit (c). White rectangle in (a) indicates the

region of (c). A line drawing of the posterior half of the mandible of *Confuciusornis*

*sanctus* IVPP V 13171 in Wang et al., 2018⁵ is appended below the line drawing of

the skull and mandible of the *Confuciusornis shifan*, and the arrow indicates the

position of the ventral process of the surangular. Abbreviations: an, angular; ar,

articular; cmf, caudal mandibular fenestra; d, dentary; f, frontal; j, jugal; l, lacrimal; m,

maxilla; o, orbit; rmf, rostral mandibular fenestra; n, nasal; p, parietal; pm, premaxilla;

q, quadrate; sa, surangular; sp, splenial. Scale bars: 1 cm; note that (c) is not to scale.

Figure 4. Photographs of the cervical vertebrae (a), synsacrum (b) and pygostyle (c)
 of the *Confuciusornis shifan* holotype (PMoL-AB00178). Abbreviation: sy,
 synsacrum. Arrows in (c) indicate the positions of foramina along the pygostyle. Scale
 bars: 0.5 cm in (a) and (c); 1 cm in (b).

Figure 5. Photographs of the pectoral (a) and pelvic (b) girdle of the *Confuciusornis*
*shifan* holotype (PMoL-AB00178). Abbreviation: fu, furcula; lc, left coracoid; lil, left
ilium; lis, left ischium; ls, left scapula; pu, pubis; rc, right coracoid; rh, right humerus;
ris, right ischium; rs, right scapula. Scale bars: 1 cm.

Figure 6. Line drawings of the pectoral girdle, forelimb, hindlimb of the
 *Confuciusornis shifan* holotype (PMoL-AB00178), and other confuciusornithids, with
 shading to indicate lengths of stylopodial, zeugopodial and metapodial limb segments.
 Values near the segments represent ratios of segment length to femoral length. All
 drawings scaled to a common, arbitrary femoral length.

Figure 7. Photographs of the left (a) and right (b) carpometacarpi and right
 tarsometatarsus (d) of *Confuciusornis shifan* holotype (PMoL-AB00178), and the left
 alular metacarpal (c) of the confuciusornithid specimen PMoL-AB00150.
 Carpometacarpi in palmar view, tarsometatarsus in cranial view. Black and white
 arrows in (d) indicate the ridge-like process on metatarsal IV and the dorsal tubercle
 on metatarsal III, respectively. Abbreviations: cb, cushion-like bone; ep, extensor
 process; pp, pisiform process. Roman numerals in (d) identify metatarsals. Scale bars:
 0.25 cm.

Figure 8. Cladogram of Mesozoic birds showing the systematic position of
 *Confuciusornis shifan*, and representing the strict consensus of the 192 most
 parsimonious trees recovered in the phylogenetic analysis performed in this study
 (length = 1404; consistency index = 0.277; retention index = 0.667). Bootstrap and
 Bremer values are given in normal font and in bold italic font, respectively, near the
 nodes to which they pertain.

Figure 9. Positions of 11 confuciusornithid specimens in a previously published

morphospace defined by wing loading (WL) and aspect ratio (AR), showing the areas

of morphospace occupied by extant birds with particular modes of flight²⁹. Specimens

are indicated by black circles, and numbered in descending order of estimated AR.

Specimens are holotypes of their respective species unless a number is indicated.

A new confuciusornithid bird with a secondary epiphyseal ossification reveals
phylogenetic changes in confuciusornithid flight mode

Renfei Wang^{1,2}, Dongyu Hu^{2*}, Meisheng Zhang¹, Shiyang Wang^{3,4}, Qi Zhao^{3,4}, Corwin
Sullivan^{5,6} & Xing Xu^{3,4*}

¹ College of Earth Sciences, Jilin University, Changchun 130061, China

² Shenyang Normal University, Paleontological Museum of Liaoning, Key Laboratory
for Evolution of Past Life in Northeast Asia, Liaoning Province, 253 North Huanghe
Street, Shenyang 110034, China

³ Key Laboratory of Vertebrate Evolution and Human Origins, Institute of Vertebrate
Paleontology and Paleoanthropology, Chinese Academy of Sciences, Beijing 100044,
China

⁴ Center for Excellence in Life and Paleoenvironment, Chinese Academy of Sciences,
Beijing 100044, China

⁵ Department of Biological Sciences, University of Alberta, Edmonton, AB T6G 2E9,
Canada

⁶ Philip J. Currie Dinosaur Museum, Wembley, AB T0H 3S0, Canada

Correspondence and requests for materials should be addressed to D.H. (email:
hdongyu@synu.edu.cn) and X.X. (email: xuxing@ivpp.ac.cn)

The confuciusornithids are the earliest known beaked birds, and also constitute
the only species-rich and morphologically diverse clade of Early Cretaceous birds that
existed prior to the cladogenesis of two major avialan groups Enantiornithes and
Ornithuromorpha. Here we report a new confuciusornithid species from the Lower
Cretaceous of western Liaoning, northeastern China. Compared to other
confuciusornithids and most contemporaneous enantiornithines and ornithuromorphs,
this new species and the recently reported *Yangavis confucii* show a higher level of
flight adaptation, but the wings of the two taxa differ from one another in many
respects. Aerodynamic analyses of the known confuciusornithids indicate
considerable variations of flight capability and style across the clade, and to a lesser
degree through ontogeny, and specifically suggest both a trend towards improved
flight capability and a change in flight strategy in confuciusornithid evolution. Most
significantly, this new confuciusornithid differs from other Mesozoic birds in having a
secondary epiphyseal ossification, located in the alular digit. This highly unusual
feature may reflect the functional demands of flight when skeletal growth was still
incomplete. The new find strikingly exemplifies the morphological, developmental
and functional diversity of the first beaked birds.

Introduction

Confuciusornithidae is a clade of Early Cretaceous pygostylian birds known from
the Jehol Biota of East Asia¹, representing the earliest known toothless, beaked birds.
Five genera and eleven species, recovered from the Dabeigou, Yixian and Jiufotang
formations (~135~120 Ma), have been described and assigned to this family, though
the validity of some species is questionable²⁻⁶. Confuciusornithids are the only
species-rich, morphologically diverse avialan clade known to have existed prior to the
cladogenesis of the major groups Enantiornithes and Ornithuromorpha, and are
represented by thousands of exceptionally well preserved specimens that collectively
provide rich information on confuciusornithid morphology, taxonomy, flight ability,
growth, diet and ecology^{3,5,7-13}. Here, we report a new confuciusornithid species,
*Confuciusornis shifan* sp. nov., from the Jiufotang Formation. *C. shifan* differs from

other confuciusornithids in a number of morphological and developmental features,
which have implications for understanding confuciusornithid **taxic** diversity,
morphological disparity, development, and flight behavior.

**Results**

**Systematic paleontology.**

**Aves** Linnaeus, 1758

Pygostylia Chiappe, 2002

Confuciusornithidae Hou et al. 1995

*Confuciusornis* Hou et al., 1995

*Confuciusornis shifan* sp. nov.

[revised manuscript text omitted]

The left and right dentaries are almost completely fused with each other along an
extended, ventrocaudally inclined mandibular symphysis as in *C. sanctus*³ (Fig. 3a,b).
The transversely compressed ventrocaudal process of the right dentary is visible, but
the caudalmost end of the process is missing. In the intact mandible, the ventrocaudal
process likely contacts the angular laterally, and extends back to the level of the
caudal margin of the anterior mandibular fenestra based on the position of the
impression of the missing end. The right angular and right surangular both appear
rod-shaped, and their caudal ends fuse into the articular. The surangular appears to
lack a ventral process as in *C. dui*, indicating that the presence of a triangular ventral

surangular process in *C. sanctus* is indeed a diagnostic feature of this species⁵ (Fig.
3b). The articular prominently projects medially to form a medial articular facet for
the quadrate¹⁶. In contrast to the displaced and partly overlapped left splenial, the right
splenial is nearly complete and preserved close to its original position. The splenial is
plate-like and rostrally forked, and protrudes into the rostral half of the rostral
mandibular fenestra. The dorsal margin of the splenial overlaps the medial surface of
the surangular, and the ventral margin, which does not contact the angular, is
thickened along most of its length and gently tapers caudally to contact the ventral
process of the surangular. Unlike in other confuciusornithids, the splenial of *C. shifan*
is centrally perforated by an oval foramen (Fig. 3a,b).

The cervical vertebrae are exposed in ventrolateral view, making it possible to see
that the cranial and caudal articular surfaces of each centrum are distinctly
heterocoelous (Fig. 4a) as previously reported for *Confuciusornis*⁵. The centra of the
thoracic series are laterally excavated as in *C. sanctus*, whereas such excavations are
absent in *Y. confucii*⁶ (Fig. 1). The two cranialmost centra each bear a prominent,
tapered ventral keel. The sacral vertebrae are completely fused with each other to
form a synsacrum (Fig. 4b). Based on the number of visible transverse processes, the
synsacrum consists of seven vertebrae as in other confuciusornithids³⁻⁶, and in ventral
view the total centra are transversely compressed to form a longitudinal ridge as in
some enantiornithines¹⁷ (Fig. 4b). However, the synsacrum of *C. sanctus* DNHM
D2454 (holotype of the putative species *Confuciusornis feducciai*) appears
dorsoventrally compressed and transversely widened⁵. Five free caudal vertebrae are
visible between the synsacrum and the pygostyle (Fig. 1), whereas additional caudal
vertebrae are likely overlaid by the left ischium and some soft tissue. The caudal
vertebrae appear significantly shorter than the thoracic ones. The pygostyle of *C.*
*shifan* appears to be exposed in left lateral view (Fig. 4c). The pygostyle is
proportionately robust and subequal in length to the tarsometatarsus, as is generally
the case in confuciusornithids^{3,18}. The transversely thickened ventral margin of the
pygostyle is distinctly made up of the co-ossified centra, which are thinned caudally.
The ventral surface does not form a longitudinal keel as in *C. sanctus* IVPP V 12352

(holotype of the putative species *Jinzhouornis zhangjiyingia*)^{5,18}. Some 11 small round
foramina form a longitudinal row along the whole lateral furrow of the pygostyle (Fig.
4c). The foramina are closer to the ventral margin of the pygostyle than the dorsal
margin, and gradually become smaller and more tightly crowded together towards the
caudal end of the series. Similar foramina occur in other *confuciusornithine*
specimens, but are fewer in number and only present on the cranial part of the
pygostyle^{18,19}.

The coracoids are fused with the scapulae to form scapulocoracoids. The left and
right scapulocoracoids are exposed in lateral and medial views, respectively (Fig. 5a).
The long axes of the scapula and coracoid define an acute angle as in *Y. confucii*⁶ and
*more advanced* birds (Fig. 5a, 6). This angle is approximately 90° in *C. sanctus*³. A
laterally directed glenoid facet is visible in the left scapulocoracoid. The cranial and
caudal margins of the glenoid facet project more strongly than the dorsal and ventral
ones, implying that the configuration of the facet probably did not strongly limit the
elevation and depression of the wing in flapping flight. The acromion of the scapula is
well-developed. The shaft of the scapula curves slightly downward; the cranial half of
the shaft appears rod-like and slightly compressed dorsoventrally, and while the
caudal half is more compressed mediolaterally and widens significantly in the
dorsoventral direction before tapering to a point as in *more advanced* birds²⁰ (Fig. 5a,
6). In other *confuciusornithids*, the scapula is generally straight, and neither flares
partway along its length nor tapers caudally³. The coracoid is strut-like and
approximately half the length of the scapula. Its proximal end appears dorsoventrally
thicker than transversely wide and projects slightly cranially to form an acrocoracoid
process as in *Y. confucii*⁶, whereas its distal end is compressed dorsoventrally. The
acrocoracoid process is not well-developed in *C. sanctus*³. The lateral surface of the
coracoid is excavated to form a shallow groove, which gradually becomes narrower
and shallower distally as in *Y. confucii*⁶. The medial margin of the coracoid is strongly
compressed to form a longitudinal keel, except at the proximal end of the bone. The
furcula is mostly exposed in caudal view. As in other *confuciusornithids*, the furcula is
boomerang-shaped and without a distinct tubercle³⁻⁶ (Fig. 5a, 6). The proximomedial

corner of the right clavicular ramus is inflated caudally to form a bulbous structure
that contacts the medial face of the proximal end of the right scapulothoracoid. **and**
meanwhile the proximolateral corner is more craniocaudally compressed than the rest
of the ramus and slightly projects laterally, to form a distinct impression on the caudal
surface of the corner. Similar **structure** is not seen in other confuciusornithids.

The forelimb is subequal in length to the hindlimb, with a forelimb (humerus +
ulna) / hindlimb (femur + tibiotarsus) ratio of 1.02 (Fig. 1; Supplementary table S1).
The equivalent ratio is 0.93 in *C. hengdaoziensis*, 0.96 in *E. zhengi*, 1.06 in *C. dui*,
0.91~1.08 in *C. sanctus* and 1.19 in *Y. confucii*⁶ (Fig. 6). The humerus is typical of
confuciusornithids in having an expanded and perforated deltopectoral crest. However,
the crest is less prominent overall than those of other confuciusornithids except *C. dui*⁵,
and the point of the greatest prominence is at the distal end of the crest rather than in the
middle as in other confuciusornithids³⁻⁶ (Fig. 5a, 6). The dorsodistal corner of the crest
projects distally, so that the distal margin appears concave. Distally, a large fossa for the
brachialis muscle is present proximal to the dorsal and ventral condyles. The dorsal
epicondyle is better developed than the ventral epicondyle, and the former projects
sharply dorsally. Both the ulna and the radius are shorter than the humerus, as in other
confuciusornithids (Fig. 1, 6; Supplementary table S1, 2). **A concave ventral cotyle, a**
**flat dorsal cotyla and a concave incisura radialis are clearly visible on the proximal**
**end of the right ulna, whereas a distinctly projecting olecranon process is lacking. The**
**proximal humeral articular facet of the radius is flat and the bicipital tubercle is**
**situated on the cranial face of the proximal end of the radius.** Both radialia are in their
original positions relative to the radii and carpometacarpi (Fig. 1). The left radiale are
exposed in palmar and obliquely proximal view and the right in palmar and obliquely
distal view. The articular facets for the radius and semilunate carpal are both
significantly concave. The cranial surface of the radiale is convex and forms a small
tubercle on the proximal side, and the caudal surface is slightly concave and directed
somewhat proximally to articulate with the ulna.

The semilunate carpal is completely fused with metacarpals II and III to form a
carpometacarpus (Fig. 1, 7a,b). Metacarpal III is much more slender, and slightly

shorter, than metacarpal II, and contact between these metacarpals is limited to their
 proximal ends as in enantiornithines²¹. The middle parts of metacarpals II and III are
 constricted, and the narrowest part of metacarpal III is less than half the width of the
 narrowest part of metacarpal II as in other confuciusornithids. However, the
 intermetacarpal space appears relatively longer and significantly wider than in other
 confuciusornithids, as in some advanced birds²⁰. The proximal end of the
 carpometacarpus forms a pulley-like carpal trochlea as in *C. sanctus*³ and extant birds.
 A distinct fossa is present in the region where the semilunate carpal is fused to
 metacarpals II and III. The bump-like pisiform process is located near the proximal
 end of metacarpal II. The alular metacarpal is about one third the length of metacarpal
 II as in *C. sanctus*³ and *E. zhengi*⁴, rather than half the length as in *Ch.*
 *hengdaoziensis*³ (Fig. 6, 7a,b). It is not fused with the carpometacarpus (Fig. 7a,b).
 The caudal portion of the alular metacarpal extends farther proximally than the cranial
 portion so that the proximal surface of the metacarpal is concave, forming a cranial
 carpal fovea. The proximal two thirds of the cranial margin projects cranially to form
 the extensor process. The craniocaudal width of the process is nearly half that of the
 distal articular surface of the alular metacarpal (Fig. 7a,b,c). Such a well-developed
 extensor process is not seen in contemporaneous basal birds, or in most
 enantiornithines and ornithuromorphs^{20,21}. Unlike in previously known
 confuciusornithids, the caudal distal condyle of the alular metacarpal is
 well-developed, and projected more distoventrally than the rest of the metacarpal,
 evident in palmar view. The proximal phalanx of digit II is robust, maintains a
 constant diameter throughout its entire length, and is slightly shorter than the
 intermediate phalanx as in other confuciusornithids except *Ch. hengdaoziensis*³(Fig.
 1). In *Ch. hengdaoziensis*, the proximal phalanx is slightly longer than the
 intermediate one³. The ungual phalanges of the alular digit and digit III are large and
 curved, whereas that of digit II is small as in most confuciusornithids. In *Y. confucii*,
 however, the ungual phalanx of digit II is large⁶.

Among the fully articulated left and right pelvic girdles, only the pubes are
 relatively completely preserved (Fig. 5b). The right ilium appears relatively low, with

a slightly convex dorsal margin and a laterally projected antitrochanter. The pubis,
remarkably, is longer than the femur, being rod-like, slender, and strongly retroverted.
Because of a sharp bend in the pubic shaft, the long axis of the pubic symphysis is
nearly perpendicular to that of the shaft's proximal portion, a condition resembling
that seen in derived ornithuromorph birds.

The femoral head, neck and trochanteric crest are well-developed. The proximal
tarsals are completely fused with each other and with the tibia, forming a tibiotarsus.
The proximal ends of metatarsals II-IV are fused with each other and with the distal
tarsals, forming a tarsometatarsus. The proximal end of metatarsal III appears
transversely compressed between metatarsals II and IV in cranial view (Fig. 7d),
whereas this metatarsal in other confuciusornithids does not change in width along its
length³⁻⁶. An oval tubercle presumably for insertion of *M. tibialis cranialis* is located
on the dorsal surface near the proximal end as in *C. sanctus*³, but similar structure
does not exist on metatarsal II. A ridge-like process is present on the lateral margin of
the distal third of metatarsal IV (Fig. 7d), but it is not seen in other confuciusornithids.
In pedal digits III-IV, the penultimate phalanx is longer than the preceding phalanx, as
in other confuciusornithids⁶. However, the penultimate phalanx of pedal digit II is
subequal in length to the preceding phalanx.

*Secondary epiphyseal ossification* The most unusual skeletal feature exhibited by
PMoL-AB00178 is the presence of a cushion-like small bone between the distal end
of the alular metacarpal and the proximal end of the first alular phalanx (Fig. 7a,b). In
the left manus, the small bone contacts the cranial portions of the opposing articular
surfaces of the metacarpal and phalanx (Fig. 7a); in the right manus, the small bone
again contacts the cranial portion of the proximal articular surface of the first alular
phalanx, but rests against the cranial margin of the left alular metacarpal due to the
slight disarticulation of the alular digit (Fig. 7b). We identify this small bone as an
ossification that arose from a secondary ossification center within the distal epiphysis
of the alular metacarpal and is equivalent to the medial condyle seen on the
metacarpal's distal articular surface in other confuciusornithids (Fig. 7c), based on the

following evidence: (1) in the right manus, the small bone remains in apparently
natural alignment with the alular metacarpal, but not with the first alular phalanx; (2)
in other confuciusornithids the distal articular surface of the alular metacarpal is
ginglymoid⁶, and the caudal condyle protrudes slightly more distally than the cranial
one (Fig. 7c), whereas in both alular metacarpals of PMoL-AB00178 the caudal
condyle is distally prominent and cranial condyle is absent; and (3) in both first alular
phalanges of PMoL-AB00178, the proximal articular surface is similar in morphology
to those of other confuciusornithids, and a well-developed flexor tubercle is visible on
the left alular phalanx.

Discussion

Our phylogenetic analysis places *C. shifan* as the sister taxon to *C. sanctus*, and *E.*
*zhengi*, *C. dui*, *Y. confucii* and *Ch. hengdaoziensis* as successive outgroups to the *C.*
*sanctus* + *C. shifan* clade. However, the various confuciusornithid subclades involved
in this arrangement are not well supported, as indicated by their low Bremer and
Bootstrap values (Fig. 8).

The discovery of *C. shifan* sheds new light on morphological evolution and flight
adaptation in confuciusornithid birds. Among early pygostylians, both
confuciusornithid and enantiornithines have relatively large pygostyles¹⁸, although the
exact number of caudal vertebrae involved in the pygostyle is typically difficult to
determine due to their complete coossification. Perforated pygostyles, such as that of
PMoL-AB00178, are thought to be the result of incomplete fusion between adjacent
neural arches, and the foramina are finally enclosed^{18,19}. The pygostyle of
PMoL-AB00178 then presumably exemplifies the developmental stage at which
fusion among the neural arches was partial rather than complete. Given the presence
of 11 preserved foramina, the number of caudal vertebrae within the pygostyle is
estimated to be 12. If the estimate for the pygostyle is accurate and PMoL-AB00178
has seven free caudal vertebrae as in *C. sanctus*³, then 19 caudal vertebrae are present
in the tail as a whole (Fig. 1, 4c), a total only slightly less than the count of 21-23 free
caudal vertebrae reported for the long bony tail of *Archaeopteryx*²². This comparison

implies that the transition in birds from a long bony tail like that plesiomorphically
present in reptiles to a short one ending in a pygostyle was driven primarily by
shortening of the individual caudal vertebrae, rather than by a reduction in vertebral
count. The fact that the pygostyle is the only compound bone in PMoL-AB00178 to
show incomplete fusion also indicates that the tail may not have played a major role in
generating aerodynamic forces during flight.

The flight capability of confuciusornithids has been debated in previous studies.
Some authors have considered confuciusornithids competent flyers based on a number
of morphological features, including long wings with strongly asymmetrical flight
feathers, strut-like coracoids, a keeled sternum, and enlarged major manual digits²³,
whereas others have suggested more limited flight capabilities¹² based on functional
analysis of the morphology of the shoulder joint⁷ or the flight feathers⁹. Nevertheless,
there has been broad agreement that confuciusornithids display a more primitive suite
of flight-related characters than contemporaneous enantiornithines and
ornithuromorphs^{1,7}, suggesting inferior flight performance. However, morphological
evidence from both *C. shifan* and the recently discovered *Y. confucii* indicates that
some variation in flight capability existed within Confuciusornithidae, as both these
species appear to have been unusually well-adapted for flight, albeit in different ways.
*Y. confucii* and *C. shifan* both exhibit a more specialized flight apparatus than is
present in other confuciusornithids, or even in most contemporaneous enantiornithines
and ornithuromorphs. In *Y. confucii* the main modification is elongation of the
forelimb to increase the wing area as in sapeornithids²⁴ and some enantiornithine and
ornithuromorph birds. In *C. shifan* the main modifications involve refinements to the
pectoral girdle, humerus and carpometacarpus, which to some extent resemble their
counterparts in enantiornithine and ornithuromorph birds. In particular, some skeletal
processes to which flight muscles would have attached are particularly well developed.
A few such features, including the extensor process of the alular metacarpal and the
pisiform process of the carpometacarpus, are seen only in the contemporaneous
enantiornithine *Xiangornis*²¹, a few Early and Late Cretaceous ornithuromorphs²⁵, and
extant flying birds (Fig. 7a,b). In extant birds, the extensor process is the point of

insertion for *M. extensor metacarpi radialis*, the primary muscle involved in extension
of the wrist joint, and the pisiform process serves for attachment of the retinaculum
flexorium, and as a pulley changing the direction of the tendon of *M. flexor digitorum*
*profundus*²⁶, which crosses the wrist and inserts on the distal phalanx of the major
digit. A well-developed extensor process and pisiform process can accordingly
increase manipulating and controlling capabilities of the distal part of the wing during
flight²⁷, collectively suggesting greater flight capability in *C. shifan* than in other
confuciusornithids.

A recent study showed that most flight modes seen in extant birds (e.g.,
continuous flapping, flap-gliding, flap-bounding, thermal soaring) could potentially
have been represented among early birds²⁸. The flight parameter estimates obtained
for confuciusornithids in that analysis would suggest that they were continuous
flappers, but only the holotype of *E. zhengi* and a few specimens of *C. sanctus* were
considered. In order to further assess the flight capabilities and strategies of
confuciusornithid birds, we followed the methods of Serrano et al. (2017)²⁸ (also see
Aerodynamic analysis in Methods below) to estimate flight parameters for 11
confuciusornithid specimens, including the only known specimens of *Y. confucii* and
*C. shifan* (Supplementary table S2, 3), and plotted estimates of two key parameters for
these confuciusornithids, namely aspect ratio (AR) and wing loading (WL), on a
previously published figure showing the regions of AR-WL morphospace associated
with the main flight modes in extant birds²⁹. AR is the ratio of wing length to wing
width, representing a simple measure of wing shape, and WL is the ratio of body mass
to the total area of the two wings and the part of the trunk between them, representing
the load that must be borne by each unit area of the available lift-producing surface.
Both metrics are commonly used in assessing the flight capabilities and strategies of
flying animals. Our analyses show that both AR and WL vary considerably among
confuciusornithid specimens and even among specimens of *C. sanctus* (Supplemental
table S3; Fig. 9), as found by previous researchers²⁸. Though two specimens of *C.*
*sanctus* fall slightly into the area of morphospace where extant facultative flap-gliders
overlap with continuous flappers, most confuciusornithids plot at least narrowly

outside the flap-gliding area, and within the area occupied solely by continuous
flappers (Fig. 9). Interestingly, confuciusornithids are distributed throughout a large
part of the continuous flapping morphospace, and two ends represented by *E. zhengi*
and *Y. confucii* with the highest WL and AR, and *C. shifan* with the lowest WL and
AR, corresponding to two distinct flight strategies seen in the extant birds;
medium-to-large birds tend to have high AR and WL to increase lift efficiency and
flight speed, and minimize energy costs for long flights, whereas small birds and
particularly small forest-dwelling birds tend to have lower AR and WL to increase
maneuverability during shorter and slower flights³⁰⁻³². *C. shifan*, with its low values of
both AR and WL, lies well within continuous flapping morphospace but adjacent to
the areas of overlap with flap-gliders and flap-bounders.

Most importantly, our analyses under phylogeny may indicate an evolutionary
trend affecting confuciusornithid flight, involving a tendency for WL and AR to
decrease from early-diverging confuciusornithids such as *E. zhengi* and *Y. confucii* to
late-diverging ones such as *C. sanctus* and *C. shifan*. This suggests that the evolution
of improved flight ability in confuciusornithids was likely associated with an
ecological shift to a more densely vegetated environment. *C. shifan* represents the
culmination of this evolutionary trend, at least among currently known
confuciusornithids, which is reflected in the proximity of this species to flap-bounding
morphospace in Figure 9. *Y. confucii* may independently become more adapted for
open environment or long flight. The modified nature of their wings and shoulder
girdles further support this inference.

[revised manuscript text omitted]

The SOC in the distal epiphysis of the alular metacarpal of *C. shifan* (Fig. 7a,b) is
presumably an autapomorphic feature, whose appearance may be explicable on the
basis of a combination of growth strategy and functional demands on the wing. A
recent study has demonstrated that epiphyseal SOC's tend to evolve as a response to
high mechanical stress⁴⁷, and in *C. shifan* the distal end of the alular metacarpal may
well have begun to experience considerable stresses as soon as a developing juvenile
began to fly. In extant birds the alular metacarpal plays an important role in aerial
maneuvers, and the wings of *C. shifan* appear better-adapted than those of other
confuciusornithids for high-maneuverability flight in relatively closed environments.
As a result, the distal end of the alular metacarpal might have been more frequently
subject to large stresses in *C. shifan* than in most confuciusornithids. Furthermore, the
histological features of *C. shifan* suggest a slower growth rate than that of other
confuciusornithids, based on Amprino's rule⁴⁹. For example, *C. shifan* has
longitudinally oriented vascular canals in the middle layer of the femoral compact
bone (Fig. 2), whereas concentrically oriented canals are more numerous in other
confuciusornithids¹³. If the spread of ossification from the primary centers of the long
bones into the epiphyses was also slow, then the establishment of an epiphyseal SOC
might have been necessary to reinforce the distal end of the alular metacarpal at a
comparatively early ontogenetic stage, when the juvenile was beginning to fly but

skeletal growth was still incomplete. The holotype of *C. shifan*, however, clearly
represents a mature individual in which the caudal distal condyle of the alular
metacarpal had fully ossified by the time of death, presumably via the normal
mechanism of spread from the diaphyseal POC. We infer that the secondary
epiphyseal ossification would have fused with the rest of the alular metacarpal to
remodel a cranial distal condyle of more normal appearance, had the individual lived
longer.

**Methods**

**Histological preparation of *C. shifan* (PMoL-AB00178).** We took the bone sample
near the mid-diaphysis of the right femur of PMoL-AB00178 (Fig. 1a). Cross-section
was prepared using standard procedures⁵⁰. The sample was embedded in
one-component resin (EXAKT Technovit 7200), which was then hardened in a light
polymerization device (EXAKT 520). Thin cross-section was cut using an accurate
circular saw (EXAKT 300CP). The section was ground down using the EXAKT
400CS grinding system until the desired optical transparency was obtained. The
histological section was examined under a polarized light microscope (ZEISS Axio
Imager 2 Pol), and photographed with a ZEISS AxioCam 705 digital microscope
camera.

**Phylogenetic analysis.** To investigate the systematic position of *C. shifan* relative to
other Mesozoic birds, a phylogenetic analysis was carried out using the most
comprehensive data matrix currently available for Mesozoic birds⁵¹, with *C. shifan*
added in based on scorings from PMoL-AB00178. The revised matrix consists of 81
taxa and 280 characters. The matrix was analyzed using TNT v1.5 (22)⁵² with default
settings. All characters were equally weighted, with 35 characters ordered; the dataset
was analyzed using the “New Technology search” methods with sectorial search,
ratchet, tree drift, and tree fusion with default settings; the minimum-length tree was
found in 10 replicates to recover as many tree islands as possible. The recovered trees
were then used as the basis for a traditional TBR search. Zero-length branches were

collapsed. Decay indices (Bremer support values) were calculated using the Bremer
script embedded in TNT, and absolute Bootstrap frequencies were calculated using
1000 pseudoreplicates in TNT with default settings.

**Aerodynamic analysis.** In order to assess the flight capabilities and strategies of
confuciusornithid birds, we estimated values of several flight parameters for 11
confuciusornithid individuals, including the holotypes of *C. dui*, *Ch. hengdaoziensis*, *E.*
*zhengi*, *Y. confucii* and *C. shifan* (see Supplementary tables S2, 3). Specifically, we
estimated the body mass (BM), wingspan (B), and lift surface area (SL) of these
specimens following the methods of Seranno et al. (2015, 2017)^{15,28}, and subsequently
used these estimates as a basis for calculating aspect ratio (AR, $AR = B^2/SL$) and wing
loading (WL, $WL = BM/SL$) values (see Supplementary table S3). The anatomical
data needed for these estimates were obtained by measuring the specimens directly
with digital calipers, or alternatively from high-resolution images using tpsDig 2.17⁵³
(available at: <https://sbmorphometrics.org/>) from measurements published in the
literature, or from estimated measurements based on the methods of Serrano et
al. 2018²⁹ (see Supplementary table S2 for details).

[revised manuscript text omitted]

Figure 1. Photograph (a) and line drawing (b) of *Confuciusornis shifan* holotype
 (PMoL-AB00178). Arrow indicates the sampling position for the histological section.

Abbreviations: cav, caudal vertebra; cev, cervical vertebra; fu, furcula; ga, gastralia; lc,
 left coracoid; ldIII, left manual digit III; lfe, left femur; lh, left humerus; lil, left ilium;
 lis, left ischium; lm, left manus; lp, left pes; lra, left radiale; lr, left radius; ls, left
 scapula; lt, left tibiotarsus; lu, left ulna; lul, left ulnare; pu, pubis; py, pygostyle; r, rib;
 rc, right coracoid; rdcIII, caw of right manual digit III; rfe, right femur; rfi, right fibula;
 rh, right humerus; ris, right ischium; rm, right manus; rp, right pes; rra, right radiale; rr,
 right radius; rs, right scapula; rt, right tibiotarsus; ru, right ulna; rul, right ulnare; sk,
 skull; sy, synsacrum; tv, thoracic vertebra. Scale bars: 2 cm.

Figure 2. Photograph of a midshaft histological section of the right femur of the
*Confuciusornis shifan* holotype (PMoL-AB00178). White arrows indicate LAGs.

Abbreviations: ICL, inner circumferential layer; OCL, outer circumferential layer.

Scale bar: 100µm.

Figure 3. Photograph of the skull and mandible of the *Confuciusornis shifan* holotype

(PMoL-AB00178) in left ventrolateral view (a) with corresponding line drawing (b),

a close-up of the anterior margin of the orbit (c). White rectangle in (a) indicates the

region of (c). A line drawing of the posterior half of the mandible of *Confuciusornis*

*sanctus* IVPP V 13171 in Wang et al., 2018⁵ is appended below the line drawing of

the skull and mandible of the *Confuciusornis shifan*, and the arrow indicates the

position of the ventral process of the surangular. Abbreviations: an, angular; ar,

articular; cmf, caudal mandibular fenestra; d, dentary; f, frontal; j, jugal; l, lacrima; m,

maxilla; o, orbit; rmf, rostral mandibular fenestra; n, nasal; p, parietal; pm, premaxilla;

q, quadrate; sa, surangular; sp, splenial. Scale bars: 1 cm; note that (c) is not to scale.

Figure 4. Photographs of the cervical vertebrae (a), **synsacrum** (b) and pygostyle (c)

of the *Confuciusornis shifan* holotype (PMoL-AB00178). Abbreviation: sy,

synsacrum. Arrows in (c) indicate the positions of foramina along the pygostyle. Scale

bars: 0.5 cm in (a) and (c); 1 cm in (b).

Figure 5. Photographs of the pectoral (a) and pelvic (b) girdle of the *Confuciusornis*
*shifan* holotype (PMoL-AB00178). Abbreviation: fu, furcula; lc, left coracoid; lil, left
ilium; lis, left ischium; ls, left scapula; pu, pubis; rc, right coracoid; rh, right humerus;
ris, right ischium; rs, right scapula. Scale bars: 1 cm.

**Figure 6.** Line drawings of the pectoral **girdle**, **forelimb**, **hindlimb** of the
 *Confuciusornis shifan* holotype (PMoL-AB00178), and other confuciusornithids, with
 shading to indicate lengths of stylopodial, zeugopodial and metapodial limb segments.
 Values near the segments represent ratios of segment length to femoral length. All
 drawings scaled to a common, arbitrary femoral length.

Figure 7. Photographs of the left (a) and right (b) carpometacarpi and right
 tarsometatarsus (d) of *Confuciusornis shifan* holotype (PMoL-AB00178), and the left
 alular metacarpal (c) of the confuciusornithid specimen PMoL-AB00150.
 Carpometacarpi in palmar view, tarsometatarsus in cranial view. Black and white
 arrows in (d) indicate the ridge-like process on metatarsal IV and the dorsal tubercle
 on metatarsal III, respectively. Abbreviations: cb, cushion-like bone; ep, extensor
 process; pp, pisiform process. Roman numerals in (d) identify metatarsals. Scale bars:
 0.25 cm.

Figure 8. Cladogram of Mesozoic birds showing the systematic position of
 *Confuciusornis shifan*, and representing the strict consensus of the 192 most
 parsimonious trees recovered in the phylogenetic analysis performed in this study
 (length = 1404; consistency index = 0.277; retention index = 0.667). Bootstrap and
 Bremer values are given in normal font and in bold italic font, respectively, near the
 nodes to which they pertain.

Figure 9. Positions of 11 confuciusornithid specimens in a previously published
 morphospace defined by wing loading (WL) and aspect ratio (AR), showing the areas
 of morphospace occupied by extant birds with particular modes of flight²⁹. Specimens
 are indicated by black circles, and numbered in descending order of estimated AR.
 Specimens are holotypes of their respective species unless a number is indicated.

REVIEWERS' COMMENTS:

Reviewer #1 (Remarks to the Author):

For this second round of review, the authors seem to be honest about their errors and weakness of their hypotheses and have tried their best to correct issues pointed out by reviewers. Their claims are more conservative and fair in the current edition.

As another reviewer pointed out, issues with confuciusornithid taxonomy still remain, and their phylogeny may not be as informative as it should be until these issues are thoroughly addressed. Nevertheless, the present study puts forward interesting hypotheses about adaptation to flight in these early avialans, to me especially in terms of the evolution of pygostyle. Therefore, this study is not only meaningful in presenting a new species of *Confuciusornis* to add to their morphological diversity, but also advocative in suggesting the need to revisit confuciusornithid taxonomy for future to better understand evolution of non-ornithothoracines.

In this edition, I still found a few minor errors, which are highlighted in the attached PDF document. These errors are easy to fix and I do not need to check the manuscript again after this round of review as I know the authors would address these issues without problems. I think the manuscript is good to go for publication and am looking forward to see it out.

Reviewer #2 (Remarks to the Author):

The authors have greatly improved this work. The figures are much clearer after revisions, the diagnosis is more clear and better distinguished from *C. dui*, the discussion section is expanded with much interesting new analysis, and the additions to the supplement show the flight predictions have refreshingly tight confidence intervals (this should be noted in the main text I think as it is a great boon to the work over similar studies).

My suggested changes, marked on the manuscript PDF, are very minor. I believe the work is currently fit for publication after the minor changes I have suggested.

I do have one remaining major concern, and that is that I am unconvinced of the authors' arguments that this specimen should be referred to *Confuciusornis*. They argue on the basis that *shifan* is more similar to *Confuciusornis sanctus* than any other two confuciusornithids are to each other, but the same could be said for plenty of other Mesozoic birds assigned to different genera (e.g. *Bohaiornis* and *Parabohaiornis* vs all other bohaiornithids). The work itself acknowledges that additional taxonomic work may necessitate removing *C. shifan* from *Confuciusornis*, so I see no reason why it should be placed there in the first place. If future work affirms a high degree of similarity between *shifan* and *C. sanctus* there would remain no taxonomic issue if they were simply different genera that consistently resolved as sister taxa.

However, I must acknowledge that this is ultimately my preference, and the line between genus and species in paleontology is subjective. Referral of this specimen to *Confuciusornis* is not strictly incorrect at this stage, and it should not be the sole factor preventing the publication of this work.

A new confuciusornithid bird with a secondary epiphyseal ossification reveals phylogenetic changes in confuciusornithid flight mode

Renfei Wang^{1,2}, Dongyu Hu^{2*}, Meisheng Zhang¹, Shiyong Wang², Qi Zhao^{3,4}, Corwin Sullivan^{5,6} & Xing Xu^{3,4*}

¹ College of Earth Sciences, Jilin University, Changchun 130061, China

² Shenyang Normal University, Paleontological Museum of Liaoning, Key Laboratory for Evolution of Past Life in Northeast Asia, Liaoning Province, 253 North Huanghe Street, Shenyang 110034, China

³ Key Laboratory of Vertebrate Evolution and Human Origins, Institute of Vertebrate Paleontology and Paleoanthropology, Chinese Academy of Sciences, Beijing 100044, China

⁴ Center for Excellence in Life and Paleoenvironment, Chinese Academy of Sciences, Beijing 100044, China

⁵ Department of Biological Sciences, University of Alberta, Edmonton, AB T6G 2E9, Canada

⁶ Philip J. Currie Dinosaur Museum, Wembley, AB T0H 3S0, Canada

Correspondence and requests for materials should be addressed to D.H. (email: hudongyu@synu.edu.cn) and X.X. (email: xuxing@ivpp.ac.cn)

The confuciusornithids are the earliest known beaked birds, and constitute the only species-rich ~~and morphologically diverse~~ clade of Early Cretaceous pygostylian birds that existed prior to the cladogenesis of Ornithothoraces ~~two major avialan groups Enantiornithes and Ornithuromorpha~~. Here, we report a new confuciusornithid species from the Lower Cretaceous of western Liaoning, northeastern China. Compared to other confuciusornithids ~~and most contemporaneous enantiornithines and ornithuromorphs~~, this new species and the recently reported *Yangavis confucii* both show evidence of ~~stronger~~ higher level of stronger flight capability ~~adaptation~~, although the wings of the two taxa differ from one another in many respects. Our aerodynamic analyses under phylogeny ~~of the known confuciusornithids~~ indicate that varying modes of flight adaptation ~~considerable variations of flight capability and style emerged across the diversity of confuciusornithids~~, and to a lesser degree over the course of their ontogeny, and specifically suggest that both a trend towards improved flight capability and a change in flight strategy occurred in confuciusornithid evolution. The new confuciusornithid differs most saliently from other Mesozoic birds in having an extra cushion-like bone in the first digit of the wing ~~a secondary epiphyseal ossification, located in the alular digit~~, a highly unusual feature that may have helped to meet the functional demands of flight at a stage when skeletal growth was still incomplete. The new find strikingly exemplifies the morphological, developmental and functional diversity of the first beaked birds.

Introduction

Confuciusornithidae is a clade of Early Cretaceous pygostylian birds known from the Jehol Biota of East Asia¹, and representing the earliest known toothless, beaked birds. Five genera and eleven species, recovered from the Dabeigou, Yixian and Jiufotang formations (~135~120 Ma), have been described and assigned to this family, though the validity of some species is questionable²⁻⁶. Confuciusornithids are the only species-rich, ~~morphologically diverse pygostylian avialan~~ clade known to have existed prior to the cladogenesis of Ornithothoraces, ~~which the major groups Enantiornithes~~

~~and Ornithuromorpha~~, and are represented by thousands of exceptionally preserved specimens that collectively provide rich information on confuciusornithid morphology, taxonomy, flight ability, growth, diet and ecology^{3,5,7-13}. Here, we report a new confuciusornithid species, *Confuciusornis shifan* sp. nov., from the Jiufotang Formation. *Confuciusornis shifan* differs from other confuciusornithids in a number of morphological and developmental features, which have implications for understanding confuciusornithid taxonomic diversity, morphological disparity, development, and flight behavior.

Institutional abbreviations

DNHM, Dalian Natural History Museum (Dalian); GMV, National Geological Museum (Beijing); IVPP, Institute of Vertebrate Paleontology and Paleoanthropology (Beijing); MCFO, CosmoCaixa (Madrid); PMoL, Paleontological Museum of Liaoning (Shenyang).

Results

Systematic paleontology.

Avialae Gauthier, 1986

Aves Linnaeus, 1758

[revised manuscript text omitted]

~~from a pre-existing sesamoid as a “traction epiphysis”^{38,43,44}, or may represent a true secondary epiphyseal ossification comparable to those present in mammals and most lizards⁴⁵. Furthermore, the tarsal elements and some carpal elements of extant birds fuse to adjacent long bones in a manner superficially resembling the fusion of epiphyseal ossifications to metaphyses, and apophyseal ossifications are also known in some species^{72,74}.~~

Epiphyseal SOC's have been demonstrated to provide additional stiffness to articular portions of bones that require reinforcement, for example as a result of stresses associated with support and locomotion in the terrestrial environment, and to facilitate the formation of topographically complex joint surfaces^{65,76,77}. To our knowledge, however, epiphyseal SOC's do not occur in non-avian archosauriforms, including in large, undoubtedly terrestrial dinosaurs that would have experienced substantial locomotor stresses. Epiphyseal SOC's probably evolved independently in mammals, birds and lepidosaurs⁶⁹, for reasons potentially relating to patterns of skeletal growth as well as to mechanical factors. Furthermore, we are unaware of any previous report of an epiphyseal SOC in a non-ornithomorph bird. The proximal tibial center appears to have been present in hesperornithiforms based on differences between adult and juvenile tibiotarsi of the Early Cretaceous genus Enaliornis⁷⁸, but it is surprising that the holotype of *C. shifan* possesses an epiphyseal SOC anywhere in the skeleton, let alone at a location – the cranial portion of the distal end of the alular metacarpal – where no such feature occurs even in crown-group birds.

The SOC in the distal epiphysis of the alular metacarpal of *C. shifan* (Fig. 7a,b) is presumably an autapomorphic feature, whose presence may be explicable on the basis of a combination of growth strategy and functional demands on the wing. A recent study has demonstrated that epiphyseal SOC's tend to evolve as a response to high mechanical stress⁷⁷. ~~For example, epiphyseal SOC's are developed only in the legs and thumbs of newborn bats, which are used from birth to cling to the mother or the roost⁷⁷. In *C. shifan* the distal end of the alular metacarpal may well have begun to experience considerable stresses as soon as a developing juvenile began to fly.~~ In extant birds the alular metacarpal plays an important role in aerial maneuvers^{58,59}, and

[revised manuscript text omitted]

Figure 2. Photograph of a midshaft histological section of the right femur of the *Confuciusornis shifan* holotype (PMoL-AB00178). Arrows indicate LAGs. Abbreviations: ICL, inner circumferential layer; OCL, outer circumferential layer. Scale bar: 100 μ m.

Figure 3. Photograph of the skull and mandible of the *Confuciusornis shifan* holotype (PMoL-AB00178) in left ventrolateral view (a) with corresponding line drawing (b), and a close-up of the anterior margin of the orbit (c). White rectangle in (a) indicates the region shown in (c). A line drawing of the posterior half of the mandible of *Confuciusornis sanctus* IVPP V 13171 (drafted based on Figure 1B in Wang et al., 2018⁵) is presented below the line drawing of the skull and mandible of *Confuciusornis shifan*; arrow indicates the ventral process of the surangular. Abbreviations: an, angular; ar, articular; cmf, caudal mandibular fenestra; d, dentary; f, frontal; j, jugal; l, lacrimal; m, maxilla; o, orbit; rmf, rostral mandibular fenestra; n, nasal; p, parietal; pm, premaxilla; q, quadrate; sa, surangular; sp, splenial. Scale bars: 1 cm; note that (c) is not to scale.

Figure 4. Photographs of the cervical vertebrae **(a)**, synsacrum **(b)** and pygostyle **(c)** of the *Confuciusornis shifan* holotype (PMoL-AB00178) (Also see Supplementary figure S1 for a close-up of the distal part of the pygostyle). Arrows in **(b)** and **(c)** indicate the longitudinal ridge on the ventral surface of the synsacrum and the positions of foramina along the pygostyle, respectively. Abbreviation: sy, synsacrum. Scale bars: 0.5 cm in **(a)** and **(c)**; 1 cm in **(b)**.

Figure 5. Photographs of the pectoral (a) and pelvic (b) girdles of the *Confuciusornis shifan* holotype (PMoL-AB00178). Abbreviations: fu, furcula; lc, left coracoid; lil, left ilium; lis, left ischium; ls, left scapula; pu, pubis; rc, right coracoid; rh, right humerus; ris, right ischium; rs, right scapula. Scale bars: 1 cm.

Figure 6. Line drawings of the pectoral girdles, forelimbs, and hindlimbs of the *Confuciusornis shifan* holotype (PMoL-AB00178) and other confuciusornithids, with shading to indicate lengths of stylopodial, zeugopodial and metapodial limb segments. Values near the segments represent ratios of segment length to femoral length. All drawings scaled to a common, arbitrary femoral length.

Figure 7. Photographs of the left (a) and right (b) carpometacarpi ~~and right tarsometatarsus (d)~~ of the *Confuciusornis shifan* holotype (PMoL-AB00178), the left alular metacarpal (c) of the confuciusornithid specimen PMoL-AB00150 and the right tarsometatarsus (d) of the *Confuciusornis shifan* holotype (PMoL-AB00178).

Carpometacarpi in palmar view, tarsometatarsus in cranial view. Black and white arrows in (d) indicate the ridge-like process on metatarsal IV and the dorsal tubercle on metatarsal III, respectively. Abbreviations: cb, cushion-like bone; ep, extensor process; pp, pisiform process. Roman numerals in (d) identify metatarsals. Scale bars: 0.25 cm.

Figure 8. Cladogram of Mesozoic birds showing the systematic position of *Confuciusornis shifan*, and representing the strict consensus of the 192 most parsimonious trees recovered in the phylogenetic analysis performed in this study (length = 1404; consistency index = 0.277; retention index = 0.667). The bootstrap and bremer values over the the minimum threshold are given in normal font and bold italic font, respectively, near the nodes to which they pertain.

Figure 9. Positions of 11 confuciusornithid specimens in a previously published morphospace defined by wing loading (WL) and aspect ratio (AR), showing the areas of morphospace occupied by extant birds with particular modes of flight⁶². Specimens are indicated by black circles, and numbered in descending order of estimated body mass. Specimens are holotypes of their respective species unless a number is indicated.

A new confuciusornithid bird with a secondary epiphyseal ossification reveals phylogenetic changes in confuciusornithid flight mode

Renfei Wang^{1,2}, Dongyu Hu^{2*}, Meisheng Zhang¹, Shiyong Wang², Qi Zhao^{3,4}, Corwin Sullivan^{5,6} & Xing Xu^{3,4*}

¹ College of Earth Sciences, Jilin University, Changchun 130061, China

² Shenyang Normal University, Paleontological Museum of Liaoning, Key Laboratory for Evolution of Past Life in Northeast Asia, Liaoning Province, 253 North Huanghe Street, Shenyang 110034, China

³ Key Laboratory of Vertebrate Evolution and Human Origins, Institute of Vertebrate Paleontology and Paleoanthropology, Chinese Academy of Sciences, Beijing 100044, China

⁴ Center for Excellence in Life and Paleoenvironment, Chinese Academy of Sciences, Beijing 100044, China

⁵ Department of Biological Sciences, University of Alberta, Edmonton, AB T6G 2E9, Canada

⁶ Philip J. Currie Dinosaur Museum, Wembley, AB T0H 3S0, Canada

Correspondence and requests for materials should be addressed to D.H. (email: hudongyu@synu.edu.cn) and X.X. (email: xuxing@ivpp.ac.cn)

The confuciusornithids are the earliest known beaked birds, and constitute the only species-rich ~~and morphologically diverse~~ clade of Early Cretaceous ~~pygostylian~~ birds that existed prior to the cladogenesis of ~~Ornithothorace~~ two major ~~avialan~~ groups ~~Enantiornithes and Ornithuromorpha~~. Here, we report a new confuciusornithid species from the Lower Cretaceous of western Liaoning, northeastern China. Compared to other confuciusornithids ~~and most contemporaneous enantiornithines and ornithuromorphs~~, this new species and the recently reported *Yangavis confucii* both show evidence of ~~stronger~~ ~~higher level of~~ ~~stronger~~ flight ~~capability~~ ~~adaptation~~, although the wings of the two taxa differ from one another in many respects. ~~Our~~ aerodynamic analyses under phylogeny ~~of the known confuciusornithids~~ indicate ~~that~~ ~~varying modes of flight adaptation~~ ~~considerable variations of flight capability and style~~ ~~emerged across the diversity of confuciusornithids~~, and to a lesser degree over the course of their ontogeny, and specifically suggest ~~that~~ both a trend towards improved flight capability and a change in flight strategy occurred in confuciusornithid evolution. ~~The~~ new confuciusornithid differs most saliently from other Mesozoic birds in having ~~an extra cushion-like bone in the first digit of the wing~~ ~~a secondary epiphyseal ossification, located in the alular digit~~, a highly unusual feature that may have helped to meet the functional demands of flight at a stage when skeletal growth was still incomplete. The new find strikingly exemplifies the morphological, developmental and functional diversity of the first beaked birds.

Introduction

Confuciusornithidae is a clade of Early Cretaceous pygostylian birds known from the Jehol Biota of East Asia¹, and representing the earliest known toothless, beaked birds. Five genera and eleven species, recovered from the Dabeigou, Yixian and Jiufotang formations (~135~120 Ma), have been described and assigned to this family, though the validity of some species is questionable²⁻⁶. Confuciusornithids are the only species-rich, ~~morphologically diverse pygostylian avialan~~ clade known to have existed prior to the cladogenesis of ~~Ornithothoraces, which~~ ~~the major groups~~ ~~Enantiornithes~~

~~and Ornithuromorpha~~, and are represented by thousands of exceptionally preserved specimens that collectively provide rich information on confuciusornithid morphology, taxonomy, flight ability, growth, diet and ecology^{3,5,7-13}. Here, we report a new confuciusornithid species, *Confuciusornis shifan* sp. nov., from the Jiufotang Formation. *Confuciusornis shifan* differs from other confuciusornithids in a number of morphological and developmental features, which have implications for understanding confuciusornithid taxonomic diversity, morphological disparity, development, and flight behavior.

Institutional abbreviations

DNHM, Dalian Natural History Museum (Dalian); GMV, National Geological Museum (Beijing); IVPP, Institute of Vertebrate Paleontology and Paleoanthropology (Beijing); MCFO, CosmoCaixa (Madrid); PMoL, Paleontological Museum of Liaoning (Shenyang).

Results

Systematic paleontology.

Avialae Gauthier, 1986

Aves Linnaeus, 1758

[revised manuscript text omitted]

~~from a pre-existing sesamoid as a “traction epiphysis”^{38,43,44}, or may represent a true secondary epiphyseal ossification comparable to those present in mammals and most lizards⁴⁵.~~ Furthermore, the tarsal elements and some carpal elements of extant birds fuse to adjacent long bones in a manner superficially resembling the fusion of epiphyseal ossifications to metaphyses, and apophyseal ossifications are also known in some species^{72,74}.

Epiphyseal SOC's have been demonstrated to provide additional stiffness to articular portions of bones that require reinforcement, for example as a result of stresses associated with support and locomotion in the terrestrial environment, and to facilitate the formation of topographically complex joint surfaces^{65,76,77}. To our knowledge, however, epiphyseal SOC's do not occur in non-avian archosauriforms, including in large, undoubtedly terrestrial dinosaurs that would have experienced substantial locomotor stresses. Epiphyseal SOC's probably evolved independently in mammals, birds and lepidosaurs⁶⁹, for reasons potentially relating to patterns of skeletal growth as well as to mechanical factors. Furthermore, we are unaware of any previous report of an epiphyseal SOC in a non-ornithomorph bird. The proximal tibial center appears to have been present in hesperornithiforms based on differences between adult and juvenile tibiotarsi of the Early Cretaceous genus Enaliornis⁷⁸, but it is surprising that the holotype of *C. shifan* possesses an epiphyseal SOC anywhere in the skeleton, let alone at a location – the cranial portion of the distal end of the alular metacarpal – where no such feature occurs even in crown-group birds.

The SOC in the distal epiphysis of the alular metacarpal of *C. shifan* (Fig. 7a,b) is presumably an autapomorphic feature, whose presence may be explicable on the basis of a combination of growth strategy and functional demands on the wing. A recent study has demonstrated that epiphyseal SOC's tend to evolve as a response to high mechanical stress⁷⁷. ~~For example, epiphyseal SOC's are developed only in the legs and thumbs of newborn bats, which are used from birth to cling to the mother or the roost⁷⁷. In *C. shifan* the distal end of the alular metacarpal may well have begun to experience considerable stresses as soon as a developing juvenile began to fly.~~ In extant birds the alular metacarpal plays an important role in aerial maneuvers^{58,59}, and

the wings of *C. shifan* appear better-adapted for high-maneuverability flight than those of other confuciusornithids ~~for high-maneuverability flight in relatively closed environments~~. As a result, the distal end of the alular metacarpal of *C. shifan* may well have begun to experience~~been more frequently subject to large~~ considerable stresses ~~in *C. shifan* than in most confuciusornithids~~ as soon as a developing juvenile began to fly. Furthermore, the histological features of *C. shifan* suggest a slower growth rate than that of other confuciusornithids, based on Amprino's rule⁷⁹ (Fig. 2). For example, *C. shifan* has longitudinally oriented vascular canals in the middle layer of the femoral compact bone (Fig. 2), whereas concentrically oriented canals are more numerous in other confuciusornithids¹³. If the spread of ossification from the primary centers of the long bones into the epiphyses was also slow, then the establishment of an epiphyseal SOC might have been necessary to reinforce the distal end of the alular metacarpal at a comparatively early ontogenetic stage, when the juvenile was beginning to fly but skeletal growth was still incomplete. The holotype of *C. shifan*, however, clearly represents a mature individual in which the caudal distal condyle of the alular metacarpal had fully ossified by the time of death, presumably via the normal mechanism of spread from the diaphyseal POC. ~~We infer that the secondary epiphyseal ossification would have fused with the rest of the alular metacarpal to remodel a cranial distal condyle of more normal appearance, had the individual lived longer.~~

[revised manuscript text omitted]

32. Xu, X., Chen, Y. N., Wang, X. L. & Chang, C. Pygostyle-like Structure from *Beipiaosaurus* (Theropoda, Therizinosauroida) from the Lower Cretaceous Yixian Formation of Liaoning, China. *Acta. Geol. Sin.* **77**, 294–298 (2003).
33. Barsbold, R. et al. A pygostyle from a non-avian theropod. *Nature* **403**, 155–156 (2000).
34. Persons, W. S., Currie, P. J. & Norell, M. A. *Oviraptorosaurus* tail forms and functions. *Acta. Palaeont. Pol.* **59(3)**, 553–567 (2014).
35. Zhang, F. C., Zhou, Z. H., Xu, X., Wang, X. L. & Sullivan, C. A bizarre Jurassic maniraptoran from China with elongate ribbon-like feathers. *Nature* **455**, 1105–1108 (2008).
36. Wang, M., O'Connor, J. K., Xu, X. & Zhou, Z. H. A new Jurassic scansoriopterygid and the loss of membranous wings in theropod dinosaurs. *Nature* **569**, 256–259 (2019).
37. Pittman, M. et al. In *Pennaraptoran theropod dinosaurs past progress and new frontiers* (eds. Pittman, M. & Xu, X.) Pennaraptoran Systematics (American Museum of Natural History, New York, 2020).
38. Rashid, D. J. et al. From dinosaurs to birds: a tail of evolution. *EvoDevo.* **5(25)**, 1–20 (2014).
39. Chiappe, L. M. & Walker, C. In *Mesozoic Birds: Above the Heads of Dinosaurs* (eds. Chiappe, L. M. & Witmer, L. M.) Skeletal morphology and systematics of the Cretaceous Enantiornithes (University of California Press, Berkeley, 2002).
40. Zhou, S., Zhou, Z. H. & O'Connor, J. K. Anatomy of the Early Cretaceous Archaeorhynchus spathula. *J. Vertebr. Paleontol.* **33**, 141–152 (2013).
41. Baumel, J. J., King, A. S., Breazile, J. E., Evans, H. E. & Berge, J. C. V. (eds.) *Handbook of Avian anatomy: Nomina Anatomica Avium*, 2nd edn. (Nuttall Ornithological Club, Cambridge, 1993).

[revised manuscript text omitted]

Figure 2. Photograph of a midshaft histological section of the right femur of the *Confuciusornis shifan* holotype (PMoL-AB00178). Arrows indicate LAGs. Abbreviations: ICL, inner circumferential layer; OCL, outer circumferential layer. Scale bar: 100 μ m.

Figure 3. Photograph of the skull and mandible of the *Confuciusornis shifan* holotype (PMoL-AB00178) in left ventrolateral view (a) with corresponding line drawing (b), and a close-up of the anterior margin of the orbit (c). White rectangle in (a) indicates the region shown in (c). A line drawing of the posterior half of the mandible of *Confuciusornis sanctus* IVPP V 13171 (drafted based on Figure 1B in Wang et al., 2018⁵) is presented below the line drawing of the skull and mandible of *Confuciusornis shifan*; arrow indicates the ventral process of the surangular. Abbreviations: an, angular; ar, articular; cmf, caudal mandibular fenestra; d, dentary; f, frontal; j, jugal; l, lacrimal; m, maxilla; o, orbit; rmf, rostral mandibular fenestra; n, nasal; p, parietal; pm, premaxilla; q, quadrate; sa, surangular; sp, splenial. Scale bars: 1 cm; note that (c) is not to scale.

Figure 4. Photographs of the cervical vertebrae (a), synsacrum (b) and pygostyle (c) of the *Confuciusornis shifan* holotype (PMoL-AB00178) (Also see Supplementary figure S1 for a close-up of the distal part of the pygostyle). Arrows in (b) and (c) indicate the longitudinal ridge on the ventral surface of the synsacrum and the positions of foramina along the pygostyle, respectively. Abbreviation: sy, synsacrum. Scale bars: 0.5 cm in (a) and (c); 1 cm in (b).

Figure 5. Photographs of the pectoral (a) and pelvic (b) girdles of the *Confuciusornis shifan* holotype (PMoL-AB00178). Abbreviations: fu, furcula; lc, left coracoid; lil, left ilium; lis, left ischium; ls, left scapula; pu, pubis; rc, right coracoid; rh, right humerus; ris, right ischium; rs, right scapula. Scale bars: 1 cm.

Figure 6. Line drawings of the pectoral girdles, forelimbs, and hindlimbs of the *Confuciusornis shifan* holotype (PMoL-AB00178) and other confuciusornithids, with shading to indicate lengths of stylopodial, zeugopodial and metapodial limb segments. Values near the segments represent ratios of segment length to femoral length. All drawings scaled to a common, arbitrary femoral length.

Figure 7. Photographs of the left (a) and right (b) carpometacarpi ~~and right tarsometatarsus (d)~~ of the *Confuciusornis shifan* holotype (PMoL-AB00178), the left alular metacarpal (c) of the confuciusornithid specimen PMoL-AB00150 and the right tarsometatarsus (d) of the *Confuciusornis shifan* holotype (PMoL-AB00178).

Carpometacarpi in palmar view, tarsometatarsus in cranial view. **Black and white arrows in (d)** indicate the ridge-like process on metatarsal IV and the dorsal tubercle on metatarsal III, respectively. Abbreviations: cb, cushion-like bone; ep, extensor process; pp, pisiform process. Roman numerals in (d) identify metatarsals. Scale bars: 0.25 cm.

Figure 8. Cladogram of Mesozoic birds showing the systematic position of *Confuciusornis shifan*, and representing the strict consensus of the 192 most parsimonious trees recovered in the phylogenetic analysis performed in this study (length = 1404; consistency index = 0.277; retention index = 0.667). The bootstrap and bremer values over the the minimum threshold are given in normal font and bold italic font, respectively, near the nodes to which they pertain.

Figure 9. Positions of 11 confuciusornithid specimens in a previously published morphospace defined by wing loading (WL) and aspect ratio (AR), showing the areas of morphospace occupied by extant birds with particular modes of flight⁶². Specimens are indicated by black circles, and numbered in descending order of estimated body mass. Specimens are holotypes of their respective species unless a number is indicated.

A new confuciusornithid bird with a secondary epiphyseal ossification reveals phylogenetic changes in confuciusornithid flight mode

Renfei Wang^{1,2}, Dongyu Hu^{2*}, Meisheng Zhang¹, Shiyong Wang², Qi Zhao^{3,4}, Corwin Sullivan^{5,6} & Xing Xu^{3,4*}

¹ College of Earth Sciences, Jilin University, Changchun 130061, China

² Shenyang Normal University, Paleontological Museum of Liaoning, Key Laboratory for Evolution of Past Life in Northeast Asia, Liaoning Province, 253 North Huanghe Street, Shenyang 110034, China

³ Key Laboratory of Vertebrate Evolution and Human Origins, Institute of Vertebrate Paleontology and Paleoanthropology, Chinese Academy of Sciences, Beijing 100044, China

⁴ Center for Excellence in Life and Paleoenvironment, Chinese Academy of Sciences, Beijing 100044, China

⁵ Department of Biological Sciences, University of Alberta, Edmonton, AB T6G 2E9, Canada

⁶ Philip J. Currie Dinosaur Museum, Wembley, AB T0H 3S0, Canada

Correspondence and requests for materials should be addressed to D.H. (email: hudongyu@synu.edu.cn) and X.X. (email: xuxing@ivpp.ac.cn)

The confuciusornithids are the earliest known beaked birds, and constitute the only species-rich ~~and morphologically diverse~~ clade of Early Cretaceous ~~pygostylian~~ birds that existed prior to the cladogenesis of ~~Ornithothorace~~ ~~two major avialan groups Enantiornithes and Ornithuromorpha~~. Here, we report a new confuciusornithid species from the Lower Cretaceous of western Liaoning, northeastern China. Compared to other confuciusornithids ~~and most contemporaneous enantiornithines and ornithuromorphs~~, this new species and the recently reported *Yangavis confucii* both show evidence of ~~stronger~~ ~~higher level of stronger~~ flight ~~capability~~ ~~adaptation~~, although the wings of the two taxa differ from one another in many respects. ~~Our aerodynamic analyses under phylogeny of the known confuciusornithids indicate that varying modes of flight adaptation~~ ~~considerable variations of flight capability and style emerged across the diversity of confuciusornithids~~, and to a lesser degree over the course of their ontogeny, and specifically suggest ~~that~~ both a trend towards improved flight capability and a change in flight strategy occurred in confuciusornithid evolution. ~~The~~ new confuciusornithid differs most saliently from other Mesozoic birds in having ~~an extra cushion-like bone in the first digit of the wing~~ ~~a secondary epiphyseal ossification, located in the alular digit~~, a highly unusual feature that may have helped to meet the functional demands of flight at a stage when skeletal growth was still incomplete. The new find strikingly exemplifies the morphological, developmental and functional diversity of the first beaked birds.

Introduction

Confuciusornithidae is a clade of Early Cretaceous pygostylian birds known from the Jehol Biota of East Asia¹, and ~~representing~~ the earliest known toothless, beaked birds. Five genera and eleven species, recovered from the Dabeigou, Yixian and Jiufotang formations (~135~120 Ma), have been described and assigned to this family, though the validity of some species is questionable²⁻⁶. Confuciusornithids are the only species-rich, ~~morphologically diverse pygostylian avialan~~ clade known to have existed prior to the cladogenesis of ~~Ornithothoraces, which~~ ~~the major groups Enantiornithes~~

~~and Ornithuromorpha~~, and are represented by thousands of exceptionally preserved specimens that collectively provide rich information on confuciusornithid morphology, taxonomy, flight ability, growth, diet and ecology^{3,5,7-13}. Here, we report a new confuciusornithid species, *Confuciusornis shifan* sp. nov., from the Jiufotang Formation. *Confuciusornis shifan* differs from other confuciusornithids in a number of morphological and developmental features, which have implications for understanding confuciusornithid taxonomic diversity, morphological disparity, development, and flight behavior.

Institutional abbreviations

DNHM, Dalian Natural History Museum (Dalian); GMV, National Geological Museum (Beijing); IVPP, Institute of Vertebrate Paleontology and Paleoanthropology (Beijing); MCFO, CosmoCaixa (Madrid); PMoL, Paleontological Museum of Liaoning (Shenyang).

Results

Systematic paleontology.

Avialae Gauthier, 1986

Aves Linnaeus, 1758

[revised manuscript text omitted]

~~from a pre-existing sesamoid as a “traction epiphysis”^{38,43,44}, or may represent a true secondary epiphyseal ossification comparable to those present in mammals and most lizards⁴⁵.~~ Furthermore, the tarsal elements and some carpal elements of extant birds fuse to adjacent long bones in a manner superficially resembling the fusion of epiphyseal ossifications to metaphyses, and apophyseal ossifications are also known in some species^{72,74}.

Epiphyseal SOC^s have been demonstrated to provide additional stiffness to articular portions of bones that require reinforcement, for example as a result of stresses associated with support and locomotion in the terrestrial environment, and to facilitate the formation of topographically complex joint surfaces^{65,76,77}. To our knowledge, however, epiphyseal SOC^s do not occur in non-avian archosauriforms, including in large, undoubtedly terrestrial dinosaurs that would have experienced substantial locomotor stresses. Epiphyseal SOC^s probably evolved independently in mammals, birds and lepidosaurs⁶⁹, for reasons potentially relating to patterns of skeletal growth as well as to mechanical factors. Furthermore, we are unaware of any previous report of an epiphyseal SOC in a non-ornithomorph bird. The proximal tibial center appears to have been present in hesperornithiforms based on differences between adult and juvenile tibiotarsi of the Early Cretaceous genus Enaliornis⁷⁸, but it is surprising that the holotype of *C. shifan* possesses an epiphyseal SOC anywhere in the skeleton, let alone at a location – the cranial portion of the distal end of the alular metacarpal – where no such feature occurs even in crown-group birds.

The SOC in the distal epiphysis of the alular metacarpal of *C. shifan* (Fig. 7a,b) is presumably an autapomorphic feature, whose presence may be explicable on the basis of a combination of growth strategy and functional demands on the wing. A recent study has demonstrated that epiphyseal SOC^s tend to evolve as a response to high mechanical stress⁷⁷. ~~For example, epiphyseal SOC^s are developed only in the legs and thumbs of newborn bats, which are used from birth to cling to the mother or the roost⁷⁷. In *C. shifan* the distal end of the alular metacarpal may well have begun to experience considerable stresses as soon as a developing juvenile began to fly.~~ In extant birds the alular metacarpal plays an important role in aerial maneuvers^{58,59}, and

the wings of *C. shifan* appear better-adapted for high-maneuverability flight than those of other confuciusornithids ~~for high-maneuverability flight in relatively closed environments~~. As a result, the distal end of the alular metacarpal of *C. shifan* may well have begun to experience~~been more frequently subject to large~~ considerable stresses ~~in *C. shifan* than in most confuciusornithids~~ as soon as a developing juvenile began to fly. Furthermore, the histological features of *C. shifan* suggest a slower growth rate than that of other confuciusornithids, based on Amprino's rule⁷⁹ (Fig. 2). For example, *C. shifan* has longitudinally oriented vascular canals in the middle layer of the femoral compact bone (Fig. 2), whereas concentrically oriented canals are more numerous in other confuciusornithids¹³. If the spread of ossification from the primary centers of the long bones into the epiphyses was also slow, then the establishment of an epiphyseal SOC might have been necessary to reinforce the distal end of the alular metacarpal at a comparatively early ontogenetic stage, when the juvenile was beginning to fly but skeletal growth was still incomplete. The holotype of *C. shifan*, however, clearly represents a mature individual in which the caudal distal condyle of the alular metacarpal had fully ossified by the time of death, presumably via the normal mechanism of spread from the diaphyseal POC. ~~We infer that the secondary epiphyseal ossification would have fused with the rest of the alular metacarpal to remodel a cranial distal condyle of more normal appearance, had the individual lived longer.~~

[revised manuscript text omitted]

32. Xu, X., Chen, Y. N., Wang, X. L. & Chang, C. Pygostyle-like Structure from *Beipiaosaurus* (Theropoda, Therizinosauroida) from the Lower Cretaceous Yixian Formation of Liaoning, China. *Acta. Geol. Sin.* **77**, 294–298 (2003).
33. Barsbold, R. et al. A pygostyle from a non-avian theropod. *Nature* **403**, 155–156 (2000).
34. Persons, W. S., Currie, P. J. & Norell, M. A. *Oviraptorosaurus* tail forms and functions. *Acta. Palaeont. Pol.* **59(3)**, 553–567 (2014).
35. Zhang, F. C., Zhou, Z. H., Xu, X., Wang, X. L. & Sullivan, C. A bizarre Jurassic maniraptoran from China with elongate ribbon-like feathers. *Nature* **455**, 1105–1108 (2008).
36. Wang, M., O'Connor, J. K., Xu, X. & Zhou, Z. H. A new Jurassic scansoriopterygid and the loss of membranous wings in theropod dinosaurs. *Nature* **569**, 256–259 (2019).
37. Pittman, M. et al. In *Pennaraptoran theropod dinosaurs past progress and new frontiers* (eds. Pittman, M. & Xu, X.) Pennaraptoran Systematics (American Museum of Natural History, New York, 2020).
38. Rashid, D. J. et al. From dinosaurs to birds: a tail of evolution. *EvoDevo.* **5(25)**, 1–20 (2014).
39. Chiappe, L. M. & Walker, C. In *Mesozoic Birds: Above the Heads of Dinosaurs* (eds. Chiappe, L. M. & Witmer, L. M.) Skeletal morphology and systematics of the Cretaceous Enantiornithes (University of California Press, Berkeley, 2002).
40. Zhou, S., Zhou, Z. H. & O'Connor, J. K. Anatomy of the Early Cretaceous Archaeorhynchus spathula. *J. Vertebr. Paleontol.* **33**, 141–152 (2013).
41. Baumel, J. J., King, A. S., Breazile, J. E., Evans, H. E. & Berge, J. C. V. (eds.) *Handbook of Avian anatomy: Nomina Anatomica Avium*, 2nd edn. (Nuttall Ornithological Club, Cambridge, 1993).

[revised manuscript text omitted]

Figure 2. Photograph of a midshaft histological section of the right femur of the *Confuciusornis shifan* holotype (PMoL-AB00178). Arrows indicate LAGs. Abbreviations: ICL, inner circumferential layer; OCL, outer circumferential layer. Scale bar: 100 μ m.

Figure 3. Photograph of the skull and mandible of the *Confuciusornis shifan* holotype (PMoL-AB00178) in left ventrolateral view (a) with corresponding line drawing (b), and a close-up of the anterior margin of the orbit (c). White rectangle in (a) indicates the region shown in (c). A line drawing of the posterior half of the mandible of *Confuciusornis sanctus* IVPP V 13171 (drafted based on Figure 1B in Wang et al., 2018⁵) is presented below the line drawing of the skull and mandible of *Confuciusornis shifan*; arrow indicates the ventral process of the surangular. Abbreviations: an, angular; ar, articular; cmf, caudal mandibular fenestra; d, dentary; f, frontal; j, jugal; l, lacrimal; m, maxilla; o, orbit; rmf, rostral mandibular fenestra; n, nasal; p, parietal; pm, premaxilla; q, quadrate; sa, surangular; sp, splenial. Scale bars: 1 cm; note that (c) is not to scale.

Figure 4. Photographs of the cervical vertebrae (a), synsacrum (b) and pygostyle (c) of the *Confuciusornis shifan* holotype (PMoL-AB00178) (Also see Supplementary figure S1 for a close-up of the distal part of the pygostyle). Arrows in (b) and (c) indicate the longitudinal ridge on the ventral surface of the synsacrum and the positions of foramina along the pygostyle, respectively. Abbreviation: sy, synsacrum. Scale bars: 0.5 cm in (a) and (c); 1 cm in (b).

Figure 5. Photographs of the pectoral (a) and pelvic (b) girdles of the *Confuciusornis shifan* holotype (PMoL-AB00178). Abbreviations: fu, furcula; lc, left coracoid; lil, left ilium; lis, left ischium; ls, left scapula; pu, pubis; rc, right coracoid; rh, right humerus; ris, right ischium; rs, right scapula. Scale bars: 1 cm.

Figure 6. Line drawings of the pectoral girdles, forelimbs, and hindlimbs of the *Confuciusornis shifan* holotype (PMoL-AB00178) and other confuciusornithids, with shading to indicate lengths of stylopodial, zeugopodial and metapodial limb segments. Values near the segments represent ratios of segment length to femoral length. All drawings scaled to a common, arbitrary femoral length.

Figure 7. Photographs of the left (a) and right (b) carpometacarpi ~~and right tarsometatarsus (d)~~ of the *Confuciusornis shifan* holotype (PMoL-AB00178), the left alular metacarpal (c) of the confuciusornithid specimen PMoL-AB00150 and the right tarsometatarsus (d) of the *Confuciusornis shifan* holotype (PMoL-AB00178).

Carpometacarpi in palmar view, tarsometatarsus in cranial view. Black and white arrows in (d) indicate the ridge-like process on metatarsal IV and the dorsal tubercle on metatarsal III, respectively. Abbreviations: cb, cushion-like bone; ep, extensor process; pp, pisiform process. Roman numerals in (d) identify metatarsals. Scale bars: 0.25 cm.

Figure 8. Cladogram of Mesozoic birds showing the systematic position of *Confuciusornis shifan*, and representing the strict consensus of the 192 most parsimonious trees recovered in the phylogenetic analysis performed in this study (length = 1404; consistency index = 0.277; retention index = 0.667). The bootstrap and bremer values over the the minimum threshold are given in normal font and bold italic font, respectively, near the nodes to which they pertain.

Figure 9. Positions of 11 confuciusornithid specimens in a previously published morphospace defined by wing loading (WL) and aspect ratio (AR), showing the areas of morphospace occupied by extant birds with particular modes of flight⁶². Specimens are indicated by black circles, and numbered in descending order of estimated body mass. Specimens are holotypes of their respective species unless a number is indicated.

A new confuciusornithid bird with a secondary epiphyseal ossification reveals phylogenetic changes in confuciusornithid flight mode

Renfei Wang^{1,2}, Dongyu Hu^{2*}, Meisheng Zhang¹, Shiyong Wang², Qi Zhao^{3,4}, Corwin Sullivan^{5,6} & Xing Xu^{3,4*}

¹ College of Earth Sciences, Jilin University, Changchun 130061, China

² Shenyang Normal University, Paleontological Museum of Liaoning, Key Laboratory for Evolution of Past Life in Northeast Asia, Liaoning Province, 253 North Huanghe Street, Shenyang 110034, China

³ Key Laboratory of Vertebrate Evolution and Human Origins, Institute of Vertebrate Paleontology and Paleoanthropology, Chinese Academy of Sciences, Beijing 100044, China

⁴ Center for Excellence in Life and Paleoenvironment, Chinese Academy of Sciences, Beijing 100044, China

⁵ Department of Biological Sciences, University of Alberta, Edmonton, AB T6G 2E9, Canada

⁶ Philip J. Currie Dinosaur Museum, Wembley, AB T0H 3S0, Canada

Correspondence and requests for materials should be addressed to D.H. (email: hudongyu@synu.edu.cn) and X.X. (email: xuxing@ivpp.ac.cn)

The confuciusornithids are the earliest known beaked birds, and constitute the only species-rich ~~and morphologically diverse~~ clade of Early Cretaceous pygostylian birds that existed prior to the cladogenesis of Ornithothoraces ~~two major avialan groups Enantiornithes and Ornithuromorpha~~. Here, we report a new confuciusornithid species from the Lower Cretaceous of western Liaoning, northeastern China. Compared to other confuciusornithids ~~and most contemporaneous enantiornithines and ornithuromorphs~~, this new species and the recently reported *Yangavis confucii* both show evidence of ~~stronger~~ higher level of stronger flight capability ~~adaptation~~, although the wings of the two taxa differ from one another in many respects. Our aerodynamic analyses under phylogeny ~~of the known confuciusornithids~~ indicate that varying modes of flight adaptation ~~considerable variations of flight capability and style emerged across the diversity of confuciusornithids~~, and to a lesser degree over the course of their ontogeny, and specifically suggest that both a trend towards improved flight capability and a change in flight strategy occurred in confuciusornithid evolution. The new confuciusornithid differs most saliently from other Mesozoic birds in having an extra cushion-like bone in the first digit of the wing ~~a secondary epiphyseal ossification, located in the alular digit~~, a highly unusual feature that may have helped to meet the functional demands of flight at a stage when skeletal growth was still incomplete. The new find strikingly exemplifies the morphological, developmental and functional diversity of the first beaked birds.

Introduction

Confuciusornithidae is a clade of Early Cretaceous pygostylian birds known from the Jehol Biota of East Asia¹, and representing the earliest known toothless, beaked birds. Five genera and eleven species, recovered from the Dabeigou, Yixian and Jiufotang formations (~135~120 Ma), have been described and assigned to this family, though the validity of some species is questionable²⁻⁶. Confuciusornithids are the only species-rich, ~~morphologically diverse pygostylian avialan~~ clade known to have existed prior to the cladogenesis of Ornithothoraces, ~~which the major groups Enantiornithes~~

~~and Ornithuromorpha~~, and are represented by thousands of exceptionally preserved specimens that collectively provide rich information on confuciusornithid morphology, taxonomy, flight ability, growth, diet and ecology^{3,5,7-13}. Here, we report a new confuciusornithid species, *Confuciusornis shifan* sp. nov., from the Jiufotang Formation. *Confuciusornis shifan* differs from other confuciusornithids in a number of morphological and developmental features, which have implications for understanding confuciusornithid taxonomic diversity, morphological disparity, development, and flight behavior.

Institutional abbreviations

DNHM, Dalian Natural History Museum (Dalian); GMV, National Geological Museum (Beijing); IVPP, Institute of Vertebrate Paleontology and Paleoanthropology (Beijing); MCFO, CosmoCaixa (Madrid); PMoL, Paleontological Museum of Liaoning (Shenyang).

Results

Systematic paleontology.

Avialae Gauthier, 1986

Aves Linnaeus, 1758

[revised manuscript text omitted]

~~from a pre-existing sesamoid as a “traction epiphysis”^{38,43,44}, or may represent a true secondary epiphyseal ossification comparable to those present in mammals and most lizards⁴⁵.~~ Furthermore, the tarsal elements and some carpal elements of extant birds fuse to adjacent long bones in a manner superficially resembling the fusion of epiphyseal ossifications to metaphyses, and apophyseal ossifications are also known in some species^{72,74}.

Epiphyseal SOC's have been demonstrated to provide additional stiffness to articular portions of bones that require reinforcement, for example as a result of stresses associated with support and locomotion in the terrestrial environment, and to facilitate the formation of topographically complex joint surfaces^{65,76,77}. To our knowledge, however, epiphyseal SOC's do not occur in non-avian archosauriforms, including in large, undoubtedly terrestrial dinosaurs that would have experienced substantial locomotor stresses. Epiphyseal SOC's probably evolved independently in mammals, birds and lepidosaurs⁶⁹, for reasons potentially relating to patterns of skeletal growth as well as to mechanical factors. Furthermore, we are unaware of any previous report of an epiphyseal SOC in a non-ornithomorph bird. The proximal tibial center appears to have been present in hesperornithiforms based on differences between adult and juvenile tibiotarsi of the Early Cretaceous genus Enaliornis⁷⁸, but it is surprising that the holotype of *C. shifan* possesses an epiphyseal SOC anywhere in the skeleton, let alone at a location – the cranial portion of the distal end of the alular metacarpal – where no such feature occurs even in crown-group birds.

The SOC in the distal epiphysis of the alular metacarpal of *C. shifan* (Fig. 7a,b) is presumably an autapomorphic feature, whose presence may be explicable on the basis of a combination of growth strategy and functional demands on the wing. A recent study has demonstrated that epiphyseal SOC's tend to evolve as a response to high mechanical stress⁷⁷. ~~For example, epiphyseal SOC's are developed only in the legs and thumbs of newborn bats, which are used from birth to cling to the mother or the roost⁷⁷. In *C. shifan* the distal end of the alular metacarpal may well have begun to experience considerable stresses as soon as a developing juvenile began to fly.~~ In extant birds the alular metacarpal plays an important role in aerial maneuvers^{58,59}, and

the wings of *C. shifan* appear better-adapted for high-maneuverability flight than those of other confuciusornithids ~~for high-maneuverability flight in relatively closed environments~~. As a result, the distal end of the alular metacarpal of *C. shifan* may well have begun to experience~~been more frequently subject to large~~ considerable stresses ~~in *C. shifan* than in most confuciusornithids~~ as soon as a developing juvenile began to fly. Furthermore, the histological features of *C. shifan* suggest a slower growth rate than that of other confuciusornithids, based on Amprino's rule⁷⁹ (Fig. 2). For example, *C. shifan* has longitudinally oriented vascular canals in the middle layer of the femoral compact bone (Fig. 2), whereas concentrically oriented canals are more numerous in other confuciusornithids¹³. If the spread of ossification from the primary centers of the long bones into the epiphyses was also slow, then the establishment of an epiphyseal SOC might have been necessary to reinforce the distal end of the alular metacarpal at a comparatively early ontogenetic stage, when the juvenile was beginning to fly but skeletal growth was still incomplete. The holotype of *C. shifan*, however, clearly represents a mature individual in which the caudal distal condyle of the alular metacarpal had fully ossified by the time of death, presumably via the normal mechanism of spread from the diaphyseal POC. ~~We infer that the secondary epiphyseal ossification would have fused with the rest of the alular metacarpal to remodel a cranial distal condyle of more normal appearance, had the individual lived longer.~~

[revised manuscript text omitted]

32. Xu, X., Chen, Y. N., Wang, X. L. & Chang, C. Pygostyle-like Structure from *Beipiaosaurus* (Theropoda, Therizinosauroida) from the Lower Cretaceous Yixian Formation of Liaoning, China. *Acta. Geol. Sin.* **77**, 294–298 (2003).
33. Barsbold, R. et al. A pygostyle from a non-avian theropod. *Nature* **403**, 155–156 (2000).
34. Persons, W. S., Currie, P. J. & Norell, M. A. *Oviraptorosaurus* tail forms and functions. *Acta. Palaeont. Pol.* **59(3)**, 553–567 (2014).
35. Zhang, F. C., Zhou, Z. H., Xu, X., Wang, X. L. & Sullivan, C. A bizarre Jurassic maniraptoran from China with elongate ribbon-like feathers. *Nature* **455**, 1105–1108 (2008).
36. Wang, M., O'Connor, J. K., Xu, X. & Zhou, Z. H. A new Jurassic scansoriopterygid and the loss of membranous wings in theropod dinosaurs. *Nature* **569**, 256–259 (2019).
37. Pittman, M. et al. In *Pennaraptoran theropod dinosaurs past progress and new frontiers* (eds. Pittman, M. & Xu, X.) Pennaraptoran Systematics (American Museum of Natural History, New York, 2020).
38. Rashid, D. J. et al. From dinosaurs to birds: a tail of evolution. *EvoDevo.* **5(25)**, 1–20 (2014).
39. Chiappe, L. M. & Walker, C. In *Mesozoic Birds: Above the Heads of Dinosaurs* (eds. Chiappe, L. M. & Witmer, L. M.) Skeletal morphology and systematics of the Cretaceous Enantiornithes (University of California Press, Berkeley, 2002).
40. Zhou, S., Zhou, Z. H. & O'Connor, J. K. Anatomy of the Early Cretaceous Archaeorhynchus spathula. *J. Vertebr. Paleontol.* **33**, 141–152 (2013).
41. Baumel, J. J., King, A. S., Breazile, J. E., Evans, H. E. & Berge, J. C. V. (eds.) *Handbook of Avian anatomy: Nomina Anatomica Avium*, 2nd edn. (Nuttall Ornithological Club, Cambridge, 1993).

[revised manuscript text omitted]

Figure 2. Photograph of a midshaft histological section of the right femur of the *Confuciusornis shifan* holotype (PMoL-AB00178). Arrows indicate LAGs. Abbreviations: ICL, inner circumferential layer; OCL, outer circumferential layer. Scale bar: 100 μ m.

Figure 3. Photograph of the skull and mandible of the *Confuciusornis shifan* holotype (PMoL-AB00178) in left ventrolateral view (a) with corresponding line drawing (b), and a close-up of the anterior margin of the orbit (c). White rectangle in (a) indicates the region shown in (c). A line drawing of the posterior half of the mandible of *Confuciusornis sanctus* IVPP V 13171 (drafted based on Figure 1B in Wang et al., 2018⁵) is presented below the line drawing of the skull and mandible of *Confuciusornis shifan*; arrow indicates the ventral process of the surangular. Abbreviations: an, angular; ar, articular; cmf, caudal mandibular fenestra; d, dentary; f, frontal; j, jugal; l, lacrimal; m, maxilla; o, orbit; rmf, rostral mandibular fenestra; n, nasal; p, parietal; pm, premaxilla; q, quadrate; sa, surangular; sp, splenial. Scale bars: 1 cm; note that (c) is not to scale.

Figure 4. Photographs of the cervical vertebrae (a), synsacrum (b) and pygostyle (c) of the *Confuciusornis shifan* holotype (PMoL-AB00178) (Also see Supplementary figure S1 for a close-up of the distal part of the pygostyle). Arrows in (b) and (c) indicate the longitudinal ridge on the ventral surface of the synsacrum and the positions of foramina along the pygostyle, respectively. Abbreviation: sy, synsacrum. Scale bars: 0.5 cm in (a) and (c); 1 cm in (b).

Figure 5. Photographs of the pectoral (a) and pelvic (b) girdles of the *Confuciusornis shifan* holotype (PMoL-AB00178). Abbreviations: fu, furcula; lc, left coracoid; lil, left ilium; lis, left ischium; ls, left scapula; pu, pubis; rc, right coracoid; rh, right humerus; ris, right ischium; rs, right scapula. Scale bars: 1 cm.

Figure 6. Line drawings of the pectoral girdles, forelimbs, and hindlimbs of the *Confuciusornis shifan* holotype (PMoL-AB00178) and other confuciusornithids, with shading to indicate lengths of stylopodial, zeugopodial and metapodial limb segments. Values near the segments represent ratios of segment length to femoral length. All drawings scaled to a common, arbitrary femoral length.

Figure 7. Photographs of the left (a) and right (b) carpometacarpi ~~and right tarsometatarsus (d)~~ of the *Confuciusornis shifan* holotype (PMoL-AB00178), the left alular metacarpal (c) of the confuciusornithid specimen PMoL-AB00150 and the right tarsometatarsus (d) of the *Confuciusornis shifan* holotype (PMoL-AB00178). Carpometacarpi in palmar view, tarsometatarsus in cranial view. **Black and white arrows in (d)** indicate the ridge-like process on metatarsal IV and the dorsal tubercle on metatarsal III, respectively. Abbreviations: cb, cushion-like bone; ep, extensor process; pp, pisiform process. Roman numerals in (d) identify metatarsals. Scale bars: 0.25 cm.

Figure 8. Cladogram of Mesozoic birds showing the systematic position of *Confuciusornis shifan*, and representing the strict consensus of the 192 most parsimonious trees recovered in the phylogenetic analysis performed in this study (length = 1404; consistency index = 0.277; retention index = 0.667). The bootstrap and bremer values over the the minimum threshold are given in normal font and bold italic font, respectively, near the nodes to which they pertain.

Figure 9. Positions of 11 confuciusornithid specimens in a previously published morphospace defined by wing loading (WL) and aspect ratio (AR), showing the areas of morphospace occupied by extant birds with particular modes of flight⁶². Specimens are indicated by black circles, and numbered in descending order of estimated body mass. Specimens are holotypes of their respective species unless a number is indicated.

Dear referees,

Based on referees' comments, we made further revisions. As you can see, we followed nearly all suggestions from the referees.

Below is a detailed response to the referees' comments, and hopefully the revised ms is acceptable to the referees.

Best regards,

REVIEWERS' COMMENTS:

Reviewer #1 (Remarks to the Author):

For this second round of review, the authors seem to be honest about their errors and weakness of their hypotheses and have tried their best to correct issues pointed out by reviewers. Their claims are more conservative and fair in the current edition.

As another reviewer pointed out, issues with confuciusornithid taxonomy still remain, and their phylogeny may not be as informative as it should be until these issues are thoroughly addressed. Nevertheless, the present study puts forward interesting hypotheses about adaptation to flight in these early avialans, to me especially in terms of the evolution of pygostyle. Therefore, this study is not only meaningful in presenting a new species of *Confuciusornis* to add to their morphological diversity, but also advocative in suggesting the need to revisit confucisornithid taxonomy for future to better understand evolution of non-ornithothoracines.

In this edition, I still found a few minor errors, which are highlighted in the attached PDF document. These errors are easy to fix and I do not need to check the manuscript again after this round of review as I know the authors would address these issues without problems. I think the manuscript is good to go for publication and am looking forward to see in it out.

Thanks for the positive comments, and we made further revisions following the referee's suggestions.

Reviewer #2 (Remarks to the Author):

The authors have greatly improved this work. The figures are much clearer after revisions, the diagnosis is more clear and better distinguished from *C. dui*, the discussion section is expanded with much interesting new analysis, and the additions to the supplement show the flight predictions have refreshingly tight

confidence intervals (this should be noted in the main text I think as it is a great boon to the work over similar studies).

Thanks for the positive comments. We highlighted the tight confidence intervals in the main text, following the referee's suggestion.

My suggested changes, marked on the manuscript PDF, are very minor. I believe the work is currently fit for publication after the minor changes I have suggested.

We followed the referee's suggestions and made all the revisions.

I do have one remaining major concern, and that is that I am unconvinced of the authors' arguments that this specimen should be referred to *Confuciusornis*. They argue on the basis that *shifan* is more similar to *Confuciusornis sanctus* than any other two *confuciusornithids* are to each other, but the same could be said for plenty of other Mesozoic birds assigned to different genera (e.g. *Bohaiornis* and *Parabohaiornis* vs all other *bohaiornithids*). The work itself acknowledges that additional taxonomic work may necessitate removing *C. shifan* from *Confuciusornis*, so I see no reason why it should be placed there in the first place. If future work affirms a high degree of similarity between *shifan* and *C. sanctus* there would remain no taxonomic issue if they were simply different genera that consistently resolved as sister taxa.

However, I must acknowledge that this is ultimately my preference, and the line between genus and species in paleontology is subjective. Referral of this specimen to *Confuciusornis* is not strictly incorrect at this stage, and it should not be the sole factor preventing the publication of this work.

We fully agree with the referee's comments on the taxonomy of early birds, and it is likely that *C. shifan* will be moved outside the genus *Confuciusornis* in future. However, our phylogenetic analysis places *C. shifan* adjacent to *C. sanctus*. Given this result, naming a new genus would probably be considered unacceptable by some Mesozoic bird researchers. If future phylogenetic analysis places this species outside *Confuciusornis*, however, we see no reason that a new genus name should not be proposed at that time.